# Information-Theoretic Generalization Bounds for VAEs:
# A Role of Encoder and Latent Variable

**Futoshi Futami** [* 1 2 3]   **Masahiro Fujisawa** [* 1 3 4]

## Abstract

Despite their remarkable success, a rigorous theoretical understanding of how latent variables (LVs) govern the generalization performance of Variational Autoencoders (VAEs) remains largely elusive. Existing theoretical analyses are confined to supervised learning or models with discrete latent spaces, leaving their role in standard VAEs with continuous LVs poorly understood. This paper establishes the first information-theoretic generalization analysis for VAEs by adapting a theoretical framework from supervised learning—the leave-one-out conditional mutual information framework—to the unsupervised, continuous latent space of these models. Our analysis reveals that their generalization error is bounded solely by the information complexity of the encoder and LVs, independent of the decoder. The versatility of our framework is demonstrated through its extension to both hierarchical VAEs, for which we provide layer-wise bounds, and data generation, where we link our information-theoretic principles to a novel bound on the 2-Wasserstein distance between true and generated distributions.

## 1. Introduction

Deep generative models with encoder-decoder architectures represent a cornerstone of modern machine learning, achieving remarkable success in tasks ranging from classification (Achille & Soatto, 2018b; Alemi et al., 2017) to data generation (Kingma, 2013; Van Den Oord et al., 2017). At their core lies a simple yet powerful principle: compressing

high-dimensional data into low-dimensional latent variables (LVs) via an encoder. The nature of this compressed representation is paramount, as it dictates the model's overall performance. Consequently, a vast body of work has sought to understand and control the information encoded in LVs. However, despite these efforts, a rigorous theoretical understanding of how LVs govern a model's generalization ability, particularly in the unsupervised setting, remains elusive.

In the supervised domain, the information bottleneck (IB) hypothesis (Tishby et al., 2000; Shamir et al., 2010) has served as a guiding principle. It posits that minimizing the mutual information (MI) between inputs and LVs improves generalization by ensuring that LVs retain only prediction-relevant information. The effectiveness of this principle has been validated in various applications (Tishby & Zaslavsky, 2015; Shwartz-Ziv & Tishby, 2017; Saxe et al., 2019; Achille & Soatto, 2018a). More recently, information-theoretic (IT) generalization analyses (Xu & Raginsky, 2017; Steinke & Zakynthinou, 2020) have refined this perspective. In particular, Kawaguchi et al. (2023) and Sefidgaran et al. (2023) rigorously clarified that generalization is not governed by MI alone, but also depends critically on the degree of encoder overfitting.

The theoretical landscape becomes particularly challenging in the unsupervised setting. Consider the variational autoencoder (VAE) (Kingma, 2013), a canonical example that learns a representation by minimizing a reconstruction loss alongside a KL divergence term that regularizes the LVs. Despite its empirical success, a theoretical grasp of how LVs contribute to its generalization remains severely limited. Existing PAC-Bayesian analyses (Chérief-Abdellatif et al., 2022; Mbacke et al., 2023), for example, either focus solely on network parameters—leaving the specific role of LVs unclear—or rely on restrictive assumptions such as a data-independent decoder. While Futami & Fujisawa (2025) successfully applied IT analysis to a vector-quantized VAE (VQ-VAE) (Van Den Oord et al., 2017) to show that its generalization is also governed by both MI and the the degree of encoder overfitting, their framework is fundamentally built upon the discreteness of the LVs. This reliance renders their analysis inapplicable to standard VAEs with continuous Gaussian LVs.

---

[1]The University of Osaka, Osaka, Japan [2]The University of Tokyo, Tokyo, Japan [3]RIKEN Center for Advanced Intelligence Project, Tokyo, Japan [4]Lattice Lab, Toyota Motor Corporation, Tokyo, Japan. Correspondence to: Futoshi Futami <futami.futoshi.es@osaka-u.ac.jp>, Masahiro Fujisawa <fujisawa@ist.osaka-u.ac.jp>.

*Proceedings of the 43$^{rd}$ International Conference on Machine Learning*, Seoul, South Korea. PMLR 306, 2026. Copyright 2026 by the author(s).

*Table 1.* Comparison with prior work on VAE generalization, highlighting that our analysis is the first to provide a comprehensive guarantee for standard and hierarchical VAEs without unrealistic assumptions.

| Study | Applicable Model | Bound Type | Assumption on Decoder | Analyzes Data Generation |
|---|---|---|---|---|
| Chérief-Abdellatif et al. (2022) | Standard VAE (Continuous LVs) | On all network parameters | Trained jointly | No |
| Mbacke et al. (2023) | Standard VAE (Continuous LVs) | Entangled (All parameters & LVs) | Data-independent | Yes |
| Futami & Fujisawa (2025) | VQ-VAE (Discrete LVs) | On Encoder & LVs only | Trained jointly | Yes |
| **Ours** | **Standard & Hierarchical VAE (Continuous LVs)** | **On Encoder & LVs only** | **Trained jointly** | **Yes** |

Given the success of IT analyses in clarifying the role of LVs in both supervised (Kawaguchi et al., 2023; Sefidgaran et al., 2023) and discrete unsupervised settings (Futami & Fujisawa, 2025), it is tempting to extend this perspective to standard VAEs. Doing so, however, is technically non-trivial and cannot be achieved by a straightforward adaptation of existing arguments. Specifically, prior frameworks either assume discrete LVs with finite cardinality (Kawaguchi et al., 2023; Futami & Fujisawa, 2025) or rely on label discreteness in classification settings (Sefidgaran et al., 2023). Then, they derive the IT bounds by exploiting discreteness to enumerate the finite set of patterns to which data points or LVs can correspond. As a result, these approaches cannot be directly applied to standard VAEs operating on continuous data and latent spaces. This leaves the generalization properties of one of the most fundamental unsupervised models unresolved—a challenge that becomes even more pronounced for expressive, hierarchically structured VAEs.

This work bridges this critical gap by developing the first IT generalization analysis explicitly designed for standard VAEs with continuous LVs. Grounded in the leave-one-out conditional mutual information (LOO-CMI) framework (Rammal et al., 2022; Haghifam et al., 2022), our analysis formulates priors and posteriors directly on the continuous latent space. Our main contributions are three-fold: First, we derive a decoder-independent generalization bound for standard Gaussian VAEs (Theorem 2). Crucially, this bound depends solely on the information complexity of the LVs and the degree of encoder overfitting, isolating the encoder's contribution from the decoder architecture. To make this analysis possible in the continuous setting, we introduce a novel **truncation-and-discretization argument** (Section 3.2) that rigorously handles continuous variables without relying on discrete data assumptions.

Second, we extend our framework to hierarchical VAEs (Theorem 3). This yields novel **layer-wise generalization bounds** that reveal a fundamental hierarchy: while the cumulative complexity of encoder parameters increases with depth, the information complexity of LVs provably decreases in higher, more abstract layers. This provides a theoretical explanation for the progressive information compression observed in hierarchical LV models.

Finally, to bridge representation learning and generation, we derive **an upper bound on the 2-Wasserstein distance**

**between the true data distribution and the generated distribution induced by the decoder** (Theorem 4). This result establishes a finite-sample, population-level guarantee on generative performance, linking it directly to the empirical reconstruction error and KL regularization term in the VAE objective. Together, these results provide the first unified IT analysis of generalization and generation for standard and hierarchical VAEs with continuous LVs.

## 2. Background

This section reviews the necessary background. We begin by formally defining the VAE in Section 2.1 and the reconstruction-based generalization error we aim to study in Section 2.2. Finally, we introduce the key information-theoretic tool for our main analysis: the LOO-CMI framework (Section 2.3).

**Notations:** We use uppercase letters for random variables and lowercase letters for their realizations. The distribution of $X$ is denoted by $p(X)$, and the conditional distribution of $Y$ given $X$ by $p(Y \mid X)$. Expectations are written as $\mathbb{E}_{p(X)}$ or $\mathbb{E}_X$. The MI and CMI are denoted by $I(X;Y)$ and $I(X;Y \mid Z)$, respectively. The KL divergence from $p(X)$ to $p(Y)$ is written as $\mathrm{KL}(p(X)\|p(Y))$. For $a \in \mathbb{N}$, we define $[a] := \{1, \ldots, a\}$.

### 2.1. Variational Autoencoder

Let $\mathcal{X} \subset \mathbb{R}^d$ denote the data space, equipped with the Euclidean metric $\|\cdot\|$, and let $\mathcal{D}$ be an unknown data-generating distribution over $\mathcal{X}$. The latent space is $\mathcal{Z} \subset \mathbb{R}^{d_z}$, also equipped with the Euclidean metric.

A VAE consists of an encoder, parameterized by $\phi \in \Phi \subset \mathbb{R}^{d_\phi}$, and a decoder, parameterized by $\theta \in \Theta \subset \mathbb{R}^{d_\theta}$. The encoder defines a variational posterior distribution $q(z \mid \phi, x)$ over the LV $z \in \mathcal{Z}$ for a given input $x \in \mathcal{X}$. We use a likelihood $p(x \mid \theta, z)$ that models the probability of reconstructing $x$ from $z$. We consider a standard VAE with a Gaussian prior $p(z) = \mathcal{N}(0, I_{d_z})$ on $\mathcal{Z}$. Here, $I_{d_z}$ denotes the identity matrix of size $d_z$. The variational posterior is also a Gaussian with a diagonal covariance matrix:

$$q(z \mid \phi, x) = \mathcal{N}(\mu_\phi(x), \mathrm{diag}(\sigma_\phi^2(x))),$$

where $\mathrm{diag}(\sigma_\phi^2(x))$ denotes the diagonal matrix with the vector $\sigma_\phi^2(x)$ on its main diagonal, and the mean $\mu_\phi : \mathcal{X} \to$

$\mathbb{R}^{d_z}$ and the standard deviation $\sigma_\phi : \mathcal{X} \to \mathbb{R}^{d_z}_{>0}$ are neural networks. The decoder is typically a deterministic function $g_\theta : \mathcal{Z} \to \mathcal{X}$ that parameterizes the mean of the likelihood distribution, e.g., a Gaussian or a Bernoulli distribution.

The per-sample loss for a data point $x$ is defined as follows.

$$\ell_{\mathrm{vae}}(x, \phi, \theta) := \mathbb{E}_{Z \sim q(z|\phi,x)}[-\log p(x \mid \theta, Z)] \\ + \beta \mathrm{KL}(q(z \mid \phi, x) \| p(z)).$$

The first term is the reconstruction loss, and the second is a regularization term weighted by $\beta > 0$ (Higgins et al., 2017). When $\beta = 1$, minimizing this loss is equivalent to maximizing the evidence lower bound (ELBO) on the marginal log-likelihood $\log p(x|\theta)$.

**Leave-one-out (LOO) setting:** Throughout this paper, we adopt the LOO setting to analyze this model. From a full dataset $X^n = \{X_1, X_2, \ldots, X_n\}$ $(n > 2)$ of i.i.d. samples, one data point is uniformly selected and removed as test data. The remaining $n - 1$ samples are used as the training data, denoted by $S = (X_1, \ldots, X_{n-1})$. The parameters are learned by minimizing the total loss on this set:

$$\ell_{\mathrm{vae}}(S, \phi, \theta) := \frac{1}{n-1} \sum_{i=1}^{n-1} \ell_{\mathrm{vae}}(X_i, \phi, \theta). \tag{1}$$

## 2.2. Generalization based on Reconstruction Loss

Hereafter, we denote the set of all model parameters as $W := (\phi, \theta) \in \mathcal{W} (:= \Phi \times \Theta)$. Given a training dataset $S$ drawn from $\mathcal{D}$, the parameters are learned by a randomized algorithm $\mathcal{A} : S \to \mathcal{W}$ that stochastically minimizes the VAE objective in Eq. (1). The output of the algorithm $W$ thus follows the conditional distribution $q(\phi, \theta \mid S)$.

The performance of a trained model is often measured by a reconstruction loss $\ell : \mathcal{X} \times \mathcal{X} \to \mathbb{R}^+$. For a given input $x$, the expected reconstruction loss over the variational posterior is defined as $\ell_0(w, x) := \mathbb{E}_{q(Z|\phi,x)}[\ell(x, g_\theta(Z))]$, where $\ell_0 : \mathcal{W} \times \mathcal{X} \to \mathbb{R}$. In this study, we specifically consider the squared Euclidean distance: $\ell_0(w, x) := \mathbb{E}_{q(Z|\phi,x)} \|x - g_\theta(Z)\|^2$.

Our goal is to theoretically analyze the generalization performance of VAEs. To this end, we define the generalization error as the difference between the expected population loss (test error) and the expected empirical loss (training error):

$$\mathrm{gen}(n, \mathcal{D}) := \Big| \underset{S,X}{\mathbb{E}} \underset{q(W|S)}{\mathbb{E}} \ell_0(W, X) - \frac{1}{n-1} \sum_{i=1}^{n-1} \ell_0(W, X_i) \Big|.$$

We analyze this generalization error by the IT analysis introduced in the next section.

## 2.3. Information-Theoretic Analysis over LOO Settings

Our analysis is grounded in the IT analysis based on the leave-one-out CMI (LOO-CMI) framework (Rammal et al., 2022; Haghifam et al., 2022). This setting involves a full dataset $X^n = \{X_1, X_2, \ldots, X_n\}$ of i.i.d. samples and a uniformly-sampled random index $U \sim \mathrm{Uniform}([n])$ over the indices of $X^n$. The training set is formed by removing the $U$-th sample, $S := X^n \setminus \{X_U\}$, and the algorithm is trained on $S$ to produce the parameters $W := A(S)$. The held-out sample $X_U$ is then used for testing. We refer to this as the LOO-CMI setting.

We impose the following assumption on the data space.

**Assumption 1.** *There exists a positive constant $\Delta$ such that $\sup_{x,x' \in \mathcal{X}} \|x - x'\| < \Delta^{1/2}$.*

This is a mild assumption for many practical domains (e.g., normalized images) and ensures the squared reconstruction loss is bounded by $\Delta < \infty$. The foundational result of the LOO-CMI framework is the following theorem.

**Theorem 1** (Rammal et al. (2022)). *Under the LOO-CMI setting with Assumption 1, we have*

$$\mathrm{gen}(n, \mathcal{D}) \leq \frac{n}{\sqrt{2}(n-1)} \sqrt{I(W; U \mid X^n)}.$$

Here, the generalization error is quantified by the conditional mutual information $I(W; U \mid X^n)$, which measures how much information the learned parameters $W$ retain about the index $U$ of the single held-out sample. The properties of this information term, including its sample complexity and relation to algorithmic stability, are further detailed in Haghifam et al. (2022) and Rammal et al. (2022).

**Motivation for our framework:** While Theorem 1 provides a general bound, its direct application fails to isolate the contribution of LVs in models like VAEs. The issue is that the bound depends on the entire set of learned parameters $W$, entangling the effects of the encoder, decoder, and LVs. This fundamental challenge—that standard IT analysis tools jointly measure the complexity of all model components—is not unique to LOO-CMI and hinders other frameworks (Steinke & Zakynthinou, 2020; Harutyunyan et al., 2021; Hellström & Durisi, 2022) as well. Therefore, a new analytical approach is required. The key insight, noted by Futami & Fujisawa (2025), is that such an approach requires a specific *symmetry* between training and test data for unsupervised models. The LOO-CMI setting naturally provides this crucial symmetry, making it the foundation upon which to build our analysis, as detailed in Section 3.

## 3. Analysis of VAEs

This section develops our main theoretical results. We first present our novel generalization bound for VAEs (Sec-

tion 3.1), then outline the proof to highlight the key ideas of our approach, specifically the truncation-and-discretization argument (Section 3.2). We conclude by discussing how to numerically evaluate these terms and their connection to algorithmic stability (Section 3.3).

### 3.1. The Role of LVs in Generalization

First, we introduce a mild assumption on the encoder.

**Assumption 2.** *For any $x \in \mathcal{X}$ and any parameter $\phi \in \Phi$, we have $\|\mu_\phi(x)\|_2 < \infty$ and $0 < \sigma_{\phi,i}(x) < \infty$ for all $i \in [d_z]$, where $\sigma_{\phi,i}(x)$ denotes the $i$-th element of $\sigma_\phi(x)$.*

This assumption ensures that the posterior mean is bounded and the posterior variance is strictly positive. It is not restrictive in practice because: (i) if it fails, the VAE objective in Eq. (1) typically diverges; and (ii) the instance space $\mathcal{X}$ is assumed to be bounded. We also emphasize that, unlike existing analysis (Mbacke et al., 2023), our framework does not require Lipschitz continuity of the encoder or decoder networks (See Appendix C for further discussion).

We now present our main result for Gaussian VAEs:

**Theorem 2.** *Under Assumptions 1 and 2, and in the LOO-CMI setting, the VAE model satisfies:* $\mathrm{gen}(n, \mathcal{D}) \leq \frac{\Delta}{\sqrt{n-1}} +$

$$\frac{3n\Delta}{\sqrt{2}(n-1)}\sqrt{I(Z^n; U \mid \phi, X^n) + I(\phi; U \mid X^n)}.$$

The complete proof is provided in Appendix C, and a proof sketch highlighting our key technical contribution is given in Section 3.2.

Theorem 2 provides an decoder-independent characterization of *the upper bound* of the generalization error. Crucially, it isolates the generalization error attributable to the complexity of the LVs and the encoder, separating these factors from the expressiveness of the decoder. This perspective is consistent with prior IT analysis (Kawaguchi et al., 2023; Sefidgaran et al., 2023; Futami & Fujisawa, 2025).

Our bound decomposes the generalization gap into two distinct information-theoretic terms. The first term,

$$I(Z^n; U \mid \phi, X^n) = H[U \mid \phi, X^n] - H[U \mid \phi, X^n, Z^n],$$

captures how much information the LVs retain about the identity of the training and the test samples. Intuitively, it quantifies how sensitive the latent representation is to which sample plays the test role in the LOO construction, and thus captures a form of index memorization through the LVs.

The second term, $I(\phi; U \mid X^n)$, quantifies how much information the training data contributes to learning the encoder parameters, analogous to the result in Theorem 1. We discuss how this term relates to *a flat minimum* in Section 3.3.

Conceptually, these two terms play roles analogous to the "representation complexity" and "parameter overfitting" components in Kawaguchi et al. (2023) and Futami & Fujisawa (2025). We discuss the dependence on the sample size $n$ under additional assumptions in Appendix I.2.

**Why leave-one-out:** While Theorem 2 is stated for the LOO split (one test sample), the same proof strategy extends to selecting a *fraction* $c \in (0, 1)$ of the samples as test data, i.e., $cn$ points (assumed integer) without replacement. For example, $c = 1/2$ corresponds to the regime considered in Futami & Fujisawa (2025). However, the LOO formulation is particularly significant for two reasons. First, it supports stability-based arguments (Section 3.3). Second, the index $U$ remains one-dimensional, which makes the MI terms computationally tractable, as shown in Section 3.4. See Appendix I.1 for a detailed discussion of fractional splits.

Finally, as emphasized in prior work (Futami & Fujisawa, 2025), our result does not imply that the decoder is irrelevant. Recall that the test loss decomposes as: **(Test Loss)** = **(Training Loss)** + **(Gen. Gap)**. Our analysis specifically bounds the generalization gap component. Therefore, a sufficiently expressive decoder remains essential to minimize the training loss, thereby ensuring overall model performance.

### 3.2. Truncation-and-discretization argument

This section presents our main technical contribution: a **truncation-and-discretization** argument that enables IT generalization analysis for VAEs with *continuous* LVs. As discussed in Section 1, existing IT analyses of encoder models fundamentally rely on the *discreteness* of LVs or data. This discreteness is essential for deriving IT bounds and, consequently, such analyses are not directly applicable to standard Gaussian VAEs.

To overcome this limitation, we adapt a strategy inspired by classical information-theoretic arguments for continuous channels (Cover & Thomas, 2012; El Gamal & Kim, 2011). In particular, analyses of Gaussian channels often proceed by first quantizing the signal, establishing results in a discrete setting, and then recovering the continuous case via a limiting argument. We adopt the same philosophy for generalization analysis.

At a high level, our proof consists of three ingredients: (i) truncating the Gaussian LVs to a bounded region, (ii) discretizing the truncated variables using a fixed codebook, and (iii) carefully taking limits so that no artificial bias is introduced into the generalization bound.

**Key technical challenges.** Applying this idea to generalization bounds presents two nontrivial challenges. First, the truncation and discretization procedures must be *data-independent*. If they depended on the training data, they

could introduce additional information leakage unrelated to the original problem, thereby spurious bias is introduced into the generalization bound. Second, we must rigorously control the approximation error induced by truncation and discretization, and show that it vanishes in the appropriate limits while preserving the IT structure of the bound.

Together, these steps yield a valid and novel IT analysis for continuous VAEs. The logical flow of the proof is outlined below (the complete proof is provided in Appendix C).

**Proof outline. Step 1 (Truncation):** We first truncate the LV, $Z$, by restricting it to a ball $B(R) := \{z \in \mathbb{R}^{d_z} : \|z\|_2 \leq R\}$, denoting the truncated variable as $Z_R$. This step is necessary because Gaussian LVs are supported on an unbounded domain, making direct discretization infeasible.

**Step 2 (Discretization):** We discretize $Z_R$ by assigning it to the nearest centroid in a *fixed, data-independent codebook* $\{c_1, \ldots, c_K\} \subset B(R)$. This induces a discrete index $J := \pi(Z_R) \in \{1, \ldots, K\}$ and a corresponding discretized LV $\hat{Z} = c_J$. Crucially, both the truncation radius $R$ and the codebook are chosen independently of the training data to avoid introducing spurious data dependence.

**Step 3 (Error Decomposition):** The difference between the loss evaluated using the original $Z$ and the discretized variable $\hat{Z}$ decomposes into two terms: (i) a truncation error, controlled by the tail probability $\Pr(\|Z\|_2 > R)$, and (ii) a discretization error, controlled by the quantization gap $\|Z_R - c_J\|$. Following the moment and tail arguments in Telgarsky & Dasgupta (2013), both terms can be made arbitrarily small by taking the limits $R \to \infty$ and $K \to \infty$, ensuring the two losses become asymptotically equivalent.

**Step 4 (Discrete Bound and Data Processing Inequality):** Given this equivalence, we apply the LOO-CMI generalization analysis to the discretized representation $\hat{Z}$. This yields a bound of the form $\mathrm{gen}(\hat{Z}) \lesssim \frac{\Delta}{\sqrt{n-1}} + \frac{3n\Delta}{\sqrt{2}(n-1)} \times$

$$\sqrt{I(\phi; U \mid X^n) + I(J^n; U \mid \phi, X^n)}.$$

The next step is to relate the discrete complexity term $I(J^n; U \mid \phi, X^n)$ to the original $Z$. Conditioned on $(\phi, X^n)$, since the truncation and discretization define the Markov chain $U \to Z \to Z_R \to J$, the data processing inequality guarantees that:

$$I(J^n; U \mid \phi, X^n) \leq I(Z^n; U \mid \phi, X^n).$$

Finally, taking the limits $(K, R) \to \infty$ eliminates the approximation errors while the information-theoretic bound holds, completing the proof. $\qquad\square$

### 3.3. Encoder Complexity and Connection to Flatness

A key advantage of our LOO framework is that it allows us to evaluate the encoder complexity term, $I(\phi; U \mid X^n)$, via a quantity directly related to the geometry of the loss landscape. This property is crucial for the numerical experiments presented in Section 3.4. Following the approach of Rammal et al. (2022), we compute a proxy for $I(\phi; U \mid X^n)$ using influence functions. The detailed derivations are provided in Appendix E.

The core idea is to estimate the change in the optimal encoder parameters, $\phi^*$, when a single sample $x_i$ is removed from the training set $x^n$. Let $w^* = (\phi^*, \theta^*) = \arg\min_{w \in \mathcal{W}} \ell_{\mathrm{vae}}(x^n, w)$ be the parameters learned on the full dataset, and let $\phi_i^*$ be the parameters learned on $x^n \setminus \{x_i\}$. Using influence functions (Koh & Liang, 2017), this perturbation, $\phi_i^* - \phi^*$, can be approximated as:

$$\phi_i^* - \phi^* \approx \frac{1}{n}(H_{w^*}^{\phi})^{-1} \nabla_\phi \ell_{\mathrm{vae}}(x_i, w^*) =: \delta_{\phi_i},$$

where $H_{w^*}^{\phi}$ is the Hessian of the VAE loss in Eq. (1) projected onto the encoder parameters. By applying the noise-calibrated bound from Rammal et al. (2022) (see Appendix E) with a noise covariance $\Sigma = (H_{w^*}^{\phi})^{-1}$, we arrive at the following computable upper bound:

$$I(\phi_\Sigma^*; U \mid x^n) \lesssim -\frac{1}{n}\sum_{i=1}^n \ln \frac{1}{n}\sum_{j=1}^n e^{-\frac{1}{2}\|\delta_{\phi_i} - \delta_{\phi_j}\|_{H_{w^*}^{\phi}}^2},$$

where $\phi_\Sigma^* = \phi^* + \epsilon$ with $\epsilon \sim \mathcal{N}(0, \Sigma)$ and $\|\delta_{\phi_i} - \delta_{\phi_j}\|_{H_{w^*}^{\phi}} := (\delta_{\phi_i} - \delta_{\phi_j})^\top H_{w^*}^{\phi}(\delta_{\phi_i} - \delta_{\phi_j})$.

This formulation provides a direct theoretical link between VAE generalization and the geometry of the loss landscape. The squared Mahalanobis distances $\|\delta_{\phi_i} - \delta_{\phi_j}\|_{H_{w^*}^{\phi}}^2$ indicates that a solution at a **flat minimum**—where the Hessian $H_{w^*}^{\phi}$ has a small norm—yields a smaller information term, thereby implying better generalization. This finding theoretically grounds the empirical observation that flat minima correlate with good generalization, a well-established concept in supervised learning (Hochreiter & Schmidhuber, 1997; Dziugaite & Roy, 2017). Our result suggests that the generalization of VAEs is similarly governed by the flatness of the learned encoder. We emphasize that this quantity is not an unbiased estimator of the exact conditional mutual information. Rather, it serves as a *computable proxy* in our numerical experiments to evaluate the encoder's contribution to the generalization gap.

### 3.4. Numerical Experiments

To study the diagnostic relevance of the information-theoretic quantities in Theorem 2, we train VAEs using importance-weighted autoencoder (IWAE) based estimator (Burda et al., 2015) on MNIST and Fashion-MNIST. We

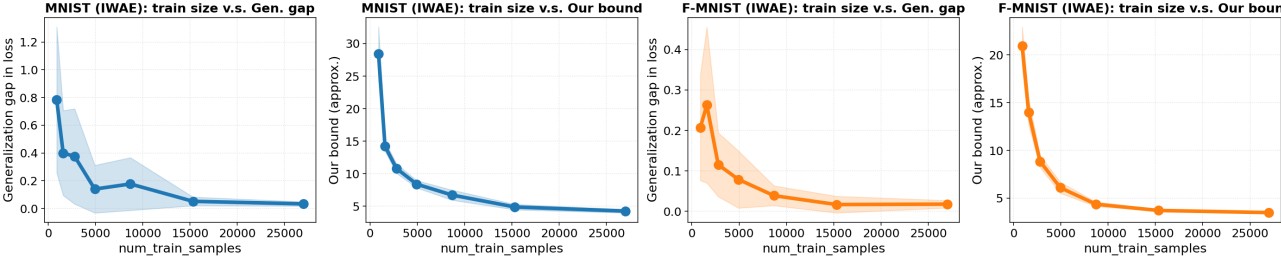

*Figure 1.* Empirical generalization gap (left) and the approximated theoretical bound (Theorem 2, right) for IWAE on MNIST (top) and Fashion-MNIST (bottom), plotted against the training set size. Shaded regions denote mean $\pm$ one standard deviation over multiple runs.

employ an MLP-based encoder with four hidden layers, a symmetric decoder of the same depth, and a 32-dimensional latent space ($d_z = 32$). Rather than aiming to numerically verify the tightness of the bound, our goal is to examine whether changes in the empirical generalization gap are qualitatively reflected by the corresponding information-theoretic terms. To this end, we compare the empirical generalization gap with computable proxies of the two complexity terms in Theorem 2.

For the encoder complexity term $I(\phi; U \mid X^n)$, we use the influence-function-based proxy described in Section 3.3. For the latent complexity term $I(Z^n; U \mid \phi, X^n)$, we evaluate a computable upper bound obtained by assuming a uniform distribution over the index variable $U$, which is justified by the weakly symmetric structure of the induced channel (Appendix J.5). For simplicity, we reuse the same notation for these computable quantities in the following. Full experimental details are provided in Appendix J.

Figure 1 visualizes the resulting trends. As the training set size $n$ increases, the empirical generalization gap decreases, and the proposed CMI quantities exhibit consistent qualitative behavior. This supports the view that the bound components capture meaningful aspects of the mechanisms governing generalization in VAEs. To further quantify this relationship, we perform a correlation analysis at $n = 30,000$, summarized in Table 2. For this correlation analysis, each data point corresponds to one cross-validation split. At $n = 30,000$, for each split we compute four scalar quantities: the empirical gap $|L_{\text{test}} - L_{\text{train}}|$, the influence-function-based proxy for $I(\phi; U \mid X^n)$, the computable upper bound for $I(Z^n; U \mid \phi, X^n)$, and the combined quantity $\sqrt{I(Z^n; U \mid \phi, X^n) + I(\phi; U \mid X^n)}$. The Pearson, Spearman, and Kendall coefficients in Table 2 are computed across these split-wise scalar values. Thus, the table should be interpreted as a diagnostic correlation analysis based on computable proxies and upper bounds, rather than as an exact direct estimation of the mutual information terms. On MNIST, $I(\phi; U \mid X^n)$ dominates the correlation, while $I(Z^n; U \mid \phi, X^n)$ shows almost no correlation. On Fashion-MNIST, however, neither component is sufficient

*Table 2.* Correlation between generalization gap in loss and our bounded components at $n = 30000$ under the IWAE experiments.

| Dataset | Metric | Pearson (↑) | Spearman (↑) | Kendall (↑) |
|---|---|---|---|---|
| MNIST | $I(\phi; U \mid X^n)$ | 0.373 | **0.406** | **0.333** |
| MNIST | $I(Z^n; U \mid \phi, X^n)$ | 0.033 | 0.091 | 0.067 |
| MNIST | $\sqrt{I(Z^n; U \mid \phi, X^n) + I(\phi; U \mid X^n)}$ | **0.374** | **0.406** | **0.333** |
| F-MNIST | $I(\phi; U \mid X^n)$ | 0.012 | 0.030 | 0.067 |
| F-MNIST | $I(Z^n; U \mid \phi, X^n)$ | 0.187 | 0.042 | 0.022 |
| F-MNIST | $\sqrt{I(Z^n; U \mid \phi, X^n) + I(\phi; U \mid X^n)}$ | **0.194** | **0.152** | **0.111** |

individually, and their combination achieves the strongest correlation with the true gap.

We further evaluate the decomposition in Theorem 2 by changing the strength of the latent regularization in standard $\beta$-VAEs. The purpose of this experiment is to test a source of variation different from the training-size sweep above: rather than varying $n$, we fix the architecture, training protocol, training size $n = 30,000$, and evaluation procedure, and vary only the KL weight $\beta \in \{0.01, 0.03, 0.3, 1.0, 3.0, 10.0\}$. Since $\beta$ directly controls the regularization of LVs, this setting allows us to examine whether the encoder term and the LV term reflect changes in the reconstruction generalization gap induced through the latent channel.

For each dataset and each value of $\beta$, we compute ten split-wise pairs between the empirical reconstruction gap $|L_{\text{test}}^{\text{recon}} - L_{\text{train}}^{\text{recon}}|$ and each of the three computable quantities in Theorem 2: the influence-function-based proxy for $I(\phi; U \mid X^n)$, the computable upper bound for $I(Z^n; U \mid \phi, X^n)$, and the combined quantity $\sqrt{I(Z^n; U \mid \phi, X^n) + I(\phi; U \mid X^n)}$. We pool the resulting $6 \times 10$ pairs for each dataset and report Pearson, Spearman, and Kendall correlations in Table 3. This pooled evaluation assesses whether each quantity tracks the empirical gap across the entire $\beta$ sweep, whereas computing correlations separately for each fixed $\beta$ primarily measures split-to-split variability under a fixed latent capacity.

Table 3 shows that the computable quantity corresponding to $I(Z^n; U \mid \phi, X^n)$ is strongly and positively correlated with the empirical reconstruction gap on all three datasets, and that the combined quantity gives comparable or stronger cor-

*Table 3.* Correlation between the reconstruction generalization gap and each computable information-theoretic component under the $\beta$-VAE sweep with $\beta \in \{0.01, 0.03, 0.3, 1.0, 3.0, 10.0\}$.

| Dataset | Metric | Pearson (↑) | Spearman (↑) | Kendall (↑) |
|---|---|---|---|---|
| MNIST | $I(\phi; U \mid X^n)$ | 0.434 | 0.567 | 0.374 |
| MNIST | $I(Z^n; U \mid \phi, X^n)$ | **0.930** | 0.775 | 0.529 |
| MNIST | $\sqrt{I(Z^n; U \mid \phi, X^n) + I(\phi; U \mid X^n)}$ | 0.928 | **0.780** | **0.531** |
| F-MNIST | $I(\phi; U \mid X^n)$ | -0.559 | -0.800 | -0.628 |
| F-MNIST | $I(Z^n; U \mid \phi, X^n)$ | 0.904 | **0.932** | **0.767** |
| F-MNIST | $\sqrt{I(Z^n; U \mid \phi, X^n) + I(\phi; U \mid X^n)}$ | **0.961** | 0.914 | 0.738 |
| CIFAR-10 | $I(\phi; U \mid X^n)$ | -0.332 | -0.857 | -0.679 |
| CIFAR-10 | $I(Z^n; U \mid \phi, X^n)$ | 0.923 | **0.883** | **0.714** |
| CIFAR-10 | $\sqrt{I(Z^n; U \mid \phi, X^n) + I(\phi; U \mid X^n)}$ | **0.929** | 0.846 | 0.667 |

relations. By contrast, the influence-function-based proxy for $I(\phi; U \mid X^n)$ is dataset-dependent and becomes negatively correlated on Fashion-MNIST and CIFAR-10. These results suggest that, when the generalization behavior is changed by directly manipulating the latent regularization, the LV term plays a central role in explaining the observed variation. Thus, the $\beta$-VAE experiment provides an additional diagnostic check of the encoder–LV decomposition.

Finally, we emphasize that these results rely on computable upper bounds rather than exact ones. As detailed in Appendix K, for simpler models or specific regimes (e.g., near-deterministic encoders), the upper bound on the LV term can become loose. This highlights the development of tighter estimators—potentially exploiting the geometric structure of the latent space—as a crucial direction for future work.

# 4. Analysis of Hierarchical VAEs

We now extend our analysis to hierarchical VAEs. These models employ a stack of stochastic LVs to learn rich data representations and are known to be highly effective in practice (Vahdat & Kautz, 2020; Burda et al., 2015). Our goal is to apply our IT framework to elucidate how this hierarchical structure governs generalization, specifically through a *layer-wise decomposition*.

## 4.1. Model Setup and Generalization Error

A hierarchical VAE consists of $L > 1$ stochastic layers. Each layer $l \in [L]$ induces a latent space $\mathcal{Z}_l \subset \mathbb{R}^{d_{z_l}}$ equipped with the Euclidean norm $\|\cdot\|$. For $l \in [L]$, the encoder defines a conditional distribution $q_l(z_l \mid \phi_l, z_{l-1}) = \mathcal{N}(\mu_{\phi_l}(z_{l-1}), \text{diag}(\sigma_{\phi_l}^2(z_{l-1})))$, where we set the input space as the base layer, $\mathcal{Z}_0 = \mathcal{X}$, and define $\mu_{\phi_l} : \mathcal{Z}_{l-1} \to \mathcal{Z}_l$ and $\sigma_{\phi_l} : \mathcal{Z}_{l-1} \to \mathbb{R}^{d_{z_l}}_{>0}$.

The decoder reconstructs the input from the final stochastic layer $z_L$. For notational convenience, we treat the reconstruction step as an additional layer $z_{L+1} \in \mathcal{X}$, with likelihood $q(x \mid \phi_{L+1}, z_L)$. With this notation, a unified formulation becomes possible. For the likelihood, typical choices include, for example, a Dirac measure of the form $q(x \mid \phi_{L+1}, z_L) = \delta(x - \mu_{\phi_{L+1}}(z_L))$, which yields a de-

terministic decoder, as well as a Gaussian likelihood given by $q(x \mid \phi_{L+1}, z_L) = \mathcal{N}(\mu_{\phi_{L+1}}(z_L), \text{diag}(\sigma_{\phi_{L+1}}^2(z_L)))$. We denote the cumulative parameters up to layer $l$ by $\phi_{1:l} = (\phi_1, \ldots, \phi_l)$ for $l = 1, \ldots, L+1$.

We assume that the parameters are obtained via a randomized training algorithm. We impose the following assumption, which is the hierarchical analogue of Assumption 2.

**Assumption 3.** *For all $l \in [L]$, for any input $z_{l-1} \in \mathcal{Z}_{l-1}$ and parameters $\phi_l \in \Phi_l$, we have $\|\mu_{\phi_l}(z_{l-1})\|_2 < \infty$ and $0 < \sigma_{\phi_l,i}(z_{l-1}) < \infty$ for all $i \in [d_{z_l}]$, where $\sigma_{\phi_l,i}(z_{l-1})$ denotes the $i$-th component of $\sigma_{\phi_l}(z_{l-1})$.*

The reconstruction loss for this model is defined over the entire generative path as $\ell_0(\phi_{1:L+1}, x) := \mathbb{E}_{q(z_{1:L+1}|x)}\|x - Z_{L+1}\|^2$, where $q(z_{1:L+1} \mid x) := \prod_{l=1}^{L+1} q_l(z_l \mid \phi_l, z_{l-1})$ with $z_0 = x$. The generalization error is then defined as

$$\text{gen}_H(n, \mathcal{D}) := \Big| \mathbb{E}_{S,X} \mathbb{E}_{q(\phi_{1:L+1}|S)} \ell_0(\phi_{1:L+1}, X)$$
$$- \frac{1}{n-1} \sum_{i=1}^{n-1} \ell_0(\phi_{1:L+1}, X_i) \Big|.$$

## 4.2. Layer-Wise Generalization Bound

Our main result for hierarchical VAEs provides a *separate generalization bound for each layer*, decomposing the error into layer-specific information-theoretic terms.

**Theorem 3.** *Under Assumptions 1 and 3, and in the LOO-CMI setting, the hierarchical VAE model satisfies for each layer $l \in [L]$:*

$$\text{gen}_H(n, \mathcal{D}) \leq \frac{\Delta}{\sqrt{n-1}} + \frac{3n\Delta}{\sqrt{2}(n-1)} \times$$
$$\sqrt{I(Z_l^n; U \mid \phi_{1:l}, X^n) + I(\phi_{1:l}; U \mid X^n)}.$$

The complete proof is provided in Appendix H. Theorem 3 offers several key insights. First, an immediate consequence is that the overall generalization error is bounded by the minimum of the layer-wise bounds:

$$\text{gen}_H(n, \mathcal{D}) \leq \min_{l \in [1,\ldots,L]} \frac{\Delta}{\sqrt{n-1}} + \frac{3n\Delta}{\sqrt{2}(n-1)} \times$$
$$\sqrt{I(Z_l^n; U \mid \phi_{1:l}, X^n) + I(\phi_{1:l}; U \mid X^n)}. \quad (2)$$

Equation (2) shows that, rather than accumulating across layers, the generalization bound is governed by the layer that best balances latent information compression and encoder complexity. The following section provides a structural interpretation of this phenomenon and explains how this balance emerges across the hierarchy.

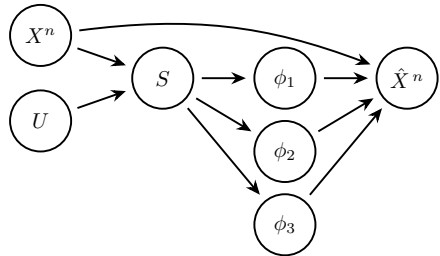

*(a)* The standard LOO-CMI bound (Theorem 1), which treats the entire model as a single block.

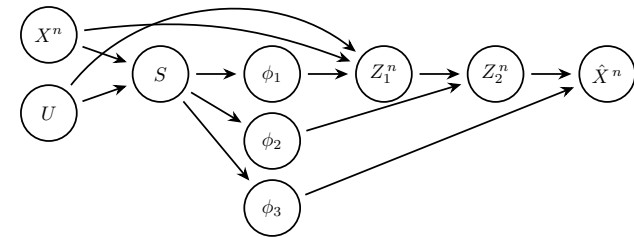

*(b)* Our proposed layer-wise bound (Theorem 3), which explicitly accounts for the LVs at each layer ($Z_1^n, Z_2^n$).

*Figure 2.* Graphical models illustrating the difference between the standard LOO-CMI analysis and our proposed layer-wise approach, for a hierarchical VAE with $L = 2$ layers. The reconstructed data is denoted by $\hat{X}^n$.

### 4.3. Implications of the Layer-Wise Bound

This section develops the structural implications of the layer-wise generalization bound in Theorem 3. To build intuition, we consider a graphical model representation of a two-layer hierarchy ($L = 2$), illustrated in Figure 2. We contrast our approach (Figure 2b), which evaluates information flow at each latent layer, with the standard LOO-CMI analysis that treats the entire model as a single block (Figure 2a).

The two graphical models differ in how the leave-one-out index $U$ enters the analysis. In the standard LOO-CMI construction, $U$ specifies the held-out component for the entire prediction block: the training data $X^n$ are fed through the learned encoder–decoder mechanism to produce the reconstructed variables $\hat{X}^n$, and the whole block is treated as the object of the information-theoretic comparison. In contrast, our layer-wise construction keeps the parameter-learning block explicit and lets $U$ act on the latent-variable channel when defining the latent CMI term. Therefore, the edge $U \to Z_1^n$ in Figure 2b should be read as a latent-channel dependence: changing the leave-one-out index changes which latent variables are compared under the posterior/prior construction used in the Donsker–Varadhan argument, rather than globally shuffling the entire prediction block. The reconstruction mechanism itself remains mediated through the learned parameters and the latent variables. A detailed posterior/prior comparison is provided in Appendix H.3.

**Layer-wise trade-off.** Our analysis reveals a fundamental trade-off across the hierarchy between *latent information compression* and *encoder complexity accumulation*. As the layer index $l$ increases from 1 to 2, the LVs undergo progressive stochastic transformations. By the data processing inequality (Cover & Thomas, 2012), this implies a monotonic decrease of the latent information term: $I(Z_2^n; U \mid \phi_{1:2}, X^n) \leq I(Z_1^n; U \mid \phi_{1:2}, X^n)$. Intuitively, the injected noise at each layer suppresses sample-specific information, yielding increasingly compressed representations that are beneficial for generalization.

In contrast, the encoder complexity term grows with depth. Since the parameter set accumulates across layers, we necessarily have $I(\phi_{1:2}; U \mid X^n) \geq I(\phi_1; U \mid X^n)$. This reflects an increased capacity for overfitting resulting from the introduction of additional encoder parameters.

The layer-wise bound, therefore, exposes a structural trade-off: early layers retain more sample-specific information but involve fewer parameters, while deeper layers benefit from stronger compression at the cost of increased encoder complexity. The minimization over layers in Eq. (2) automatically selects the layer that best balances this trade-off.

**Empirical illustration.** We empirically examine these implications in Appendix K. Using a hierarchical IWAE trained on Fashion-MNIST, we evaluate the layer-wise computable bounds for each encoder layer ($l = 1, \ldots, 4$). The results show that the layer-wise bounds consistently align with the empirical generalization gap. Most notably, in accordance with the theoretical trade-off, we observe a clear reduction of the latent information term in deeper layers (see Figure 3), providing empirical evidence for progressive information compression in hierarchical VAEs.

## 5. IT Analysis for Data Generation

While the previous sections analyzed generalization capability, we now turn to the complementary problem of data generation, which evaluates how well a trained VAE approximates the true data distribution. Focusing on the standard VAE with a single LV, as in Section 3, we provide a theoretical guarantee for data generation under the realistic setting where the encoder and decoder are trained jointly.

After training, the model generates new data by sampling a LV $z$ from the prior $p(z)$ and transforming it through the decoder network $g_\theta$. The resulting data distribution is the pushforward measure $\hat{\mu} := g_\theta \# p(z)$. Our analysis bounds the 2-Wasserstein distance $W_2(\mathcal{D}, \hat{\mu})$ between the true data distribution $\mathcal{D}$ and the generated distribution $\hat{\mu}$.

See Appendix G for the formal definition. The following theorem presents our main result on the data generation quality of standard VAEs.

**Theorem 4.** *Consider the standard VAE model with a single LV, which is identical to the model analyzed in Section 3. Let $S = (X_1, \ldots, X_{n-1})$ be i.i.d. samples drawn from $\mathcal{D}$. Suppose that the decoder $g_\theta$ is measurable for any $\theta$, and that Assumptions 1 and 2 hold. Then, $\mathbb{E}_S \mathbb{E}_{q(\phi, \theta \mid S)} W_2^2(\mathcal{D}, \hat{\mu}) \leq$*

$$\frac{2\Delta}{\sqrt{n-1}} + \mathbb{E}_{S} \mathbb{E}_{q(W \mid S)} \Big[ \frac{2}{n-1} \sum_{i=1}^{n-1} \ell_0(W, X_i)$$
$$+ 3\Delta \sqrt{\tfrac{2}{n-1} \sum_{i=1}^{n-1} \mathrm{KL}(q(Z_i \mid \phi, X_i) \| p(Z_i))} \Big].$$

The complete proof is provided in Appendix G. Theorem 4 shows that the generation quality is controlled by two empirical quantities appearing in the VAE objective: the **empirical reconstruction loss** and the **empirical KL divergence**. Importantly, this is *not* a restatement of the ELBO: our result provides a finite-sample *population-level* guarantee on $W_2(\mathcal{D}, \hat{\mu})$ under joint training, while the ELBO is an empirical optimization objective and does not by itself imply a distributional bound on the generated measure.

While the connection between the empirical KL term and generation quality has been noted in prior work (Mbacke et al., 2023; Futami & Fujisawa, 2023), our result offers a crucial advancement. Unlike Mbacke et al. (2023), who assumed a fixed, data-independent decoder, and Futami & Fujisawa (2023), who analyzed discrete VQ-VAEs, our bound applies to standard continuous VAEs under a realistic joint training setting.

**Practical implication.** Theorem 4 is not intended to introduce a new training algorithm to be optimized directly. Rather, it provides a population-level justification for monitoring and balancing the two empirical quantities already present in the VAE objective: the reconstruction error and the KL regularization term. In particular, the bound suggests that improving only one of these quantities may be insufficient for controlling the Wasserstein discrepancy between the true and generated distributions, thereby supporting their joint use in model selection and regularization design.

## 6. Discussion and Related Work

As summarized in Table 1, several recent studies have investigated the performance of VAEs. However, to the best of our knowledge, this work is the first to derive generalization bounds that explicitly characterize the role of LVs for *standard Gaussian VAEs*. Similar to prior analyses of encoder models (Kawaguchi et al., 2023; Sefidgaran et al., 2023; Futami & Fujisawa, 2025), our framework is grounded in IT generalization analysis, highlighting a importance of LV

complexity and model overfitting.

The work most closely related to ours is that of Futami & Fujisawa (2025) on VQ-VAEs, which employs permutation-based symmetries over the entire dataset. While theoretically elegant, this approach has notable practical limitations: the resulting MI terms involve high-dimensional joint distributions and numerical evaluation computationally prohibitive. In contrast, our framework leverages the *local symmetry* inherent in the LOO-CMI formulation. This leads to CMI involving a one-dimensional index variable $U$, substantially reducing the dimensionality of the quantities to be estimated. Moreover, our framework extends seamlessly to hierarchical VAEs, generalizing prior discrete analyses to a broader class of continuous latent-variable models.

In view of existing IT analyses, our approach follows a growing line of work that incorporates classical information-theoretic tools to handle complex learning settings. Related studies include rate–distortion–based analyses (Hafez-Kolahi et al., 2021; Sefidgaran et al., 2022) and optimization analyses based on noisy-channel arguments (Pensia et al., 2018). In this sense, our work integrates classical techniques for continuous signals into generalization analysis and demonstrates their effectiveness in the context of VAEs.

## 7. Conclusion and Limitations

In this paper, we developed an information-theoretic analysis of generalization and data generation for standard VAEs with continuous LVs. Our results show that the generalization gap can be decomposed into a trade-off between encoder overfitting and the complexity of LVs, and the same framework extends naturally to hierarchical VAEs through layer-wise information terms. We also derived a population-level generation guarantee that connects the generated distribution to empirical reconstruction and KL quantities.

Our work has several limitations. First, our generalization analysis focuses on bounded squared reconstruction losses; extending the theory to the full ELBO or to unbounded log-loss objectives requires additional assumptions or a different proof framework. Second, our bounds are expectation bounds, and the exact mutual information terms are generally intractable for continuous VAEs. The numerical experiments, therefore, use computable proxies and upper bounds, which serve as diagnostics but may be loose. Finally, while the decoder-independent decomposition isolates the contribution of the encoder and latent variables to the generalization gap, it does not imply that the decoder is irrelevant to the absolute reconstruction or generation quality. Developing tighter, scalable estimators and extending the framework to broader generative objectives remain important directions for future work.

## Acknowledgment

We sincerely appreciate the anonymous reviewers for their insightful feedback. FF was supported by JSPS KAKENHI Grant Number JP23K16948. FF was supported by JST, PRESTO Grant Number JPMJPR22C8, Japan. FF and MF were supported by JSPS KAKENHI Grant Number 26K02974, Japan. MF was supported by JSPS KAKENHI Grant Number JP25K21286, Japan.

## Impact Statement

This paper presents work whose goal is to advance the field of Machine Learning. There are many potential societal consequences of our work, none which we feel must be specifically highlighted here.

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

# A. Notation used in the main paper

We summarize the notation we used in the main part of our paper.

| Category | Symbol | Meaning |
|---|---|---|
| Data and VAE model | $n \in \mathbb{N}$ | The sample size (n>1) |
| | $\mathcal{X}, \mathcal{Z}$ | A data and latent space |
| | $\mathcal{D}$ | An unknown data generating distribution |
| | $\Delta \in \mathbb{R}^+$ | A radius of a data space |
| | $X \in \mathcal{X} \subset \mathbb{R}^d$ | A data |
| | $S = \{X_i\}_{i=1}^{n-1} \in \mathcal{X}^{n-1}$ | A training dataset |
| | $\phi \in \Phi \subset \mathbb{R}^{d_\phi}$ | An encoder parameter |
| | $\theta \in \Theta \subset \mathbb{R}^{d_\theta}$ | A decoder parameter |
| | $W = \{\phi, \theta\}$ | A set of model parameters |
| | $g_\theta : \mathcal{Z} \to \mathcal{X}$ | A decoder network |
| | $q(z|\phi, x)$ | A posterior distribution over $\mathcal{Z}$ defined as $\mathcal{N}(\mu_\phi(x), \text{diag}(\sigma_\phi^2(x)))$ |
| | $p(z)$ | A prior distribution over $\mathcal{Z}$ defined as $\mathcal{N}(0, I_{d_z})$ |
| Algorithm and loss functions | $\mathcal{A} : \mathcal{X}^{n-1} \to \mathcal{W}$ | A randomized algorithm |
| | $q(\phi, \theta|S)$ | A randomized algorithm given $S$ |
| | $\ell : \mathcal{X} \times \mathcal{X} \to \mathbb{R}$ | a reconstruction loss function |
| | $\ell_0 : \mathcal{W} \times \mathcal{X} \to \mathbb{R}$ | An expected loss function over $w$ and $x$ |
| | $\text{gen}(\mu, \mathcal{D})$ | The expected generalization error based on a reconstruction loss |
| | $W_2(\mathcal{D}, \hat{\mu})$ | The 2-Wasserstein distance between $\mathcal{D}$ and $\hat{\mu}$ |
| Hierarchical VAE model | $L$ | The number of stochastic layers ($L > 1$) |
| | $\mathcal{Z}_l$ | The $l$-th latent space ($l \in [L]$) induced by stochastic layers ($\mathcal{Z}_0 = \mathcal{Z}_{L+1} = \mathcal{X}$) |
| | $q_l(z_l \mid \phi_l, z_{l-1})$ | A posterior over $\mathcal{Z}_l$ for $l = 1, \ldots, L+1$ defined as $\mathcal{N}(\mu_{\phi_l}(z_{l-1}), \text{diag}(\sigma_{\phi_l}^2(z_{l-1})))$ |
| | $\phi_{1:l} = (\phi_1, \ldots, \phi_l)$ | The parameters up to layer $l$ ($l = 1, \ldots, L+1$) |
| LOO-CMI setting | $U \sim \text{Uniform}([n])$ | Random index used for the IT analysis |
| | $X_U \in X^n$ | A test dataset in the LOO-CMI setting |
| | $S = X^n \backslash X_U$ | A training dataset in the LOO-CMI setting |

# B. Detailed description of VAE models

Here, we describe the modeling of the decoder that was omitted in Section 2.1. Since the VAE is a latent variable model, the likelihood function $p(x \mid \theta, z)$ and the decoder $g_\theta(z)$ are closely related. For example, in the original VAE paper (Kingma, 2013), a Bernoulli likelihood was proposed for grayscale images, whereas a Gaussian likelihood is typically used for color images. In both cases, $g_\theta(z)$ represents the mean of these likelihood distributions, and the data are generated deterministically from the latent variables. In the Bernoulli case, for instance, a pixel value greater than $0.5$ is deterministically assigned to $1$.

In contrast, in Section 4, the decoder is modeled stochastically. That is, under a Gaussian likelihood, both the mean and variance are computed from the latent variables to generate reconstructed data. As discussed in Appendix H, our theoretical analysis applies equally to both deterministic and stochastic decoders (and to Gaussian or Bernoulli likelihoods) as long as boundedness assumptions hold. From a practical perspective, it is worth noting that when the reconstructed data are generated stochastically, the resulting images tend to appear blurrier.

# C. Proof of Theorem 2

As discussed in Section 3.2, our proof proceeds through four steps: (i) truncation and discretization with explicit error estimates (Appendix C.1); (ii) evaluation of generalization for the discretized random variables (Appendix C.2); (iii) application of the data-processing inequality to remove the truncation/discretization effect from the information-theoretic quantity (Appendix C.3); and (iv) taking limits with respect to the truncation and discretization levels (Appendix C.4). We discuss the assumptions used in the proof in Appendix C.5. Hence, the proof of Theorem 2 spans Appendices C.1–C.4.

## C.1. Trimming and truncation of latent variable

For each $x$, let

$$Z \sim q(z|\phi, x) = \mathcal{N}(\mu_\phi(x), \text{diag}(\sigma_\phi^2(x))),$$

and throughout this subsection we condition on $\phi$ and $x$ when considering $q(z|\phi, x)$. From Assumption 2, there exist constants $M > 0$ and $S > 0$ such that

$$\|\mu_\phi(x)\|_2 \leq M, \qquad \|\sigma_\phi(x)\|_2 \leq S, \tag{3}$$

for all $x$ and $\phi$.

We truncate $Z$ as follows. Let $B(R) := \{z \in \mathbb{R}^{d_z} : \|z\|_2 \leq R\}$ denote the closed Euclidean ball with volume $|B(R)|$. Consider the following *randomized projection* of $Z$ into $B(R)$:

$$Z_R = \begin{cases} Z, & \|Z\|_2 \leq R, \\ U, & \|Z\|_2 > R, \end{cases} \tag{4}$$

where

$$U \sim \text{Unif}(B(R)) \quad \text{(independent of all other random variables)}.$$

This $Z_R$ is the truncation of the original $Z$. By construction in Eq. (4), $X - Z - Z_R$ forms a Markov chain.

Let $\hat{P}$ be the law of $Z_R$ and $P$ the law of $Z$. Denote the *outside mass*

$$p_{\text{bd}} := P(\|Z\| > R).$$

Since $\|Z\|_2^2 \geq 0$, by Markov's inequality,

$$P(\|Z\|_2 > R) = P(\|Z\|_2^2 > R^2) \leq \frac{\mathbb{E}\|Z\|_2^2}{R^2}.$$

From Eq. (3), we have

$$p_{\text{bd}} = P(\|Z\|_2 > R) \leq \frac{\|\mu_\phi(x)\|_2^2 + \|\sigma_\phi(x)\|_2^2}{R^2} \leq \frac{M^2 + S^2}{R^2}.$$

Write the *inside* absolutely continuous part of $\hat{P}$ as

$$P_{\text{in}}(\mathrm{d}z) := \mathbf{1}_{\{\|z\| < R\}} f_Z(z) \mathrm{d}z, \qquad \hat{P} = P_{\text{in}} + p_{\text{bd}} \cdot U_{B(R)}.$$

Here $f_Z(z)$ denotes the Gaussian density. Note that $P_{\text{in}}$ discards the probability mass outside $B(R)$.

Next, we consider a discretization. First, introduce a fixed codebook

$$\{c_1, \ldots, c_K\} \subset B(R),$$

independent of the dataset. Assume $\{c_i\}_{i=1}^K \subset B(R)$ forms a $\tau$-cover of $B(R)$, and let $\{C_i\}_{i=1}^K$ be the induced Voronoi partition of $B(R)$ (so $\text{diam}(C_i) \leq 2\tau$ and $\bigcup_i C_i = B(R)$ up to null sets). Formally, given a finite codebook $\mathcal{C} = \{c_1, \ldots, c_K\}$, the Voronoi cell associated with $c_i$ is

$$C_i := B(R) \cap \left\{ z \in \mathbb{R}^{d_z} : \|z - c_i\|_2 \leq \|z - c_j\|_2, \quad \forall j \in \{1, \ldots, K\} \right\}. \tag{5}$$

Since the codebook is a $\tau$-cover of $B(R)$, every $z \in C_i$ satisfies $\|z - c_i\|_2 \leq \tau$, so $\text{diam}(C_i) \leq 2\tau$.

Define the cell masses

$$w_i := \hat{P}(C_i), \qquad v_i := P_{\text{in}}(C_i).$$

Define the *cell-uniform* and *inside-matched cell-uniform* distributions by

$$Q_{\text{cell}}(\mathrm{d}z) := \sum_{i=1}^K \frac{w_i}{|C_i|} \mathbf{1}_{\{z \in C_i\}} \mathrm{d}z, \qquad Q_{\text{cell}}^{\text{in}}(\mathrm{d}z) := \sum_{i=1}^K \frac{v_i}{|C_i|} \mathbf{1}_{\{z \in C_i\}} \mathrm{d}z.$$

Let $\hat{Z}$ denote a random variable with law $Q_{\text{cell}}(\mathrm{d}z)$. One can generate $\hat{Z}$ by: (i) sampling $Z_R$ via Eq. (4); (ii) determining the cell containing the sample using Eq. (5); and (iii) drawing a uniform point within that cell. Therefore, the Markov chain $X - Z - Z_R - \hat{Z}$ holds.

We now show that the loss difference under $Z$ and $\hat{Z}$ vanishes as $R \to \infty$. For notational convenience, define $L : \mathcal{Z} \to \mathbb{R}$ by $L(z) = \ell(x, g_\theta(z))$, conditioning on $x$, $\theta$, and $\phi$. We thus focus on the discretization/quantization error in $z$.

By decomposing the loss,

$$\left| \mathbb{E}_{Z \sim P} L(Z) - \mathbb{E}_{\hat{Z} \sim Q_{\text{cell}}} L(\hat{Z}) \right| \le \left| \mathbb{E}_{Z \sim P} L(Z) - \mathbb{E}_{Z_R \sim \hat{P}} L(Z_R) \right| + \left| \mathbb{E}_{Z_R \sim \hat{P}} L(Z_R) - \mathbb{E}_{\hat{Z} \sim Q_{\text{cell}}} L(\hat{Z}) \right|. \tag{6}$$

We bound each term using total variation (TV). For finite signed measures $\mu, \nu$ on $\mathcal{Z}$,

$$\mathrm{TV}(\mu, \nu) := \tfrac{1}{2} \int_{\mathcal{Z}} \left| \mathrm{d}\mu - \mathrm{d}\nu \right|.$$

Then

$$\left| \mathbb{E}_{Z \sim P} L(Z) - \mathbb{E}_{Z_R \sim \hat{P}} L(Z_R) \right| \le \Delta \mathrm{TV}(P, \hat{P}).$$

This follows from the dual representation of TV: for any $h$ with $\|h\|_\infty \le 1$, $|\int h dP - \int h dQ| \le \mathrm{TV}(P, Q) = \frac{1}{2}\|p - q\|_1$. Take $h = L/\Delta \in [0, 1]$ and $P, Q$ with densities $q_C, q_R$.

We now state the TV bounds.

**Lemma 1** (TV between $Z$ and $Z_R$). *With the above definition of $Z_R$, we have*

$$\mathrm{TV}(P, \hat{P}) = p_{\text{bd}}. \tag{7}$$

*Proof.* Decompose $P = P_{\text{in}} + P_{\text{out}}$ with $P_{\text{out}}$ supported on $\{\|z\| > R\}$ and $P_{\text{out}}(B(R)^c) = p_{\text{bd}}$. Also $\hat{P} = P_{\text{in}} + p_{\text{bd}} U_{B(R)}$, which is absolutely continuous w.r.t. Lebesgue on $B(R)$. Then

$$2\mathrm{TV}(P, \hat{P}) = \int_{B(R)} \left| f_Z(z) - \left( f_Z(z) + \tfrac{p_{\text{bd}}}{|B(R)|} \right) \right| \mathrm{d}z + \int_{\|z\| > R} f_Z(z) \mathrm{d}z$$
$$= \tfrac{p_{\text{bd}}}{|B(R)|} |B(R)| + p_{\text{bd}} = 2 p_{\text{bd}},$$

hence $\mathrm{TV}(P, \hat{P}) = p_{\text{bd}}$. $\qquad\square$

From Assumption 2, there exist constants $s, S > 0$ such that $s < \sigma_{\phi,j}(x) \le S$ for all $x$, $\phi$, and $j \in [d_z]$. We set

$$L_f := \frac{\sqrt{d_z}}{s}.$$

The following lemma is the only regularity estimate for the Gaussian density that is needed in the subsequent truncation and discretization argument.

**Lemma 2** (Cell-uniform approximation via an $L^1$-gradient bound). *Assume that $\{C_i\}_{i=1}^K$ is the Voronoi partition of $B(R)$ induced by a $\tau$-cover, so that $C_i \subset B(R)$ and $\mathrm{diam}(C_i) \le 2\tau$. Then*

$$\mathrm{TV}(\hat{P}, Q_{\text{cell}}) = \mathrm{TV}(P_{\text{in}}, Q_{\text{cell}}^{\text{in}}) \le \tau \int_{B(R)} \|\nabla f_Z(z)\|_2 \, dz \le L_f \tau. \tag{8}$$

*Proof.* We first note that the boundary mass cancels exactly. Indeed, since

$$w_i = v_i + p_{\text{bd}} \frac{|C_i|}{|B(R)|},$$

we have

$$Q_{\text{cell}}(dz) = \sum_{i=1}^K \left( \frac{v_i}{|C_i|} + \frac{p_{\text{bd}}}{|B(R)|} \right) \mathbb{1}_{\{z \in C_i\}} \, dz$$
$$= Q_{\text{cell}}^{\text{in}}(dz) + p_{\text{bd}} U_{B(R)}(dz),$$

whereas

$$\hat{P}(dz) = P_{\text{in}}(dz) + p_{\text{bd}} U_{B(R)}(dz).$$

Thus

$$\hat{P} - Q_{\text{cell}} = P_{\text{in}} - Q_{\text{cell}}^{\text{in}},$$

and hence

$$\text{TV}(\hat{P}, Q_{\text{cell}}) = \text{TV}(P_{\text{in}}, Q_{\text{cell}}^{\text{in}}).$$

For each cell, define

$$m_i := \frac{1}{|C_i|} \int_{C_i} f_Z(z) \, dz.$$

Since $C_i$ is convex and $\text{diam}(C_i) \leq 2\tau$, the $L^1$ Poincaré inequality gives

$$\int_{C_i} |f_Z(z) - m_i| \, dz \leq \text{diam}(C_i) \int_{C_i} \|\nabla f_Z(z)\|_2 \, dz$$
$$\leq 2\tau \int_{C_i} \|\nabla f_Z(z)\|_2 \, dz.$$

Therefore,

$$\text{TV}(P_{\text{in}}, Q_{\text{cell}}^{\text{in}}) = \frac{1}{2} \sum_{i=1}^{K} \int_{C_i} |f_Z(z) - m_i| \, dz$$
$$\leq \frac{1}{2} \sum_{i=1}^{K} 2\tau \int_{C_i} \|\nabla f_Z(z)\|_2 \, dz$$
$$= \tau \int_{B(R)} \|\nabla f_Z(z)\|_2 \, dz.$$

The final inequality in (8) follows from Lemma 3. $\qquad\square$

**Lemma 3** (Gaussian $L^1$-gradient bound). *Let $Z \sim \mathcal{N}(\mu_\phi(x), \text{diag}(\sigma_\phi^2(x)))$, and write $\Sigma = \text{diag}(\sigma_\phi^2(x))$. Then*

$$\int_{B(R)} \|\nabla f_Z(z)\|_2 \, dz \leq \int_{\mathbb{R}^{d_z}} \|\nabla f_Z(z)\|_2 \, dz$$
$$\leq \left( \sum_{j=1}^{d_z} \frac{1}{\sigma_{\phi,j}^2(x)} \right)^{1/2} \leq \frac{\sqrt{d_z}}{s}.$$

*Proof.* Since

$$\nabla f_Z(z) = f_Z(z) \Sigma^{-1}(\mu - z),$$

Jensen's inequality gives

$$\int_{B(R)} \|\nabla f_Z(z)\|_2 \, dz \leq \int_{\mathbb{R}^{d_z}} f_Z(z) \|\Sigma^{-1}(\mu - z)\|_2 \, dz$$
$$\leq \left( \int_{\mathbb{R}^{d_z}} f_Z(z) \|\Sigma^{-1}(\mu - z)\|_2^2 \, dz \right)^{1/2}$$
$$= \left( \sum_{j=1}^{d_z} \frac{1}{\sigma_{\phi,j}^4(x)} \mathbb{E}\big[(Z_j - \mu_j)^2\big] \right)^{1/2}$$
$$= \left( \sum_{j=1}^{d_z} \frac{1}{\sigma_{\phi,j}^2(x)} \right)^{1/2} \leq \frac{\sqrt{d_z}}{s}.$$

□

Combining Lemma 2 with the truncation bound $\mathrm{TV}(P, \hat{P}) = p_{\mathrm{bd}} \leq (M^2 + S^2)/R^2$ and from Eqs. (6) and (7), we obtain

$$\left| \mathbb{E}_{Z \sim P} L(Z) - \mathbb{E}_{\hat{Z} \sim Q_{\mathrm{cell}}} L(\hat{Z}) \right| \leq \Delta \frac{M^2 + S^2}{R^2} + \Delta L_f \tau.$$

Recall that, for any $\tau > 0$, we can set $K = \lceil (1 + 2R/\tau)^{d_z} \rceil$ to construct a $\tau$-cover. Choosing $\tau = 3/R$, we obtain

$$\left| \mathbb{E}_{Z \sim P} L(Z) - \mathbb{E}_{\hat{Z} \sim Q_{\mathrm{cell}}} L(\hat{Z}) \right| \leq \Delta \frac{M^2 + S^2}{R^2} + \frac{3\Delta \sqrt{d_z}}{sR}. \tag{9}$$

Thus the truncation and discretization error terms vanish as $R \to \infty$.

## C.2. Donsker–Varadhan lemma for discretized LVs

We analyze the generalization error for the discretized LVs. Recall that $L(z) = \ell(x, g_\theta(z))$. Given training data $S = (X_1, \ldots, X_{n-1})$, define $L_S(z) = \frac{1}{n-1} \sum_{i=1}^{n-1} \ell(X_i, g_\theta(z))$, and observe

$$\mathrm{gen}(n, \mathcal{D}) = \mathbb{E}_{X,Z} L(Z) - \mathbb{E}_{X,\hat{Z}} L(\hat{Z}) + \mathbb{E}_{X,\hat{Z}} L(\hat{Z}) - \mathbb{E}_{S,\hat{Z}} L_S(\hat{Z}) + \mathbb{E}_{S,\hat{Z}} L_S(\hat{Z}) - \mathbb{E}_{S,Z} L_S(Z)$$

$$\leq \mathbb{E}_{X,\hat{Z}} L(\hat{Z}) - \mathbb{E}_{S,\hat{Z}} L_S(\hat{Z}) + 2\Delta \frac{M^2 + S^2}{R^2} + \frac{6\Delta \sqrt{d_z}}{sR}.$$

where we used Eq. (9) for both the test and training loss.

Define the discretized absolute gap by

$$\mathrm{g\hat{e}n}(\hat{Z}) := \left| \mathbb{E}_{X,\hat{Z}} L(\hat{Z}) - \mathbb{E}_{S,\hat{Z}} L_S(\hat{Z}) \right|.$$

By the triangle inequality and the truncation/discretization estimate,

$$\mathrm{gen}(n, \mathcal{D}) \leq \mathrm{g\hat{e}n}(\hat{Z}) + 2\Delta \frac{M^2 + S^2}{R^2} + 2\Delta L_f \tau.$$

Equivalently, before fixing a particular $\tau(R)$,

$$\mathrm{gen}(n, \mathcal{D}) \leq \mathbb{E}_{X,\hat{Z}} L(\hat{Z}) - \mathbb{E}_{S,\hat{Z}} L_S(\hat{Z}) + 2\Delta \frac{M^2 + S^2}{R^2} + 2\Delta L_f \tau.$$

With $\tau = 3/R$ and $L_f = \sqrt{d_z}/s$, the remainder becomes

$$2\Delta \frac{M^2 + S^2}{R^2} + \frac{6\Delta \sqrt{d_z}}{sR}.$$

where we consider the expected generalization error. So expectation with respect to $W$ is implicitly taken in the above equation.

We then define

$$\mathrm{g\hat{e}n}(\hat{Z})$$
$$:= |\mathbb{E}_{X,\hat{Z}} L(\hat{Z}) - \mathbb{E}_{S,\hat{Z}} L_S(\hat{Z})|$$
$$= \left| \mathbb{E}_{U,X^n} \mathbb{E}_{q(\phi,\theta|X_U^{n-1})} \left( \mathbb{E}_{q(\hat{Z}_U|\phi,X_U)} \ell(X, g_\theta(\hat{Z}_U)) - \frac{1}{n-1} \sum_{m=1}^{n-1} \mathbb{E}_{q(\hat{Z}_{U,m}^{n-1}|\phi,X_{U,m}^{n-1})} \ell(X_{U,m}^{n-1}, g_\theta(\hat{Z}_{U,m}^{n-1})) \right) \right|. \tag{10}$$

Note that $\hat{Z}$ follows the mixture of uniform distributions

$$Q_{\mathrm{cell}}(\mathrm{d}z) := \sum_{i=1}^{K} \frac{w_i}{|C_i|} \mathbf{1}_{\{z \in C_i\}} \mathrm{d}z.$$

Sampling from this mixture is done by first drawing the index $J$ with probability $w_{i=J}$ and then drawing $\hat{Z}$ from $q(\hat{Z}|J)$ (note that the mixture is $\hat{Z} \sim \mathbb{E}_J q(\hat{Z}|J)$.) and this leads to a Markov chain

$$U \to Z \to Z_R \to J \to \hat{Z}. \tag{11}$$

Given $J$, since $q(\hat{Z} \mid J)$ is uniform on $C_J$,

$$\mathbb{E}_{q(\hat{Z}|J)} \ell(X, g_\theta(\hat{Z}_U)) = \frac{1}{|C_J|} \int_{C_J} \ell(X, g_\theta(z)) \, dz.$$

We therefore set

$$\ell_C(X, \theta, J) := \frac{1}{|C_J|} \int_{C_J} \ell(X, g_\theta(z)) \, dz.$$

Then $0 \leq \ell_C(X, \theta, J) \leq \Delta$.

Next, we introduce the LOO-CMI setting: $X_U$ serves as the test datum and $X_U^{n-1}$ as the training dataset. To emphasize the dependence on $U$, we write

$$q(J^n|\phi, X^n) = q(J_U, J_U^{n-1}|\phi, X^n) = q(J_U, J_U^{n-1}|\phi, X_U, X_U^{n-1}) = q(J_U|\phi, X_U)q(J_U^{n-1}|\phi, X_U^{n-1}).$$

We bound the generalization error using an information-theoretic analysis under the LOO-CV setting:

**Theorem 5.** *Under Assumptions 1 and 2, and the leave-one-out setting, we have*

$$\hat{\text{gen}}(\hat{Z}) \leq \frac{3n}{\sqrt{2}(n-1)} \Delta \sqrt{\mathbb{E}_{x^n \sim P^n}(I(\phi; U|x^n) + I(J^n; U|\phi, x^n))} + \frac{\Delta}{\sqrt{n-1}}.$$

*where $\hat{Z} = Z^n = (Z_U, Z_U^{n-1})$.*

Here

$$J^n = (J_U, J_U^{n-1}), \qquad \hat{Z}^n = (\hat{Z}_U, \hat{Z}_U^{n-1}),$$

and $\hat{Z}^n$ is generated from $J^n$ by drawing each coordinate uniformly inside the corresponding cell. The proof is given in Appendix D.

### C.3. Data-processing inequality

Conditionally on $(\phi, x^n)$, the vector-valued Markov chain

$$U \longrightarrow Z^n \longrightarrow Z_R^n \longrightarrow J^n \longrightarrow \hat{Z}^n$$

holds. Hence, by data processing,

$$I(J^n; U \mid \phi, x^n) \leq I(Z_R^n; U \mid \phi, x^n) \leq I(Z^n; U \mid \phi, x^n).$$

Thus, combining the bounds, we conclude

$$\text{gen}(n, \mathcal{D}) \leq \frac{3n}{\sqrt{2}(n-1)} \Delta \sqrt{\mathbb{E}_{x^n \sim P^n}(I(\phi; U|x^n) + I(Z^n; U|\phi, x^n))}$$
$$+ \frac{\Delta}{\sqrt{n-1}} + 2\Delta \frac{M^2 + S^2}{R^2} + \frac{6\Delta\sqrt{d_z}}{sR}.$$

### C.4. Taking the limit of trimming and discretization

Thus, truncation and discretization do not affect the mutual information term that measures overfitting. From Eq. (9), we take $R \to \infty$ and choose $\tau = \tau(R) \to 0$, for example $\tau(R) = 3/R$. Then

$$\Delta \frac{M^2 + S^2}{R^2} \to 0, \qquad \Delta L_f \tau(R) \to 0.$$

Therefore, the truncation and discretization terms vanish, while the information-theoretic term is unchanged by the limiting operation. This completes the proof.

## C.5. Discussion about the assumption

Apart from the boundedness of the loss, the main technical assumption in our analysis is Assumption 2. Among its conditions, the boundedness of the encoder outputs, $\|\mu_\phi(x)\|_2 < \infty$ and $\sigma_{\phi,i}(x) < \infty$, is not restrictive in practice. On the other hand, the condition $0 < \sigma_{\phi,i}(x)$ requires the variance of the approximate posterior to be positive definite. In many practical implementations of VAEs, the encoder network outputs the logarithm of the variance, $\log \sigma^2(x)$, and the variance is obtained by exponentiation. Therefore, ensuring that the encoder outputs remain bounded is sufficient to satisfy this assumption.

The positivity of the variance lower bound is used in our proof to ensure that the Lipschitz constant of the Gaussian distribution is bounded, which is necessary to derive Eq. (8). This bounded Lipschitz constant allows us to evaluate the effect of truncation using the TV distance. The use of TV distance is crucial, since it enables our analysis—unlike Mbacke et al. (2023)—to avoid imposing Lipschitz constraints on either the encoder or the decoder. Neural networks often exhibit large Lipschitz constants without specific regularization, so our theory remains applicable even in such unregularized settings.

Nevertheless, there may be cases where the condition $0 < \sigma_{\phi,i}(x)$ does not hold, depending on the model architecture. This corresponds to the phenomenon known as posterior collapse. In such cases, the discretization error can instead be controlled by assuming the Lipschitz continuity of the decoder network. Under this alternative setting, the required assumption becomes:

**Assumption 4.** *For a fixed training dataset $S = s$ and for any input $x \in \mathcal{X}$, we have $\mathbb{E}_{q(\phi|s)}\|\mu_\phi(x)\|_2^2 < \infty$ and $\mathbb{E}_{q(\phi|s)}\|\sigma_\phi(x)\|^2 < \infty$.*

# D. Proof of Theorem 5

### D.1. Auxiliary Lemma

In the following proofs, we repeatedly use McDiarmid's inequality. A function $f : \mathcal{X}^n \to \mathbb{R}$ is said to satisfy the *bounded differences property* if there exist nonnegative constants $c_1, \ldots, c_n$ such that, for all $i$,

$$\sup_{x_1,\ldots,x_n,x_i' \in \mathcal{X}} |f(x_1,\ldots,x_n) - f(x_1,\ldots,x_{i-1},x_i',x_{i+1},\ldots,x_n)| \le c_i, \quad 1 \le i \le n.$$

Assuming that $X_1, \ldots, X_n$ are independent random variables taking values in $\mathcal{X}$, we have the following result.

**Lemma 4** (McDiarmid's inequality)**.** *Let $f$ be a function satisfying the bounded differences property. Then, for any $t \in \mathbb{R}$, the following inequality holds:*

$$\mathbb{E}\left[e^{t(f(X_1,\ldots,X_n)-\mathbb{E}[f(X_1,\ldots,X_n)])}\right] \le e^{\frac{t^2}{8}\sum_{i=1}^{n} c_i^2}.$$

### D.2. Derivation of the DV inequality

Under the leave-one-out setting, the generalization error defined in Eq. (10) can be expressed as

$$\mathbb{E}_{U,X^n}\mathbb{E}_{q(\phi,\theta|X_U^{n-1})}\mathbb{E}_{q(J_U|\phi,X_U)}\frac{1}{|C_{J_U}|}\int_{C_{J_U}}\|X_U - g_\theta(z)\|^2 dz$$

$$- \mathbb{E}_{U,X^n}\mathbb{E}_{q(\phi,\theta|X_U^{n-1})}\frac{1}{n-1}\sum_{m=1}^{n-1}\mathbb{E}_{q(J_{U,m}^{n-1}|\phi,X_{U,m}^{n-1})}\frac{1}{|C_{J_{U,m}^{n-1}}|}\int_{C_{J_{U,m}^{n-1}}}\|X_{U,m}^{n-1} - g_\theta(z)\|^2 dz$$

$$= \mathbb{E}_{X^n,U}\sum_{k=1}^{K}\mathbb{E}_{q(J_U|\phi,X_U)q(\phi,\theta|X_U^{n-1})}\ell_C(X_U,\theta,k)\mathbb{1}_{k=J_U}$$

$$- \mathbb{E}_{X^n,U}\sum_{k=1}^{K}\frac{1}{n-1}\sum_{m=1}^{n-1}\mathbb{E}_{q(J_{U,m}^{n-1}|\phi,X_{U,m}^{n-1})q(\phi,\theta|X_U^{n-1})}\ell_C(X_{U,m}^{n-1},\theta,k)\mathbb{1}_{k=J_{U,m}^{n-1}}$$

where the first term corresponds to the test loss and the second term corresponds to the training loss.

We then decompose the loss as follows:

$$\text{gen}(n, \mathcal{D}) = \mathbb{E}_{X^n,U} \sum_{k=1}^{K} \mathbb{E}_{q(J_U|\phi,X_U)q(\phi,\theta|X_U^{n-1})} \ell_C(X_U,\theta,k) \mathbb{1}_{k=J_U} \tag{12}$$

$$- \mathbb{E}_{X^n,U} \sum_{k=1}^{K} \frac{1}{n-1} \sum_{m=1}^{n-1} \mathbb{E}_{q(J_{U,m}^{n-1}|\phi,X_{U,m}^{n-1})q(\phi,\theta|X_U^{n-1})} \ell_C(X_U,\theta,k) \mathbb{1}_{k=J_{U,m}^{n-1}}$$

$$+ \mathbb{E}_{X^n,U} \sum_{k=1}^{K} \frac{1}{n-1} \sum_{m=1}^{n-1} \mathbb{E}_{q(J_{U,m}^{n-1}|\phi,X_{U,m}^{n-1})q(\phi,\theta|X_U^{n-1})} \ell_C(X_U,\theta,k) \mathbb{1}_{k=J_{U,m}^{n-1}}$$

$$- \mathbb{E}_{X^n,U} \sum_{k=1}^{K} \frac{1}{n-1} \sum_{m=1}^{n-1} \mathbb{E}_{q(J_{U,m}^{n-1}|\phi,X_{U,m}^{n-1})q(\phi,\theta|X_U^{n-1})} \ell_C(X_{U,m}^{n-1},\theta,k) \mathbb{1}_{k=J_{U,m}^{n-1}}.$$

First, we upper bound the first two terms by applying the Donsker–Varadhan inequality. Consider the joint and prior distributions:

$$\mathbf{Q} := P(U)q(\phi,\theta|X_U^{n-1})q(J^n|\phi,X_U,X_U^{n-1}), \tag{13}$$
$$\mathbf{P} := P(U)q(\phi,\theta|X_U^{n-1}) \mathop{\mathbb{E}}_{P(U')} q(J^n|\phi,X_{U'},X_{U'}^{n-1}).$$

Applying the DV inequality yields

$$\mathbb{E}_{X^n,U} \sum_{k=1}^{K} \mathbb{E}_{q(\phi,\theta|X_U^{n-1})} \ell_C(X_U,\theta,k) \left( \mathbb{E}_{q(J_U|\phi,X_U)} \mathbb{1}_{k=J_U} - \frac{1}{n-1} \sum_{m=1}^{n-1} \mathbb{E}_{q(J_{U,m}^{n-1}|X_{U,m}^{n-1})} \mathbb{1}_{k=J_{U,m}^{n-1}} \right)$$

$$\leq \mathbb{E}_X \frac{1}{\lambda} \text{KL}(\mathbf{Q}\|\mathbf{P}) + \mathbb{E}_X \frac{1}{\lambda} \log \mathbb{E}_{\mathbf{P}} \exp \left( \lambda \sum_{k=1}^{K} \ell_C(X_U,\theta,k) \left( \mathbb{1}_{k=J_U} - \frac{1}{n-1} \sum_{m=1}^{n-1} \mathbb{1}_{k=J_{U,m}^{n-1}} \right) \right). \tag{14}$$

Note that $\mathop{\mathbb{E}}_{P(U')} q(J^n|\phi,X_{U'},X_{U'}^{n-1})$ is symmetric with respect to $U$. Thus,

$$= \log \mathbb{E}_{P(U)q(\phi,\theta|X_U^{n-1}) \mathop{\mathbb{E}}_{P(U')} q(J^n|\phi,X_{U'},X_{U'}^{n-1})P(U'')} \exp \left( \lambda \sum_{k=1}^{K} \ell_C(X_{U''},\theta,k) \left( \mathbb{1}_{k=J_{U''}} - \frac{1}{n-1} \sum_{m=1}^{n-1} \mathbb{1}_{k=J_{U'',m}^{n-1}} \right) \right)$$

$$= \log \mathbb{E}_{P(U)q(\phi,\theta|X_U^{n-1}) \mathop{\mathbb{E}}_{P(U')} q(J^n|\phi,X_{U'},X_{U'}^{n-1})} \mathbb{E}_{P(U'')} \exp \left( \lambda \sum_{k=1}^{K} \ell_C(X_{U''},\theta,k) \left( \frac{n}{n-1} \mathbb{1}_{k=J_{U''}} - \frac{1}{n-1} \sum_{m=1}^{n} \mathbb{1}_{k=J_m^n} \right) \right)$$

where $J^n = (J_1^n, \ldots, J_n^n)$, which does not depend on $U''$. We apply McDiarmid's inequality to this exponent with respect to $U''$. To invoke McDiarmid's inequality in Lemma 4, we evaluate the bounded difference coefficient under replacing one element of $U''$ with $\tilde{U}''$. The maximum change induced by this replacement is bounded by $\lambda\Delta\frac{n}{n-1}$ since we assumed $|C_i| \leq 1$, it holds that $0 < \ell_C(X,\theta,J) \leq \Delta$ for all $J$. Therefore, its log exponential moment is bounded by $(\lambda\Delta\frac{n}{n-1})^2/8 = \lambda^2\Delta^2/8 \times n^2/(n-1)^2$.

Next, we focus on the third and fourth terms in Eq. (12):

$$
\mathbb{E}_{X^n,U} \sum_{k=1}^{K} \frac{1}{n-1} \sum_{m=1}^{n-1} \mathbb{E}_{q(J_{U,m}^{n-1}|\phi,X_{U,m}^{n-1})q(\phi,\theta|X_U^{n-1})} \frac{1}{|C_k|} \int_{C_k} \|X_U - g_\theta(z)\|^2 dz \mathbb{1}_{k=J_{U,m}^{n-1}}
$$

$$
- \mathbb{E}_{X^n,U} \sum_{k=1}^{K} \frac{1}{n-1} \sum_{m=1}^{n-1} \mathbb{E}_{q(J_{U,m}^{n-1}|\phi,X_{U,m}^{n-1})q(\phi,\theta|X_U^{n-1})} \frac{1}{|C_k|} \int_{C_k} \|X_{U,m}^{n-1} - g_\theta(z)\|^2 dz \mathbb{1}_{k=J_{U,m}^{n-1}}
$$

$$
= \mathbb{E}_{X^n,U} \frac{2}{n-1} \sum_{m=1}^{n-1} \left( X_U - X_{U,m}^{n-1} \right) \cdot \mathbb{E}_{q(J_{U,m}^{n-1}|\phi,X_{U,m}^{n-1})q(\phi,\theta|X_U^{n-1})} \sum_{k=1}^{K} \frac{1}{|C_k|} \int_{C_k} g_\theta(z)dz \mathbb{1}_{k=J_{U,m}^{n-1}}
$$

$$
\leq \mathbb{E}_X \frac{1}{\lambda} \mathrm{KL}(\mathbf{Q}|\mathbf{P}) + \mathbb{E}_X \frac{1}{\lambda} \log \mathbb{E}_\mathbf{P} \exp \left( \frac{2\lambda}{n-1} \sum_{m=1}^{n-1} \left( X_U - X_{U,m}^{n-1} \right) \cdot \sum_{k=1}^{K} \frac{1}{|C_k|} \int_{C_k} g_\theta(z)dz \mathbb{1}_{k=J_{U,m}^{n-1}} \right)
$$

$$
\leq \mathbb{E}_X \frac{1}{\lambda} \mathrm{KL}(\mathbf{Q}|\mathbf{P}) + \mathbb{E}_X \frac{1}{\lambda} \log \mathbb{E}_{P(U)q(\phi,\theta|X_U^{n-1})} \underset{P(U')}{\mathbb{E}} q(J^n|\phi,X_{U'},X_{U'}^{n-1}) \times
$$

$$
\mathbb{E}_{p(U'')} \exp \left( \frac{2\lambda}{n-1} \sum_{m=1}^{n-1} \left( X_U - X_{U,m}^{n-1} \right) \cdot \sum_{k=1}^{K} \frac{1}{|C_k|} \int_{C_k} g_\theta(z)dz \mathbb{1}_{k=J_{U'',m}^{n-1}} \right). \tag{15}
$$

For every cell $C_k$, define the averaged decoder output

$$
\bar{g}_{\theta,k} := \frac{1}{|C_k|} \int_{C_k} g_\theta(z)\, dz.
$$

All subsequent expressions involving an integral of $g_\theta$ over $C_k$ are written in terms of $\bar{g}_{\theta,k}$.

We first evaluate the expectation of the exponential moment:

$$
\Omega := \mathbb{E}_{P(U)q(\phi,\theta|X_U^{n-1})} \frac{2}{n-1} \sum_{m=1}^{n-1} \left( X_U - X_{U,m}^{n-1} \right) \cdot \mathbb{E}_{\underset{P(U')}{\mathbb{E}} q(J^n|\phi,X_{U'},X_{U'}^{n-1})} \sum_{k=1}^{K} \bar{g}_{\theta,k} \mathbb{1}_{k=J_{U,m}^{n-1}}. \tag{16}
$$

Focusing on $\underset{P(U')}{\mathbb{E}} q(J^n|\phi, X_{U'}, X_{U'}^{n-1})$, then the LOO symmetry implies $\underset{P(U')}{\mathbb{E}} q(J^n|\phi,X_{U'},X_{U'}^{n-1}) \mathbb{1}_{k=J_{U,m}^{n-1}}$ is identical for all $m$. Denote this value by $P_k$. Then Eq. (16) becomes

$$
\mathbb{E}_{P(X^n)P(U)q(\phi,\theta|X_U^{n-1})} \left( X_U - \frac{1}{n-1} \sum_{m=1}^{n-1} X_{U,m}^{n-1} \right) \cdot \sum_{k=1}^{K} \bar{g}_{\theta,k} P_k
$$

$$
= \mathbb{E}_{P(U)} \mathbb{E}_{X_U} \mathbb{E}_{X_U^{n-1}} \left( X_U - \frac{1}{n-1} \sum_{m=1}^{n-1} X_{U,m}^{n-1} \right) \cdot \mathbb{E}_{q(\phi,\theta|X_U^{n-1})} \sum_{k=1}^{K} \bar{g}_{\theta,k} P_k
$$

$$
= \mathbb{E}_{P(U)} \mathbb{E}_{X_U^{n-1}} \left( \mathbb{E}_{X_U} X_U - \frac{1}{n-1} \sum_{m=1}^{n-1} X_{U,m}^{n-1} \right) \cdot \mathbb{E}_{q(\phi,\theta|X_U^{n-1})} \sum_{k=1}^{K} \bar{g}_{\theta,k} P_k
$$

$$
= \mathbb{E}_{P(U)} \mathbb{E}_{X_U^{n-1}} \left( \mathbb{E}_X X - \frac{1}{n-1} \sum_{m=1}^{n-1} X_{U,m}^{n-1} \right) \cdot \mathbb{E}_{q(\phi,\theta|X_U^{n-1})} \sum_{k=1}^{K} \bar{g}_{\theta,k} P_k
$$

$$
\leq \sqrt{ \mathbb{E}_{P(U)} \mathbb{E}_{X_U^{n-1} q(\phi,\theta|X_U^{n-1})} \left\| \frac{1}{n-1} \sum_{m=1}^{n-1} X_{U,m}^{n-1} - \mathbb{E}_X X \right\|^2 \mathbb{E}_{P(U)} \mathbb{E}_{X_U^{n-1} q(\phi,\theta|X_U^{n-1})} \left\| \sum_{k=1}^{K} \mathbb{E}_{q(\phi,\theta|X_U^{n-1})} \bar{g}_{\theta,k} P_k \right\|^2 }
$$

$$
\leq \sqrt{ \mathbb{E}_{P(U)} \mathbb{E}_{X_U^{n-1}} \left\| \frac{1}{n-1} \sum_{m=1}^{n-1} X_{U,m}^{n-1} - \mathbb{E}_X X \right\|^2 } \sqrt{\Delta}.
$$

We bound the first factor exactly as in Eq. (26), i.e., by the variance of a bounded random variable, yielding

$$\sqrt{\mathbb{E}_{P(U)}\mathbb{E}_{X_U^{n-1}}\left\|\frac{1}{n-1}\sum_{m=1}^{n-1}X_{U,m}^{n-1}-\mathbb{E}_X X\right\|^2}\leq\sqrt{\frac{\Delta}{4(n-1)}}.$$

Thus,

$$\Omega\leq\frac{\Delta}{\sqrt{n-1}},$$

which we substitute back into the exponential moment in Eq. (15). We evaluate

$$\mathbb{E}_X\frac{1}{\lambda}\mathrm{KL}\big(\mathbf{Q}|\mathbf{P}\big)+\mathbb{E}_X\frac{1}{\lambda}\log\mathbb{E}_{\mathbf{P}}\exp\left(\frac{2\lambda}{n-1}\sum_{m=1}^{n-1}\left(X_U-X_{U,m}^{n-1}\right)\cdot\sum_{k=1}^{K}\bar{g}_{\theta,k}\mathbb{1}_{k=J_{U,m}^{n-1}}-\lambda\Omega\right)+\Omega. \tag{17}$$

We then apply McDiarmid's inequality. The the exponential moment is upper bounded by $(2\lambda\Delta\frac{n}{n-1})^2/8=\lambda^2\Delta^2/2\times n^2/(n-1)^2$. Hence,

$$.\mathbb{E}_{X^n,U}\sum_{k=1}^{K}\frac{1}{n-1}\sum_{m=1}^{n-1}\mathbb{E}_{q(J_{U,m}^{n-1}|\phi,X_{U,m}^{n-1})q(\phi,\theta|X_U^{n-1})}\frac{1}{|C_k|}\int_{C_k}\|X_U-g_\theta(z)\|^2dz\mathbb{1}_{k=J_{U,m}^{n-1}}$$

$$-\mathbb{E}_{X^n,U}\sum_{k=1}^{K}\frac{1}{n-1}\sum_{m=1}^{n-1}\mathbb{E}_{q(J_{U,m}^{n-1}|\phi,X_{U,m}^{n-1})q(\phi,\theta|X_U^{n-1})}\frac{1}{|C_k|}\int_{C_k}\|X_{U,m}^{n-1}-g_\theta(z)\|^2dz\mathbb{1}_{k=J_{U,m}^{n-1}}$$

$$\leq\mathbb{E}_X\frac{1}{\lambda}\mathrm{KL}\big(\mathbf{Q}|\mathbf{P}\big)+\mathbb{E}_X\frac{1}{\lambda}\log\mathbb{E}_{\mathbf{P}}\exp\left(\frac{2\lambda}{n-1}\sum_{m=1}^{n-1}\left(X_U-X_{U,m}^{n-1}\right)\cdot\sum_{k=1}^{K}\bar{g}_{\theta,k}\mathbb{1}_{k=J_{U,m}^{n-1}}-\lambda\Omega\right)+\Omega$$

$$\leq\mathbb{E}_X\frac{1}{\lambda}\mathrm{KL}\big(\mathbf{Q}|\mathbf{P}\big)+\frac{\lambda\Delta^2n^2}{2(n-1)^2}+\frac{\Delta}{\sqrt{n-1}}.$$

In conclusion,

$$\mathrm{gen}(n,\mathcal{D})\leq\mathbb{E}_X\frac{2}{\lambda}\mathrm{KL}\big(\mathbf{Q}|\mathbf{P}\big)+\frac{5\lambda\Delta^2n^2}{8(n-1)^2}+\frac{\Delta}{\sqrt{n-1}},$$

and, upon optimizing $\lambda$, we obtain

$$\mathrm{gen}(n,\mathcal{D})\leq\Delta\sqrt{\frac{5n^2\mathbb{E}_X\mathrm{KL}(\mathbf{Q}|\mathbf{P})}{(n-1)^2}}+\frac{\Delta}{\sqrt{n-1}}.$$

We can slightly improve the coefficient of the first term as follows. We apply the Donsker–Varadhan lemma separately to the first two terms and the last two terms in Eq. (12). However, since the posterior and prior used in the lemma coincide with those in Eq. (13), it suffices to invoke the lemma once, which improves the coefficient. Specifically, combining Eqs. (14) and (17) allows us to treat all terms in Eq. (12) simultaneously. By the Donsker–Varadhan lemma,

$$\mathrm{gen}(n,\mathcal{D})\leq\mathbb{E}_X\frac{1}{\lambda}\mathrm{KL}\big(\mathbf{Q}|\mathbf{P}\big)+\mathbb{E}_X\frac{1}{\lambda}\log\mathbb{E}_{\mathbf{P}}\exp\Big(\frac{\lambda}{n-1}\sum_{k=1}^{K}\sum_{m=1}^{n-1}\ell_C(X_U,\theta,k)\Big(\mathbb{1}_{k=J_U}-\mathbb{1}_{k=J_{U,m}^{n-1}}\Big)$$

$$+\frac{2\lambda}{n-1}\sum_{m=1}^{n-1}\left(X_U-X_{U,m}^{n-1}\right)\cdot\sum_{k=1}^{K}\bar{g}_{\theta,k}\mathbb{1}_{k=J_{U,m}^{n-1}}-\lambda\Omega\Big)+\Omega.$$

The exponential moment term can be rewritten using the invariance under $U''$ as

$$\log\mathbb{E}_{P(U)q(\phi,\theta|X_U^{n-1})\underset{P(U')}{\mathbb{E}}q(J^n|\phi,X_{U'},X_{U',m}^{n-1})}\mathbb{E}_{P(U'')}$$

$$\exp\left(\frac{\lambda}{n-1}\sum_{k=1}^{K}\sum_{m=1}^{n-1}\ell_C(X_U,\theta,k)\Big(\mathbb{1}_{k=J_{U''}}-\mathbb{1}_{k=J_{U'',m}^{n-1}}\Big)+\frac{2\lambda}{n}\sum_{m=1}^{n}\left(X_U-X_{U,m}^{n-1}\right)\cdot\sum_{k=1}^{K}\bar{g}_{\theta,k}\mathbb{1}_{k=J_{U,m}^{n-1}}-\lambda\Omega\right),$$

and, since $\{\mathbf{T}''_{j,m}\}$ are independent, McDiarmid's inequality applies. The the exponential moment is upper bounded by $((1+2)\Delta\lambda/n)^2/8 \times n^2/(n-1)^2 = 9\lambda^2\Delta^2/8n^2/(n-1)^2$. Therefore,

$$\text{gen}(n,\mathcal{D}) \leq \mathbb{E}_X \frac{1}{\lambda}\text{KL}(\mathbf{Q}|\mathbf{P}) + (9\lambda^2\Delta^2/8)n^2/(n-1)^2 + \frac{\Delta}{\sqrt{n-1}}. \tag{18}$$

Optimizing $\lambda$ yields

$$\text{gen}(n,\mathcal{D}) \leq \frac{3n}{\sqrt{2}(n-1)}\Delta\sqrt{\mathbb{E}_X\text{KL}(\mathbf{Q}|\mathbf{P})} + \frac{\Delta}{\sqrt{n-1}}.$$

Finally, we will show $\mathbb{E}_X\text{KL}(\mathbf{Q}|\mathbf{P}) \leq \mathbb{E}_{x^n \sim P^n}(I(\phi;U|x^n) + I(Z^n;U|\phi,x^n))$ in Appendix D.3.

### D.3. Derivation of KL divergence decomposition

Here we discuss how we can upper bound of the complexity term of the obtained bound. From the definition, we have the following relation;

$$\mathbb{E}_{X^n}\text{KL}(\mathbf{Q}\|\mathbf{P})$$

$$= \mathbb{E}_{P(X^n)p(U)q(\phi,\theta|X_U^{n-1})q(J^n|\phi,X_U,X_U^{n-1})} \log \frac{q(J^n|\phi,X_U,X_U^{n-1})}{\underset{P(U')}{\mathbb{E}} q(J^n|\phi,X_{U'},X_{U'}^{n-1})}$$

$$= \mathbb{E}_{P(X^n)p(U)q(\phi|X_U^{n-1})q(J^n|\phi,X_U,X_U^{n-1})} \log \frac{q(J^n|\phi,X_U,X_U^{n-1})}{\underset{P(U')}{\mathbb{E}} q(J^n|\phi,X_{U'},X_{U'}^{n-1})}$$

$$= \mathbb{E}_{P(X^n)P(\phi|X^n)}\mathbb{E}_{P(U|\phi,X^n)q(J^n|\phi,X_U,X_U^{n-1})} \log \frac{q(J^n|\phi,X_U,X_U^{n-1})}{\underset{P(U')}{\mathbb{E}} q(J^n|\phi,X_{U'},X_{U'}^{n-1})}$$

$$= \mathbb{E}_{P(X^n)P(\phi|X^n)}\mathbb{E}_{P(U|\phi,X^n)q(J^n|\phi,X_U,X_U^{n-1})} \log \frac{q(J^n|\phi,X_U,X_U^{n-1})}{\underset{P(U'|\phi,X^n)}{\mathbb{E}} q(J^n|\phi,X_{U'},X_{U'}^{n-1})}$$

$$\quad + \mathbb{E}_{P(X^n)P(\phi|X^n)}\mathbb{E}_{P(U|\phi,X^n)q(J^n|\phi,X_U,X_U^{n-1})} \log \frac{\underset{P(U'|\phi,X^n)}{\mathbb{E}} q(J^n|\phi,X_{U'},X_{U'}^{n-1})}{\underset{P(U')}{\mathbb{E}} q(J^n|\phi,X_{U'},X_{U'}^{n-1})}$$

$$= I(J^n;U|\phi,X^n) + \mathbb{E}_{P(X^n)P(\phi|X^n)}\mathbb{E}_{P(U|\phi,X^n)q(J^n|\phi,X_U,X_U^{n-1})} \log \frac{\underset{P(U'|\phi,X^n)}{\mathbb{E}} q(J^n|\phi,X_{U'},X_{U'}^{n-1})}{\underset{P(U')}{\mathbb{E}} q(J^n|\phi,X_{U'},X_{U'}^{n-1})}$$

$$\leq I(J^n;U|\phi,X^n) + \mathbb{E}_{P(X^n)P(\phi|X^n)}\mathbb{E}_{P(U|\phi,X^n)} \log \frac{P(U'|\phi,X^n)}{P(U')}$$

$$= I(J^n;U|\phi,X^n) + \mathbb{E}_{P(X^n)P(\phi|X^n)}\mathbb{E}_{P(U|\phi,X^n)} \log \frac{P(\phi|X^n,U')P(U'|X^n)}{\mathbb{E}_{P(U''|X^n)}P(\phi|X^n,U'')P(U')}$$

$$= I(J^n;U|\phi,X^n) + \mathbb{E}_{P(X)P(\phi|X^n)}\mathbb{E}_{P(U|\phi,X^n)} \log \frac{P(\phi|X^n,U')P(U')}{\mathbb{E}_{P(U'')}P(\phi|X^n,U'')P(U')}$$

$$= I(J^n;U|\phi,X^n) + I(\phi;U|X),$$

where we used the data processing inequality of the KL divergence.

## E. Qualitative approximation of the LOO-CMI

Here we derive the approximation in Section 3.3 following the approach described in Rammal et al. (2022).

First, we inject Gaussian noise into the output of the algorithm: $A_\Sigma(x_U^{n-1}) = A(x_U^{n-1}) + \epsilon$, $\epsilon \sim \mathcal{N}(0,\Sigma)$, where $\Sigma$ is a positive definite matrix specified later. We denote this as $W_\Sigma = (\theta_\Sigma,\phi_\Sigma) = A_\Sigma(x_U^{n-1})$. Moreover, let $W_{\Sigma,i} = (\theta_{\Sigma,i},\phi_{\Sigma,i}) = A_\Sigma(x_{U=i}^{n-1})$ denote the output when the $i$-th sample is removed from the training set.

**Corollary 1** (Modified version of Corollary 4.1 in Rammal et al. (2022)). *Let $W_{\Sigma,-i} = (\theta_{\Sigma,-i}, \phi_{\Sigma,-i}) = A_\Sigma(x_{U=i}^{n-1})$ denote the output when the $i$-th sample is removed from the training set. Then*

$$I(W_\Sigma; U|x^n) \le -\frac{1}{n}\sum_{i=1}^n \ln\left(\frac{1}{n}\sum_{j=1}^n \exp(-\Omega_{ij})\right), \quad \Omega_{ij} = \tfrac{1}{2}(W_{\Sigma,-i} - W_{\Sigma,-j})^\top \Sigma^{-1}(W_{\Sigma,-i} - W_{\Sigma,-j}).$$

The connection between classical notions of stability and the right-hand side above is clear: to compute it, one only needs to evaluate how the algorithm's weights change when two datasets differ by a single sample.

Finally, Rammal et al. (2022) proposed to approximate Corollary 1 using the influence function, thereby deriving a qualitative approximation of the bound in terms of local quantities. A similar argument can be derived in our VAE setting.

Assume that the training algorithm minimizes the loss

$$w^* = A(x^n) = \arg\min_{w\in\mathcal{W}} \ell_{\mathrm{vae}}(x^n, w),$$

where $\ell_{\mathrm{vae}}$ is defined in Eq. (1). Similarly, let $w_{U=i}^* = A(X_{U=i}^n)$ denote the output when the $i$-th sample is replaced. Using influence functions (Koh & Liang, 2017), we approximate:

$$w_i^* - w^* \approx \frac{1}{n} H_{w^*}^{-1} \nabla_w \ell_{\mathrm{vae}}(x_i, w^*),$$

where $H_{w^*}$ is the Hessian of $\ell_{\mathrm{vae}}(x^n, w)$ at $w^*$, $\nabla_w \ell_{\mathrm{vae}}(x_i, w^*)$ is the gradient of the loss on $x_i$ at $w^*$, and we define the per-sample gradient at convergence, rescaled by the Hessian, as

$$\delta_i = \frac{1}{n} H_{w^*}^{-1} \nabla_w \ell_{\mathrm{vae}}(x_i, w^*).$$

By setting $\Sigma = H_{w^*}^{-1}$ in Corollary 1, we obtain

$$
\begin{aligned}
I(W_\Sigma; U|x^n) &\le \ln n - \frac{1}{n}\sum_{i=1}^n \ln\left(\sum_{j=1}^n \exp\{-\tfrac{1}{2}(w_i - w_j)^\top \Sigma^{-1}(w_i - w_j)\}\right), \\
&\approx \ln n - \frac{1}{n}\sum_{i=1}^n \ln\left(\sum_{j=1}^n \exp\{-\tfrac{1}{2}(\delta_i - \delta_j)^\top \Sigma^{-1}(\delta_i - \delta_j)\}\right), \\
&= \ln n - \frac{1}{n}\sum_{i=1}^n \ln\left(\sum_{j=1}^n \exp\{-\tfrac{1}{2}(\delta_i - \delta_j)^\top H_{w^*}(\delta_i - \delta_j)\}\right) \\
&\le \frac{1}{n}\sum_{i=1}^n \frac{1}{n}\sum_{j=1}^n \frac{1}{2}(\delta_i - \delta_j)^\top H_{w^*}(\delta_i - \delta_j).
\end{aligned}
\tag{19}
$$

Actually, this entails all the parameters ($W$), and what we need to estimate is the MI related only to $\phi$. Therefore, we slightly modify above approximation. We introduce a projection matrix $P_\phi$ onto the $\phi$ component such that $\phi_\Sigma = P_\phi W_\Sigma$. We then define the corresponding partial covariance matrix as $\Sigma_\phi = P_\phi \Sigma P_\phi^\top$. Given $a, b \in \mathbb{R}^{d_\phi}$, we define $\|a - b\|_{\Sigma_\phi^{-1}}^2 = (a - b)^\top \Sigma_\phi^{-1}(a - b)$. We then obtain the following result. Under the above notations, we have

$$I(\phi_\Sigma; U|x^n) \le -\frac{1}{n}\sum_{i=1}^n \ln \frac{1}{n}\sum_{j=1}^n e^{-\frac{1}{2}\|\phi_{\Sigma,i} - \phi_{\Sigma,j}\|_{\Sigma_\phi^{-1}}^2}.$$

This result is a modified version of Corollary 1

A similar argument can be derived in our VAE setting. Assume that the training algorithm of the VAEs minimizes the loss

$$w^* = A(x^n) = \arg\min_{w\in\mathcal{W}} \ell_{\mathrm{vae}}(x^n, w),$$

where $\ell_{\text{vae}}$ is defined in Eq. (1). Similarly, let $w_i^* = (\theta_i^*, \phi_i^*) = A(x_{U=i}^n)$ denote the output when the $i$-th sample is replaced. Using influence functions (Koh & Liang, 2017), we approximate:

$$\phi_i^* - \phi^* \approx \frac{1}{n}(H_{w^*}^\phi)^{-1}\nabla_\phi \ell_{\text{vae}}(x_i, w^*) \coloneqq \delta_{\phi_i},$$

where $H_{w^*}$ is the Hessian of $\ell_{\text{vae}}(x^n, w)$ at $w^*$ and $H_{w^*}^\phi$ is the projection onto $\phi$ component, and $\nabla_\phi \ell_{\text{vae}}(x_i, w^*)$ is the gradient of the loss on $x_i$ at $w^*$.

By setting $\Sigma = H_{w^*}^{-1}$, we can obtain the following approximation from Eq. (19):

$$I(\phi_\Sigma; U | x^n) \lesssim -\frac{1}{n}\sum_{i=1}^n \ln \frac{1}{n}\sum_{j=1}^n e^{-\frac{1}{2}\|\delta_{\phi_i} - \delta_{\phi_j}\|_{H_{w^*}^\phi}^2}.$$

## F. Proofs for Section 2 and Additional Discussion

### F.1. Proof of Theorem 1

This result follows directly from the existing LOO-CMI bounds in Rammal et al. (2022) and Haghifam et al. (2022). Let $\hat{X}$ denote the reconstructed data. Given that the loss is bounded within $[0, \Delta]$, the integrated loss is a $\Delta$-sub-Gaussian random variable. Therefore, by applying the functional LOO-CMI framework of Rammal et al. (2022) (Theorem 3.2), we obtain

$$\text{gen}(n, \mathcal{D}) \leq \frac{n}{\sqrt{2}(n-1)}\sqrt{I(\hat{X}; U \mid X^n)}.$$

By the data processing inequality (DPI), we further have $I(\hat{X}; U \mid X^n) \leq I(W; U \mid X^n)$, which completes the proof.

### F.2. Limitations of Theorem 1 in Analyzing Latent Variables

Here, we show that the upper bound in Theorem 1 is unsuitable for analyzing LVs. The bound depends on the LOO-CMI terms $I(\hat{X}; U \mid X^n)$ or $I(W; U \mid X^n)$, both of which involve the decoder and encoder information. This differs fundamentally from our main results, which remain independent of the decoder.

To clarify this distinction, observe that

$$I(\hat{X}; U \mid X^n) \leq I(\theta; U \mid X^n) + I(Z^n; U \mid X^n, \theta),$$

where the inequality follows from the chain rule of conditional mutual information and the DPI. This result shows that the decoder information cannot be eliminated from the IT bound, revealing a key conceptual difference from our formulation. Furthermore, since both the encoder and decoder are trained jointly on the same data, they are inherently dependent. This interdependence makes it difficult to isolate the effects of the LVs and the encoder's capacity on generalization, as the decoder's influence cannot be easily disentangled.

## G. Proof of Theorem 4

Before the proof, we define the Wasserstein distance. Given a metric $d(\cdot, \cdot)$ and probability distributions $p$ and $q$ on $\mathcal{X}$, let $\Pi(p, q)$ denote the set of all couplings of $p$ and $q$. The 2-Wasserstein distance is defined as:

$$W_2(p, q) = \sqrt{\inf_{\rho \in \Pi}\int_{\mathcal{X}\times\mathcal{X}} d(x, x')^2 d\rho(x, x')}.$$

In this work, we use the Euclidean metric $|\cdot|$ as $d(\cdot, \cdot)$.

Next, we define the pushforward. Let $\pi$ represent a distribution on $\mathcal{Z}$, and let us assume that for any $\theta \in \Theta$, the decoder $g_\theta(\cdot): \mathcal{Z} \to \mathcal{X}$ is measurable. The pushforward of the distribution $\pi$ by the decoder, denoted as $g_\theta \# \pi$, defines a distribution on $\mathcal{X}$ as $g_\theta \# \pi(A) = \pi(g_\theta^{-1}(A))$ for any measurable set $A \subseteq \mathcal{X}$.

*Proof.* We first define the distribution obtained by the training dataset as follows; conditioned on $\phi, S = (X_1, \ldots, X_{n-1})$, we have

$$\hat{\mu}_S = \frac{1}{n-1} \sum_{m=1}^{n-1} g_\theta \# q(z|\phi, X_m).$$

All Wasserstein quantities below are evaluated conditionally on $(\phi, \theta, S)$, since both the encoder distribution $q(z \mid \phi, X_m)$ and the decoder pushforward $g_\theta \# q(z \mid \phi, X_m)$ depend on these variables. From the triangle inequality, we have

$$W_2(\mathcal{D}, \hat{\mu}) \leq W_2(\mathcal{D}, \hat{\mu}_S) + W_2(\hat{\mu}_S, \hat{\mu}). \tag{20}$$

We then have

$$W_2^2(\mathcal{D}, \hat{\mu}) \leq 2W_2^2(\mathcal{D}, \hat{\mu}_S) + 2W_2^2(\hat{\mu}_S, \hat{\mu}).$$

The first term of Eq. (20) is bounded as follows;

$$\mathbb{E}_S \mathbb{E}_{q(\phi, \theta|S)} W_2^2(\mathcal{D}, \hat{\mu}_S) \leq \mathbb{E}_S \mathbb{E}_{q(\phi, \theta|S)} \mathbb{E}_X \frac{1}{n-1} \sum_{m=1}^{n-1} \mathbb{E}_{q(z|\phi, X_m)} \|X - g_\theta(z)\|^2$$

This inequality is obtained by the definition of the Wasserstein distance. From the truncation and discretization argument, using Eq. (9), we have

$$\mathbb{E}_S \mathbb{E}_{q(\phi, \theta|S)} \mathbb{E}_X \frac{1}{n-1} \sum_{m=1}^{n-1} \mathbb{E}_{q(Z_m|\phi, X_m)} \|X - g_\theta(Z_m)\|^2$$

$$\leq \mathbb{E}_{X,S} \sum_{k=1}^{K} \frac{1}{n-1} \sum_{m=1}^{n-1} \mathbb{E}_{q(J_m^{n-1}|\phi, X_m^{n-1})q(\phi, \theta|S)} \ell_C(X, \theta, k) \mathbb{1}_{k=J_m^{n-1}} + \Delta \frac{M^2 + S^2}{R^2} + \frac{3\Delta\sqrt{d_z}}{sR}, \tag{21}$$

where we used the notations in Appendix D.2.

Then we show in Appendix G.2 that

$$\mathbb{E}_{X,S} \sum_{k=1}^{K} \frac{1}{n-1} \sum_{m=1}^{n-1} \mathbb{E}_{q(J_m^{n-1}|\phi, X_m^{n-1})q(\phi, \theta|S)} \ell_C(X, \theta, k) \mathbb{1}_{k=J_m^{n-1}}$$

$$\leq \mathbb{E}_{X,S} \sum_{k=1}^{K} \frac{1}{n-1} \sum_{m=1}^{n-1} \mathbb{E}_{q(J_m^{n-1}|\phi, X_m^{n-1})q(\phi, \theta|S)} \ell_C(X_m^{n-1}, \theta, k) \mathbb{1}_{k=J_{U,m}^{n-1}}$$

$$+ \frac{1}{\lambda} \mathrm{KL}(\mathbf{Q}|\mathbf{P}) + \frac{\lambda\Delta^2}{2(n-1)} + \frac{\Delta}{\sqrt{(n-1)}} \tag{22}$$

where

$$\mathbf{Q} := q(\phi, \theta|S) \prod_{m=1}^{n-1} q(Z_m|\phi, X_m), \qquad \mathbf{P} := q(\phi, \theta|S) \prod_{m=1}^{n-1} p(Z_m).$$

For the discretized variables, define

$$\mathbf{Q}_J := q(\phi, \theta \mid S) \prod_{m=1}^{n-1} q(J_m \mid \phi, X_m), \qquad \mathbf{P}_J := q(\phi, \theta \mid S) \prod_{m=1}^{n-1} \pi(J_m).$$

The DV bound is applied to $\mathbf{Q}_J$ and $\mathbf{P}_J$. The resulting KL term is then upper bounded by the corresponding continuous-latent KL via data processing under the map $Z_m \mapsto J_m$.

Then similarly to Eq. (21), by using the truncation and discretization argument, we have

$$\mathbb{E}_{X,S} \sum_{k=1}^{K} \frac{1}{n-1} \sum_{m=1}^{n-1} \mathbb{E}_{q(J_m^{n-1}|\phi,X_m^{n-1})q(\phi,\theta|S)} \ell_C(X_m^{n-1},\theta,k) \mathbb{1}_{k=J_{U,m}^{n-1}}$$

$$\leq \mathbb{E}_S \mathbb{E}_{q(\phi,\theta|S)} \frac{1}{n-1} \sum_{m=1}^{n-1} \mathbb{E}_{q(Z_m|\phi,X_m)} \|X_m - g_\theta(Z_m)\|^2 + \Delta \frac{M^2+S^2}{R^2} + \frac{3\Delta\sqrt{d_z}}{sR}$$

Combining these we have

$$\mathbb{E}_S \mathbb{E}_{q(\phi,\theta|S)} W_2^2(\mathcal{D}, \hat{\mu}_S)$$

$$\leq \mathbb{E}_S \mathbb{E}_{q(\phi,\theta|S)} \frac{1}{n-1} \sum_{m=1}^{n-1} \mathbb{E}_{q(Z_m|\phi,X_m)} \|X_m - g_\theta(Z_m)\|^2 + \frac{1}{\lambda} \mathrm{KL}(\mathbf{Q}|\mathbf{P}) + \frac{\lambda\Delta^2}{2(n-1)} + \frac{\Delta}{\sqrt{(n-1)}}$$

$$+ 2\Delta \frac{M^2+S^2}{R^2} + \frac{6\Delta\sqrt{d_z}}{sR}$$

Next, the second term of Eq. (20) is bounded as follows; we use the weighted CKP inequality (Bolley & Villani, 2005). From the particular case 2.5. in Bolley & Villani (2005), we directly have

$$\mathbb{E}_S \mathbb{E}_{q(\phi,\theta|S)} W_2^2(\hat{\mu}_S, \hat{\mu}) \leq \Delta\sqrt{2\mathrm{KL}(\hat{\mu}_S\|\hat{\mu})} \leq \Delta\sqrt{2\frac{1}{n-1} \sum_{m=1}^{n-1} \mathrm{KL}(g_\theta \# q(z|\phi,X_m)\|g_\theta \# p(z))} \quad (23)$$

$$\leq \Delta\sqrt{2\frac{1}{n-1} \sum_{m=1}^{n-1} \mathrm{KL}(q(Z_m|\phi,X_m)\|p(Z_m))}$$

We can further improve the constant as follows; From Lemma 5, and $\mathcal{X}$: since $\|x-x'\|^2 \leq \Delta$ for $x,x' \in \mathcal{X}$, we have

$$W_2^2(\mu,\nu) \leq \Delta \mathrm{TV}(\mu,\nu)$$

Then we use the Pinsker inequality, we have

$$W_2^2(\mu,\nu) \leq \Delta\sqrt{\frac{1}{2}\mathrm{KL}(\mu\|\nu)}.$$

Thus, we have

$$\mathbb{E}_S \mathbb{E}_{q(\phi,\theta|S)} W_2^2(\hat{\mu}_S, \hat{\mu}) \leq \Delta\sqrt{\frac{1}{2}\mathrm{KL}(\hat{\mu}_S\|\hat{\mu})} \leq \Delta\sqrt{\frac{1}{2}\frac{1}{n-1} \sum_{m=1}^{n-1} \mathrm{KL}(g_\theta \# q(z|\phi,X_m)\|g_\theta \# p(z))}$$

$$\leq \Delta\sqrt{\frac{1}{2}\frac{1}{n-1} \sum_{m=1}^{n-1} \mathrm{KL}(q(Z_m|\phi,X_m)\|p(Z_m))}$$

Combining Eqs. (22) and (23), we have

$$\mathbb{E}_S \mathbb{E}_{q(\phi,\theta|S)} W_2^2(\mathcal{D}, \hat{\mu}) \leq 2\mathbb{E}_S \mathbb{E}_{q(\phi,\theta|S)} \frac{1}{n-1} \sum_{m=1}^{n-1} \mathbb{E}_{q(Z_m|\phi,X_m)} \|X_m - g_\theta(Z_m)\|^2$$

$$+ \frac{2}{\lambda} \mathrm{KL}(\mathbf{Q}|\mathbf{P}) + \frac{\lambda\Delta^2}{(n-1)} + \frac{2\Delta}{\sqrt{(n-1)}} + \Delta\sqrt{2\frac{1}{n-1} \sum_{m=1}^{n-1} \mathrm{KL}(q(Z_m|\phi,X_m)\|p(Z_m))}$$

$$+ 4\Delta \frac{M^2+S^2}{R^2} + \frac{12\Delta\sqrt{d_z}}{sR}$$

Then by optimizing $\lambda$, we have

$$\mathbb{E}_S \mathbb{E}_{q(\phi,\theta|S)} W_2^2(\mathcal{D}, \hat{\mu})$$

$$\leq 2\mathbb{E}_S \mathbb{E}_{q(\phi,\theta|S)} \frac{1}{n-1} \sum_{m=1}^{n-1} \mathbb{E}_{q(Z_m|\phi,X_m)} \|X_m - g_\theta(Z_m)\|^2 + 3\Delta \sqrt{2\frac{1}{n-1} \sum_{m=1}^{n-1} \mathrm{KL}(q(Z_m|\phi,X_m)\|p(Z_m)) + \frac{2\Delta}{\sqrt{n-1}}}$$

$$+ 4\Delta \frac{M^2 + S^2}{R^2} + \frac{12\Delta\sqrt{d_z}}{sR}.$$

Then similar to the argument in Appendix C.4, we take the limit $R \to \infty$ and the last two terms will vanish but other terms will not be affected by this limit. This concludes the proof $\qquad\square$

### G.1. Proof of Eq. (G)

We use the following bounded-diameter coupling estimate.

**Lemma 5** (Bounded-diameter coupling estimate). *Let $(X, d)$ be a metric space with finite diameter*

$$D := \mathrm{diam}(X) < \infty.$$

*Then, for any probability measures $\mu, \nu$ on $X$ and any $p \geq 1$,*

$$W_p^p(\mu, \nu) \leq D^p \, \mathrm{TV}(\mu, \nu),$$

*where*

$$\mathrm{TV}(\mu, \nu) = \frac{1}{2}\|\mu - \nu\|_1.$$

*Proof.* Let

$$\rho := \mu + \nu.$$

Write

$$f := \frac{d\mu}{d\rho}, \qquad g := \frac{d\nu}{d\rho}.$$

Define the common part $\lambda := \mu \wedge \nu$ by

$$d\lambda := \min\{f, g\} \, d\rho.$$

Then $\lambda \leq \mu$ and $\lambda \leq \nu$. Moreover,

$$\min\{f, g\} = \frac{f + g - |f - g|}{2}.$$

Therefore,

$$\lambda(X) = \int_X \min\{f, g\} \, d\rho = \frac{1}{2} \int_X f \, d\rho + \frac{1}{2} \int_X g \, d\rho - \frac{1}{2} \int_X |f - g| \, d\rho = 1 - \frac{1}{2}\|\mu - \nu\|_1 = 1 - \mathrm{TV}(\mu, \nu).$$

Put

$$\eta := \mathrm{TV}(\mu, \nu).$$

If $\eta = 0$, then $\mu = \nu$, and hence

$$W_p^p(\mu, \nu) = 0,$$

so the claim is trivial.

Assume now that $\eta > 0$. Since $\lambda \leq \mu$ and $\lambda \leq \nu$, the signed measures $\mu - \lambda$ and $\nu - \lambda$ are nonnegative finite measures. Moreover,

$$(\mu - \lambda)(X) = \mu(X) - \lambda(X) = 1 - (1 - \eta) = \eta,$$

and similarly

$$(\nu - \lambda)(X) = \eta.$$

Hence

$$\mu_r := \frac{\mu - \lambda}{\eta}, \qquad \nu_r := \frac{\nu - \lambda}{\eta}$$

are probability measures on $X$.

Define a probability measure $\pi$ on $X \times X$ by

$$\pi(dx, dy) = \lambda(dx)\delta_x(dy) + \eta \, \mu_r(dx)\nu_r(dy).$$

First, $\pi$ has total mass one, because

$$\pi(X \times X) = \lambda(X) + \eta \, \mu_r(X)\nu_r(X) = (1 - \eta) + \eta = 1.$$

Next, its first marginal is $\mu$. Indeed, for any measurable set $A \subseteq X$,

$$\begin{aligned}
\pi(A \times X) &= \lambda(A) + \eta \, \mu_r(A)\nu_r(X) \\
&= \lambda(A) + \eta \, \mu_r(A) \\
&= \lambda(A) + (\mu - \lambda)(A) \\
&= \mu(A).
\end{aligned}$$

Similarly, its second marginal is $\nu$, since for any measurable set $B \subseteq X$,

$$\begin{aligned}
\pi(X \times B) &= \lambda(B) + \eta \, \mu_r(X)\nu_r(B) \\
&= \lambda(B) + \eta \, \nu_r(B) \\
&= \lambda(B) + (\nu - \lambda)(B) \\
&= \nu(B).
\end{aligned}$$

Therefore $\pi \in \Pi(\mu, \nu)$.

Using this coupling, we obtain

$$\begin{aligned}
W_p^p(\mu, \nu) &= \inf_{\gamma \in \Pi(\mu,\nu)} \int_{X \times X} d(x, y)^p \, d\gamma(x, y) \\
&\leq \int_{X \times X} d(x, y)^p \, d\pi(x, y) \\
&= \int_X d(x, x)^p \, d\lambda(x) + \eta \int_{X \times X} d(x, y)^p \, \mu_r(dx)\nu_r(dy) \\
&= \eta \int_{X \times X} d(x, y)^p \, \mu_r(dx)\nu_r(dy) \\
&\leq \eta D^p \\
&= D^p \, \mathrm{TV}(\mu, \nu).
\end{aligned}$$

This proves the claim. $\square$

### G.2. Proof of Eq. (22)

Bounding Eq. (21) is equivalent to bounding third and fourth terms in Eq. (12).

Here, we do not have to utilize the LOO-CMI setting, the third and fourth terms can be rewritten as

$$
\mathbb{E}_{X,S} \sum_{k=1}^{K} \frac{1}{n-1} \sum_{m=1}^{n-1} \mathbb{E}_{q(J_m^{n-1}|\phi, X_m^{n-1})q(\phi,\theta|S)} \ell_C(X, \theta, k) \mathbb{1}_{k=J_m^{n-1}}
$$

$$
- \mathbb{E}_{X,S} \sum_{k=1}^{K} \frac{1}{n-1} \sum_{m=1}^{n-1} \mathbb{E}_{q(J_m^{n-1}|\phi, X_m^{n-1})q(\phi,\theta|S)} \ell_C(X_m^{n-1}, \theta, k) \mathbb{1}_{k=J_{U,m}^{n-1}}
$$

$$
= \mathbb{E}_{X,S} \sum_{k=1}^{K} \frac{1}{n-1} \sum_{m=1}^{n-1} \mathbb{E}_{q(J_m^{n-1}|\phi, X_m^{n-1})q(\phi,\theta|S)} \frac{1}{|C_k|} \int_{C_k} \|X - g_\theta(z)\|^2 dz \mathbb{1}_{k=J_m^{n-1}}
$$

$$
- \mathbb{E}_{X,S} \sum_{k=1}^{K} \frac{1}{n-1} \sum_{m=1}^{n-1} \mathbb{E}_{q(J_m^{n-1}|\phi, X_m^{n-1})q(\phi,\theta|S)} \frac{1}{|C_k|} \int_{C_k} \|X_m^{n-1} - g_\theta(z)\|^2 dz \mathbb{1}_{k=J_m^{n-1}}
$$

$$
= \mathbb{E}_{X,S} \frac{2}{n-1} \sum_{m=1}^{n-1} (X - X_m^{n-1}) \cdot \mathbb{E}_{q(J_m^{n-1}|\phi, X_m^{n-1})q(\phi,\theta|S)} \sum_{k=1}^{K} \bar{g}_{\theta,k} \mathbb{1}_{k=J_m^{n-1}}
$$

$$
\leq \mathbb{E}_S \frac{1}{\lambda} \mathrm{KL}(\mathbf{Q}|\mathbf{P}) + \mathbb{E}_X \frac{1}{\lambda} \log \mathbb{E}_{\mathbf{P}} \exp \left( \frac{2\lambda}{n-1} \sum_{m=1}^{n-1} (\mathbb{E}_X X - X_m^{n-1}) \cdot \sum_{k=1}^{K} \bar{g}_{\theta,k} \mathbb{1}_{k=J_m^{n-1}} \right) \qquad (24)
$$

where we used the Donsker-Valadhan inequality between

$$
\mathbf{Q} := q(\phi, \theta|S) \prod_{m=1}^{n} q(J_m|\phi, S_m),
$$

$$
\mathbf{P} := q(\phi, \theta|S) \prod_{m=1}^{n} \pi(J_m|\phi), \qquad (25)
$$

in the last line. Here $\pi(J_m|\phi)$ is the prior distribution, which never depends on the training data. We remark that any data dependent prior is possible to use. (Dependency on $\phi$ will not be a problem.)

Then Eq. (24) will be bounded as

$$
\leq \mathbb{E}_X \frac{1}{\lambda} \mathrm{KL}(\mathbf{Q}|\mathbf{P}) + \mathbb{E}_X \frac{1}{\lambda} \log \mathbb{E}_{\mathbf{P}} \exp \left( \frac{2\lambda}{n-1} \sum_{m=1}^{n-1} (\mathbb{E}_X X - X_m^{n-1}) \cdot \sum_{k=1}^{K} \bar{g}_{\theta,k} \mathbb{1}_{k=J_m^{n-1}} \right)
$$

$$
\leq \mathbb{E}_S \frac{1}{\lambda} \mathrm{KL}(\mathbf{Q}|\mathbf{P})
$$

$$
+ \mathbb{E}_S \frac{1}{\lambda} \log \mathbb{E}_{\mathbf{P}} \exp \left( \frac{2\lambda}{n-1} \sum_{m=1}^{n-1} (\mathbb{E}_X X - X_m^{n-1}) \cdot \sum_{k=1}^{K} \bar{g}_{\theta,k} (\mathbb{1}_{k=J_m} - P''_{k,m}) \right)
$$

$$
+ \mathbb{E}_S \mathbb{E}_{\mathbf{P}} \frac{2}{n} \sum_{m=1}^{n-1} (\mathbb{E}_X X - X_m^{n-1}) \cdot \sum_{k=1}^{K} \bar{g}_{\theta,k} P''_{k,m},
$$

where $P''_{k,m} = \mathbb{E}_{q(J_m|\phi)} \mathbb{1}_{k=J_m}$. Clearly, this does not depend on the index $m$, so we express $P''_{k,m} = P''_k$. Then the last

term becomes

$$
\mathbb{E}_S \mathbb{E}_{\mathbf{P}} \frac{1}{n-1} \sum_{m=1}^{n-1} \left( \mathbb{E}_X X - X_m^{n-1} \right) \cdot \sum_{k=1}^{K} \bar{g}_{\theta,k} P_k'' \leq \mathbb{E}_S \mathbb{E}_{\mathbf{P}} \left\| \mathbb{E}_X X - \frac{1}{n-1} \sum_{m=1}^{n-1} X_m^{n-1} \right\|_2 \| \sum_{k=1}^{K} \bar{g}_{\theta,k} P_k'' \|_2
$$

$$
\leq \mathbb{E}_S \left\| \mathbb{E}_X X - \frac{1}{n-1} \sum_{m=1}^{n-1} X_m^{n-1} \right\| \sqrt{\Delta}
$$

$$
\leq \sqrt{ \Delta \mathrm{Var} \left( \frac{1}{n-1} \sum_{m=1}^{n-1} X_m^{n-1} \right) }
$$

$$
\leq \sqrt{ \Delta \frac{\mathrm{Var}(X)}{n-1} }
$$

$$
\leq \sqrt{ \frac{\Delta}{4(n-1)} } \sqrt{\Delta} = \frac{\Delta}{2\sqrt{n-1}}, \tag{26}
$$

where we used the fact that the variance of random variables with bounded in $(a, b]$ is upper bounded by $(b-a)^2/4n$ (the extension to the $d$-dimensional random variable is straightforward) and thus, $\mathrm{Var}(X) \leq \Delta/4$. Then the exponential moment term becomes

$$
\mathbb{E}_S \frac{1}{\lambda} \log \mathbb{E}_{\mathbf{P}} \exp \left( \frac{2\lambda}{n-1} \sum_{m=1}^{n-1} \left( \mathbb{E}_X X - X_m^{n-1} \right) \cdot \sum_{k=1}^{K} \bar{g}_{\theta,k} ( \mathbb{1}_{k=J_m} - P_{k,m}'' ) \right)
$$

$$
= \mathbb{E}_S \frac{1}{\lambda} \log \mathbb{E}_{\mathbf{P}} \exp \left( \frac{2\lambda}{n-1} \sum_{m=1}^{n-1} \left( \mathbb{E}_X X - X_m^{n-1} \right) \cdot \sum_{k=1}^{K} \bar{g}_{\theta,k} ( \mathbb{1}_{k=J_m} - P_k'' ) \right).
$$

Here we use McDiarmid's inequality for $n-1$ random variables $\mathbf{J}$. Then we estimate the bounded difference constants using the stability argument, which is upper bounded by $2\lambda \Delta/(n-1)$. Then from Lemma 4, the exponential moment is bounded by $(2\lambda\Delta/n-1)^2/8 \times (n-1) = \lambda\Delta^2/2(n-1)$

Thus, the second term is upper bounded by

$$
\frac{1}{\lambda} \mathrm{KL}(\mathbf{Q}|\mathbf{P}) + \frac{\lambda\Delta^2}{2(n-1)} + \frac{\Delta}{\sqrt{n-1}}.
$$

Finally, by definition (Eq. (25)), KL divergence term can be evaluated by

$$
\mathrm{KL}(\mathbf{Q}|\mathbf{P}) = \mathbb{E}_{q(\phi,\theta|S) \prod_{m=1}^{n} q(J_m|\phi, S_m)} \log \frac{q(\phi,\theta|S) \prod_{m=1}^{n} q(J_m|\phi, S_m)}{q(\phi,\theta|S) \prod_{m=1}^{n} \pi(J_m|\phi)}
$$

$$
\leq \mathbb{E}_{q(\phi,\theta|S) \prod_{m=1}^{n} q(Z_m|\phi, S_m)} \log \frac{\prod_{m=1}^{n} q(Z_m|\phi, S_m)}{\prod_{m=1}^{n} \pi(Z_m|\phi)}
$$

where we used the fact that $X - Z - Z_R - J$ forms a Markov chain and above inequality comes from the data-processing inequality of KL divergence. This concludes the proof.

## H. Proofs for the Hierarchical VAEs

### H.1. Proof of Theorem 3

This proof closely follows the proof of Theorem 2 in Appendix C. Specifically, the random variable $Z$ in Appendix C can be interpreted as $Z_l$ in the hierarchical setting. Since we impose equivalent assumptions on each layer, the same proof strategy applies directly.

The only difference lies in the decoder setting: Theorem 2 assumes a deterministic decoder $g_\theta(x)$, whereas Theorem 3 extend it to the probabilistic decoder. When considering the dirac mass, it corresponds to a deterministic decoder. Accordingly, the

generalization error is defined as

$$\mathrm{gen}_{\mathrm{H}}(n, \mathcal{D}) := \left| \mathbb{E}_{S,X} \mathbb{E}_{q(\phi_{1:L+1}|S)} \left[ \ell_0(\phi_{1:L+1}, X) - \frac{1}{n-1} \sum_{i=1}^{n-1} \ell_0(\phi_{1:L+1}, X_i) \right] \right|.$$

This definition is nearly identical to that in Theorem 2, except for the reconstruction loss:

$$\begin{aligned}
\ell_0(\phi_{1:L+1}, x) &:= \mathbb{E}_{p(z_{1:L+1}|x)} \|x - Z_{L+1}\|^2 \\
&= \mathbb{E}_{\prod_{l=1}^{L+1} q_l(z_l|\phi_l, z_{l-1})} \|x - Z_{L+1}\|^2 \\
&= \mathbb{E}_{\hat{x} \sim p(x|\phi_{L+1}, z_L) \prod_{l=1}^{L} q_l(z_l|\phi_l, z_{l-1})} \|x - \hat{x}\|^2,
\end{aligned}$$

where the reconstructed data is generated stochastically.

Since the reconstruction loss $\ell : \mathcal{X} \times \mathcal{X} \to \mathbb{R}$ is defined over $\mathcal{X}$, we implicitly assume that $\hat{x} \in \mathcal{X}$. When the decoder distribution is Gaussian and $\hat{x} \notin \mathcal{X}$, we may project it back to $\mathcal{X}$ (e.g., via clipping). For the squared loss, even if $\hat{x} \notin \mathcal{X}$, the reconstruction error remains upper bounded when the second moment of the decoder distribution is bounded. This modification only changes the coefficient of the bound, not the proof strategy. Thus, for simplicity, we assume $\hat{x} \in \mathcal{X}$ so that the reconstruction loss is always bounded by $\Delta$.

To make the correspondence to Theorem 2 in Appendix C explicit, we rewrite the loss function as follows. When focusing on $Z_l$ for $l = 1, \ldots, L$, we define

$$\int \cdots \int p(x \mid \phi_{L+1}, z_L) \prod_{l'=l}^{L-1} q_{l'}(z_{l'+1} \mid \phi_{l'+1}, z_{l'}) dz_{l+1} \ldots dz_L =: p_l(x \mid \phi_{l+1:L+1}, z_l)$$

and

$$\int \cdots \int p(z_l \mid \phi_l, z_{l-1}) \prod_{l'=2}^{l-1} q_l(z_{l'} \mid \phi_{l'}, z_{l'-1}) q_l(z_1 \mid \phi_1, x) dz_{l-1} \ldots dz_1 =: q_{1:l-1}(z_l \mid \phi_{1:l}, x).$$

Then, the loss $\ell_0$ can be written as

$$\begin{aligned}
\ell_0(\phi_{1:L+1}, x) &= \mathbb{E}_{\hat{x} \sim p_l(x|\phi_{l+1:L+1}, z_l) q_{1:l-1}(z_l|\phi_{1:l}, x)} \|x - \hat{x}\|^2 \\
&= \mathbb{E}_{q_{1:l-1}(z_l|\phi_{1:l}, x)} \ell_l(\phi_{l+1:L+1}, x, z_l),
\end{aligned}$$

where we define $\ell_l(\phi_{l+1:L+1}, x, z_l) := \mathbb{E}_{\hat{x} \sim p_l(x|\phi_{l+1:L+1}, z_l)} \|x - \hat{x}\|^2$.

With this notation, the correspondence to the proof of Theorem 2 in Appendix C becomes clear. The posterior $q_{1:l-1}(z_l \mid \phi_{1:l}, x)$ corresponds to $q(z \mid \phi, x)$, and $\ell_l(\phi_{l+1:L+1}, x, z_l)$ corresponds to $L(z)$ in Appendix C.1. Since the loss is bounded by $\Delta$, we can apply the truncation and discretization arguments in exactly the same way as in Appendix C.1. In the discretization argument, the Lipschitz property of the posterior distribution, shown in Lemma 3, is required. A similar result holds for $q_{1:l-1}(z_l \mid \phi_{1:l}, x)$ since it is also Gaussian and satisfies the same assumptions.

We define $\hat{Z}$ as the discretized version of $z_l$ (as in Appendix C.1), and there exists a Markov chain $X - Z_l - J - \hat{Z}$, where $J$ denotes the discretization index as defined in Eq. (11).

Next, we analyze the generalization error for the discretized $Z_l$, following Appendix C.2. The proof proceeds analogously to Appendix D.2. We define

$$\mathbb{E}_{q(\hat{Z}|J)} \ell_l(\phi_{l+1:L+1}, x, \hat{Z}) = \int_{C_J} \ell_l(\phi_{l+1:L+1}, x, z) dz = \int_{C_J} \mathbb{E}_{\hat{x} \sim p_l(x|\phi_{l+1:L+1}, z)} \|x - \hat{x}\|^2 dz.$$

Since $q(\hat{Z} \mid J)$ is uniform on $C_J$,

$$\mathbb{E}_{q(\hat{Z}|J)} \ell_l(\phi_{l+1:L+1}, x, \hat{Z}) = \frac{1}{|C_J|} \int_{C_J} \ell_l(\phi_{l+1:L+1}, x, z) \, dz.$$

Set

$$\ell_C(x, \phi_{l+1:L+1}, J) := \frac{1}{|C_J|} \int_{C_J} \ell_l(\phi_{l+1:L+1}, x, z) \, dz.$$

Then $0 \leq \ell_C \leq \Delta$.

We now introduce the LOO-CMI setting. Let $X_U$ denote the test data and $X_U^{n-1}$ the training dataset. To emphasize the dependence of the dataset on $U$, we write the posterior as

$$q(J^n|\phi_{1:l}, X^n) = q(J_U, J_U^{n-1}|\phi_{1:l}, X^n) = q(J_U|\phi_{1:l}, X_U)q(J_U^{n-1}|\phi_{1:l}, X_U^{n-1}).$$

Under the leave-one-out setting, the generalization error in Eq. (10) becomes

$$\mathbb{E}_{U,X^n}\mathbb{E}_{q(\phi_{1:L+1}|X_U^{n-1})}\Big(\mathbb{E}_{q(J_U|\phi_{1:l},X_U)}\ell_C(X_U, \phi_{l+1:L+1}, J_U) - \frac{1}{n-1}\sum_{m=1}^{n-1}\mathbb{E}_{q(J_{U,m}^{n-1}|\phi_{1:l},X_{U,m}^{n-1})}\ell_C(X_{U,m}^{n-1}, \phi_{l+1:L+1}, J_{U,m}^{n-1})\Big)$$

$$= \mathbb{E}_{X^n,U}\sum_{k=1}^{K}\mathbb{E}_{q(J_U|\phi_{1:l},X_U)q(\phi_{1:L+1}|X_U^{n-1})}\ell_C(X_U, \phi_{l+1:L+1}, k)\mathbb{1}_{k=J_U}$$

$$- \mathbb{E}_{X^n,U}\sum_{k=1}^{K}\frac{1}{n-1}\sum_{m=1}^{n-1}\mathbb{E}_{q(J_{U,m}^{n-1}|\phi_{1:l},X_{U,m}^{n-1})q(\phi_{1:L+1}|X_U^{n-1})}\ell_C(X_{U,m}^{n-1}, \phi_{l+1:L+1}, k)\mathbb{1}_{k=J_{U,m}^{n-1}}.$$

We decompose the generalization error as in Eq. (12):

$$= \mathbb{E}_{X^n,U}\sum_{k=1}^{K}\mathbb{E}_{q(J_U|\phi_{1:l},X_U)q(\phi_{1:L+1}|X_U^{n-1})}\ell_C(X_U, \phi_{l+1:L+1}, k)\mathbb{1}_{k=J_U}$$

$$- \mathbb{E}_{X^n,U}\sum_{k=1}^{K}\frac{1}{n-1}\sum_{m=1}^{n-1}\mathbb{E}_{q(J_{U,m}^{n-1}|\phi_{1:l},X_{U,m}^{n-1})q(\phi_{1:L+1}|X_U^{n-1})}\ell_C(X_U, \phi_{l+1:L+1}, k)\mathbb{1}_{k=J_{U,m}^{n-1}}$$

$$+ \mathbb{E}_{X^n,U}\sum_{k=1}^{K}\frac{1}{n-1}\sum_{m=1}^{n-1}\mathbb{E}_{q(J_{U,m}^{n-1}|\phi_{1:l},X_{U,m}^{n-1})q(\phi_{1:L+1}|X_U^{n-1})}\ell_C(X_U, \phi_{l+1:L+1}, k)\mathbb{1}_{k=J_{U,m}^{n-1}}$$

$$- \mathbb{E}_{X^n,U}\sum_{k=1}^{K}\frac{1}{n-1}\sum_{m=1}^{n-1}\mathbb{E}_{q(J_{U,m}^{n-1}|\phi_{1:l},X_{U,m}^{n-1})q(\phi_{1:L+1}|X_U^{n-1})}\ell_C(X_{U,m}^{n-1}, \phi_{l+1:L+1}, k)\mathbb{1}_{k=J_{U,m}^{n-1}}.$$

The first two terms can be bounded exactly as in Eq. (13). The latter two terms can be rewritten as

$$\mathbb{E}_{X^n,U}\sum_{k=1}^{K}\mathbb{E}_{q(J_{U,m}^{n-1}|\phi_{1:l},X_{U,m}^{n-1})q(\phi_{1:L+1}|X_U^{n-1})}\ell_C(X_U, \phi_{l+1:L+1}, k)\mathbb{1}_{k=J_{U,m}^{n-1}}$$

$$- \mathbb{E}_{X^n,U}\sum_{k=1}^{K}\frac{1}{n-1}\sum_{m=1}^{n-1}\mathbb{E}_{q(J_{U,m}^{n-1}|\phi_{1:l},X_{U,m}^{n-1})q(\phi_{1:L+1}|X_U^{n-1})}\ell_C(X_{U,m}^{n-1}, \phi_{l+1:L+1}, k)\mathbb{1}_{k=J_{U,m}^{n-1}}$$

$$= \mathbb{E}_{X^n,U}\frac{1}{n-1}\sum_{m=1}^{n-1}\Big(X_U - X_{U,m}^{n-1}\Big) \cdot \mathbb{E}_{q(J_{U,m}^{n-1}|\phi_{1:l},X_{U,m}^{n-1})q(\phi_{1:L+1}|X_U^{n-1})}\sum_{k=1}^{K}\frac{1}{|C_k|}\int_{C_k}\mathbb{E}_{\hat{x}\sim p_l(x|\phi_{l+1:L+1},z)}[\hat{x}]dz\,\mathbb{1}_{k=J_{U,m}^{n-1}}.$$

We can evaluate this expression in the same manner as Eq. (15).

**Remark 1.** *Note that the mean of the stochastic decoder $\mathbb{E}_{\hat{x}\sim p_l(x|\phi_{l+1:L+1},z)}[\hat{x}]$ corresponds to $g_\theta(z)$ in Appendix D.2. Therefore, if we regard the mean of the stochastic decoder as the deterministic decoder, the analysis proceeds identically.*

After establishing the generalization error for the discretized $Z_l$, we apply the data-processing inequality to remove the effects of truncation and discretization from the information-theoretic quantities, as in Appendix C.3. By taking the limit with respect to the truncation and discretization levels (Appendix C.4), we obtain the desired result.

## H.2. Discussion of the bound in Theorem 3

As discussed in the proof above, the result of Theorem 3 is stated for stochastic decoders, but it also holds for deterministic ones. When a stochastic decoder produces reconstructed data outside the domain $\mathcal{X}$, the upper bound of its variance ($\sigma_0^2$) leads to a reconstruction loss bounded within $\sigma_0^2 + \Delta$, which only changes the constant factor in the theorem. The proof strategy, however, remains the same. Although Theorem 3 assumes a Gaussian stochastic decoder, the Gaussianity itself is not crucially used in the proof—it is adopted merely for notational consistency with the Gaussian stochastic layer. Therefore, similar results also hold for other stochastic decoders, such as those based on Bernoulli distributions.

## H.3. Additional Explanation of the Graphical Models in Figure 2

We further clarify the distinction between the two graphical models in Figure 2. The important point is that Figure 2a represents the standard LOO-CMI construction, where the index $U$ specifies the train/test split for the entire prediction block. In that construction, all data $X^n$ are fed into the encoder/prediction mechanism to form the reconstructed outputs $\hat{X}^n$, while $U$ selects which sample is held out for the loss. Consequently, $X^n$ has a direct graphical dependence on $\hat{X}^n$ in Figure 2a.

In contrast, Figure 2b illustrates the latent-variable refinement used in our layer-wise analysis. The index $U$ is not used to shuffle the whole prediction block; instead, it is used to define the leave-one-out latent-variable channel. This distinction is most transparent by comparing the posterior and prior distributions that enter the Donsker–Varadhan representation. For simplicity, consider the single-latent-layer case. In the standard block-level construction, the posterior can be written schematically as

$$Q_{\text{std}}^{\text{post}} = P(X^n)p(U)\,q(\phi,\theta \mid X_U^{n-1})q(Z^n \mid \phi, X_U, X_U^{n-1})p(\hat{X}^n \mid \theta, Z^n),$$

where changing $U$ changes both the training set used for parameter learning and the input assignment used to construct the prediction block. Its corresponding prior marginalizes this entire prediction-side dependence on $U$:

$$Q_{\text{std}}^{\text{prior}} = P(X^n)p(U)\,q_{\text{std}}(\hat{X}^n \mid X^n),$$
$$q_{\text{std}}(\hat{X}^n \mid X^n) = \mathbb{E}_{p(U)q(\phi,\theta\mid X_U^{n-1})q(Z^n\mid\phi,X_U,X_U^{n-1})}\big[p(\hat{X}^n \mid \theta, Z^n)\big].$$

Our proposed construction separates these two effects. The posterior has the same learned parameters but isolates the latent-variable channel,

$$Q_{\text{LV}}^{\text{post}} = P(X^n)p(U)\,q(\phi,\theta \mid X_U^{n-1})q(Z^n \mid \phi, X_U, X_U^{n-1})p(\hat{X}^n \mid \theta, Z^n),$$

whereas the prior used for the latent term replaces only the latent-variable channel by its $U$-marginalized version,

$$Q_{\text{LV}}^{\text{prior}} = P(X^n)p(U)\,q(\phi,\theta \mid X_U^{n-1})q(Z^n \mid \phi, X^n)p(\hat{X}^n \mid \theta, Z^n),$$
$$q(Z^n \mid \phi, X^n) = \mathbb{E}_{p(U)}\big[q(Z^n \mid \phi, X_U, X_U^{n-1})\big].$$

Thus, the marginalization over $U$ is performed only on the hidden-variable side $Z^n$, not on the parameter-learning block $q(\phi,\theta \mid X_U^{n-1})$. This is why Figure 2b contains the direct dependence $U \to Z_1^n$: the held-out index controls which latent variables are compared in the LOO-CMI term. For hierarchical VAEs, the same reasoning applies layer-wise to $Z_l^n$, yielding the terms $I(Z_l^n; U \mid X^n, \phi_{1:l})$ in Theorem 3.

# I. Extension to Fraction setting and uniform convergence

## I.1. Extension to fraction dataset

In the main text, we analyzed the generalization behavior under the leave-one-out (LOO) setting. In this section, we extend this idea to a more general test split with ratio $c \in (0, 1)$.

Specifically, given the full dataset $X^n = (X_1, \ldots, X_n)$, we randomly select

$$|U| = cn \qquad (c \in (0, 1))$$

samples as a test set, where $U \subset [n]$, and use the remaining samples indexed by $U^c$ as the training set. For simplicity, we assume that $cn$ is an integer. The random index set $U$ is sampled *without replacement*, i.e., each subset $U \subset [n]$ with $|U| = cn$ is selected uniformly with probability $1/\binom{n}{cn}$.

Apart from this modification of the test split, the VAE model and all assumptions follow Section 2 and Assumption 3.1. In the main paper, the LOO setting corresponds to $|U| = 1$, and the model parameters are learned from the training set $S := X^n \setminus X_U$ of size $n - 1$. In contrast, under the present setting, we train the model using

$$X_{U^c} := \{X_i \mid i \in U^c\},$$

and the test data are denoted by

$$X_U := \{X_i \mid i \in U\}.$$

We refer to this setup as the *fraction-c setting*.

We consider the following generalization error:

$$\text{gen}_c(n, \mathcal{D}) := \left| \mathbb{E}_{X^n, U} \mathbb{E}_{q(W|S)} \frac{1}{|U|} \sum_{i \in U} \ell_0(W, X_i) - \frac{1}{|U^c|} \sum_{i \notin U} \ell_0(W, X_i) \right|.$$

The first term corresponds to the test error, while the second term corresponds to the training error. This definition naturally extends the generalization gap used in the main paper: the leave-one-out setting is recovered by choosing $c = 1/n$.

**Theorem 6.** *Under Assumptions 1 and 2, and under the fraction-c setting with $c \in (0, 1)$, we have*

$$\text{gen}_c(n, \mathcal{D}) \leq 3\sqrt{2}\Delta \sqrt{\frac{(I(\phi; U \mid X^n) + I(Z^n; U \mid \phi, X^n))}{nc(1-c)^2}} + \frac{\Delta}{\sqrt{cn}} + \frac{\Delta}{\sqrt{(1-c)n}}.$$

**Remark 2.** *An important feature of the bound is its explicit dependence on the test fraction c. This allows us to investigate which choice of c is optimal. When the conditional mutual information (CMI) is sufficiently large (i.e., in the limit as it tends to infinity), the bound is minimized at $c = 1/3$. In contrast, when the CMI is sufficiently small (i.e., in the limit as it tends to zero), the optimal choice becomes $c = 1/2$.*

*As discussed in Appendix I.2, the CMI typically scales on the order of more than $\log n$ with respect to the sample size $n$. In such cases, the CMI is expected to grow slowly as $n$ increases. Consequently, the bound corresponding to $c = 1/3$ is preferable in this regime. Formal proofs of these claims are provided below.*

An important implication of this bound is that the convergence rate depends explicitly on the effective test size $cn$. For the optimal choices of $c$ discussed above, the dominant term of the bound behaves as $\mathcal{O}\left(\sqrt{\text{LOO-CMI terms}/n}\right)$. Taking into account that, as discussed in Appendix I.2, the CMI typically scales on the order of $\log n$, this rate can be regarded as a moderate convergence behavior.

In contrast, the leave-one-out CMI (LOO-CMI) setting considered in the main paper corresponds to $c = 1/n$, since only a single sample is used as test data. In this case, the dominant term of the bound scales as $\mathcal{O}\left(\sqrt{\text{LOO-CMI terms}}\right)$, and the explicit dependence on $n$ disappears. While this is a drawback in terms of convergence with respect to the sample size, the LOO-CMI setting offers complementary advantages. As discussed in the main paper, (i) it enables a more refined analysis of the parameter learning dynamics, including connections to flat minima, and (ii) since the index $U$ is one-dimensional, the resulting quantities are numerically more tractable.

Taken together, these considerations highlight that the choice of $c$ should be made by balancing convergence rates with these additional analytical and computational advantages. Selecting an appropriate $c$ in this manner is therefore crucial for developing meaningful theoretical guarantees.

*Proof of Theorem 6.* The proof largely follows that of Theorem 2, and the arguments in Appendix C apply with minor modifications. That is, the truncation step, discretization procedure, evaluation of the loss difference, and the subsequent limiting argument are identical to those used in the proof of Theorem 2.

The main difference arises in the evaluation of the latent-variable contribution in the fraction-c setting. We first follow the same bounded-loss reduction as in Theorem 5, and then use the data-processing upper bound to express the final result in terms of the continuous latent CMI $I(Z^n; U \mid \phi, X^n)$, as in the main theorem.

**Theorem 7.** *Under Assumptions 1 and 2, and under the fraction-$c$ setting with $c \in (0, 1)$, we have*

$$\text{gen}_c(n, \mathcal{D}) \leq 3\sqrt{2}\Delta\sqrt{\frac{(I(\phi; U \mid X^n) + I(Z^n; U \mid \phi, X^n))}{nc(1-c)^2}} + \frac{\Delta}{\sqrt{cn}} + \frac{\Delta}{\sqrt{(1-c)n}}.$$

The proof of this proceeds by rewriting the leave-one-out argument in Appendix D under the fraction-$c$ setting. In particular, the posterior and prior distributions used in the Donsker–Varadhan argument (cf. Eq. (13)) are modified as follows.

First, the conditional distribution of the latent variables is defined as

$$q(J^n \mid \phi, X^n) = q(J_U, J_{U^c} \mid \phi, X_U, X_{U^c}) = q(J_U \mid \phi, X_U)\, q(J_{U^c} \mid \phi, X_{U^c}).$$

Let $X_{U_m^c}$ denote the $m$-th training sample with the corresponding latent variable $J_{U_m^c}$. Similarly, let $X_{U_m}$ and $J_{U_m}$ denote the $m$-th test sample and its corresponding latent variable. We define the posterior and prior distributions as

$$\begin{aligned}
\mathbf{Q} &:= P(U)\, q(\phi, \theta \mid X_{U^c})\, q(J^n \mid \phi, X_U, X_{U^c}), \\
\mathbf{P} &:= P(U)\, q(\phi, \theta \mid X_{U^c})\, \mathbb{E}_{P(U')}\big[q(J^n \mid \phi, X_{U'}, X_{U'^c})\big].
\end{aligned} \tag{27}$$

With these definitions, the proof based on the Donsker–Varadhan variational formula in Appendix D carries over verbatim by replacing the posterior and prior distributions with those defined in Eq. (27). As in Eq. (12), we begin by applying the following decomposition:

$$\begin{aligned}
\text{gen}(n, \mathcal{D}) = & \mathbb{E}_{X^n, U} \sum_{k=1}^{K} \frac{1}{|U|} \sum_{m \in U} \mathbb{E}_{q(J_{U_m} \mid \phi, X_{U_m}) q(\phi, \theta \mid X_{U^c})} \ell_C(X_{U_m}, \theta, k) \mathbb{1}_{k=J_{U_m}} \\
& - \mathbb{E}_{X^n, U} \sum_{k=1}^{K} \frac{1}{|U^c|} \sum_{m \notin U} \mathbb{E}_{q(J_{U_m^c} \mid \phi, X_{U_m^c}) q(\phi, \theta \mid X_{U^c})} \left( \frac{1}{|U|} \sum_{m' \in U} \ell_C(X_{U_{m'}}, \theta, k) \right) \mathbb{1}_{k=J_{U_m^c}} \\
& + \mathbb{E}_{X^n, U} \sum_{k=1}^{K} \frac{1}{|U^c|} \sum_{m \notin U} \mathbb{E}_{q(J_{U_m^c} \mid \phi, X_{U_m^c}) q(\phi, \theta \mid X_{U^c})} \left( \frac{1}{|U|} \sum_{m' \in U} \ell_C(X_{U_{m'}}, \theta, k) \right) \mathbb{1}_{k=J_{U_m^c}} \\
& - \mathbb{E}_{X^n, U} \sum_{k=1}^{K} \frac{1}{|U^c|} \sum_{m \notin U} \mathbb{E}_{q(J_{U_m^c} \mid \phi, X_{U_m^c}) q(\phi, \theta \mid X_{U^c})} \ell_C(X_{U_m^c}, \theta, k) \mathbb{1}_{k=J_{U_m^c}}.
\end{aligned} \tag{28}$$

For notational convenience, we define

$$\bar{\ell}_C(X_U, \theta, k) := \frac{1}{|U|} \sum_{m' \in U} \ell_C(X_{U_{m'}}, \theta, k).$$

We first evaluate the first and second terms. Applying the Donsker–Varadhan variational formula, we obtain the following inequality, which replaces Eq. (14):

$$\begin{aligned}
\mathbb{E}_{X^n, U} \sum_{k=1}^{K} \mathbb{E}_{q(\phi, \theta \mid X_{U^c})} & \bigg( \frac{1}{|U|} \sum_{m \in U} \mathbb{E}_{q(J_{U_m} \mid \phi, X_{U_m})} \ell_C(X_{U_m}, \theta, k) \mathbb{1}_{k=J_{U_m}} \\
& - \frac{1}{|U^c|} \sum_{m \notin U} \mathbb{E}_{q(J_{U_m^c} \mid \phi, X_{U_m^c})} \bar{\ell}_C(X_U, \theta, k) \mathbb{1}_{k=J_{U_m^c}} \bigg) \\
\leq \mathbb{E}_X \frac{1}{\lambda} \text{KL}(\mathbf{Q}|\mathbf{P}) & + \mathbb{E}_X \frac{1}{\lambda} \log \mathbb{E}_{\mathbf{P}} \exp \times \\
& \left( \lambda \sum_{k=1}^{K} \left( \frac{1}{|U|} \sum_{m \in U} \ell_C(X_{U_m}, \theta, k) \mathbb{1}_{k=J_{U_m}} - \frac{1}{|U^c|} \sum_{m \notin U} \bar{\ell}_C(X_U, \theta, k) \mathbb{1}_{k=J_{U_m^c}} \right) \right).
\end{aligned} \tag{29}$$

The exponential moment term is defined as

$$\Omega := \mathbb{E}_{X,\mathbf{P}} \sum_{k=1}^{K} \left( \frac{1}{|U|} \sum_{m \in U} \ell_C(X_{U_m}, \theta, k) \mathbb{1}_{k=J_{U_m}} - \frac{1}{|U^c|} \sum_{m \notin U} \bar{\ell}_C(X_U, \theta, k) \mathbb{1}_{k=J_{U_m^c}} \right).$$

By the definition of sampling without replacement for $U'$, the quantity

$$\mathbb{E}_{\mathbb{E}_{P(U')} q(J^n | \phi, X_{U'}, X_{U'^c})} \mathbb{1}_{k=J_{U_m}}$$

does not depend on the index $m$. We denote this common value by $P_k$. This follows from the fact that, under sampling without replacement, each sample $X_m$ for $m = 1, \ldots, n$ has an equal probability of appearing in the expectation.

Therefore, we obtain

$$\Omega = \mathbb{E}_{X,\theta,U} \sum_{k=1}^{K} \left( \frac{1}{|U|} \sum_{m \in U} \ell_C(X_{U_m}, \theta, k) P_k - \frac{1}{|U^c|} \sum_{m \notin U} \bar{\ell}_C(X_U, \theta, k) P_k \right)$$

$$= \mathbb{E}_{X,\theta,U} \sum_{k=1}^{K} \left( \frac{1}{|U|} \sum_{m \in U} \ell_C(X_{U_m}, \theta, k) P_k - \frac{1}{|U|} \sum_{m \in U} \ell_C(X_{U_m}, \theta, k) P_k \right) = 0.$$

Thus, the exponential term in Eq. (29) vanishes. Note that $\underset{P(U')}{\mathbb{E}} q(J^n | \phi, X_{U'}, X_{U'^c})$ is symmetric with respect to the choice of $U$. Therefore, we obtain

$$\text{Eq. (29)} \le \mathbb{E}_X \frac{1}{\lambda} \text{KL}(\mathbf{Q}\|\mathbf{P}) + \mathbb{E}_X \frac{1}{\lambda} \log \mathbb{E}_{\mathbf{P}P(U'')} \exp \left( \lambda \sum_{k=1}^{K} \left( \frac{1}{|U''|} \sum_{m \in U''} \ell_C(X_{U_m''}, \theta, k) \mathbb{1}_{k=J_{U_m''}} \right. \right.$$

$$\left. \left. - \frac{1}{|U''^c|} \sum_{m \notin U''} \bar{\ell}_C(X_{U''}, \theta, k) \mathbb{1}_{k=J_{U_m''^c}} \right) \right), \tag{30}$$

where the auxiliary random variable $U'' = \{U_1'', \ldots, U_{cn}''\}$ is $cn$-dimensional. Since $U''$ is obtained by random sampling without replacement, its components are not mutually independent. As a consequence, standard concentration inequalities applicable to independent random variables—such as those used in the supersample setting—cannot be applied directly.

To address this issue, we invoke the results of Joag-Dev & Proschan (1983) concerning negative association for sampling without replacement. In particular, by Section 3.2(a) of Joag-Dev & Proschan (1983), the distribution $P(U)$ satisfies negative association. Exploiting this property, we obtain

$$\text{Eq. (30)} \le \mathbb{E}_X \frac{1}{\lambda} \text{KL}(\mathbf{Q}\|\mathbf{P}) + \mathbb{E}_X \frac{1}{\lambda} \log \mathbb{E}_{\mathbf{P} \prod_{m'=1}^{cn} P(U_{m'}'')} \exp \left( \lambda \sum_{k=1}^{K} \left( \frac{1}{|U''|} \sum_{m \in U''} \ell_C(X_{U_m''}, \theta, k) \mathbb{1}_{k=J_{U_m''}} \right. \right.$$

$$\left. \left. - \frac{1}{|U''^c|} \sum_{m \notin U''} \bar{\ell}_C(X_{U''}, \theta, k) \mathbb{1}_{k=J_{U_m''^c}} \right) \right),$$

where $P(U_{m'}'')$ denotes the marginal distribution of $U_{m'}''$. This step effectively replaces the dependent sampling without replacement by $cn$ independent random variables with the same marginals using the property of the negative association.

Since the random variables $\{U_m''\}_{m=1}^{cn}$ are now independent, we may apply McDiarmid's inequality. This yields

$$\mathbb{E}_X \frac{1}{\lambda} \log \mathbb{E}_{\mathbf{P} \prod_{m'=1}^{cn} P(U_{m'}'')} \exp \left( \lambda \sum_{k=1}^{K} \left( \frac{1}{|U''|} \sum_{m \in U''} \ell_C(X_{U_m''}, \theta, k) \mathbb{1}_{k=J_{U_m''}} - \frac{1}{|U''^c|} \sum_{m \notin U''} \bar{\ell}_C(X_{U''}, \theta, k) \mathbb{1}_{k=J_{U_m''^c}} \right) \right)$$

$$\le \frac{\lambda^2 \Delta^2}{2nc(1-c)^2}.$$

This bound is derived using McDiarmid's inequality.

Note that there are $cn$ random variables involved. We therefore evaluate the corresponding bounded-difference constants as follows.

Let $c_i$ denote the maximum change in the exponent when a single index $U_m$ is replaced. Its magnitude can be bounded as

$$c_i \le \frac{2\lambda\Delta}{|U|} + \frac{2\lambda\Delta}{|U^c|} = \frac{2\lambda\Delta}{cn} + \frac{2\lambda\Delta}{(1-c)n} = \frac{2\lambda\Delta}{n}\left(\frac{1}{c} + \frac{1}{1-c}\right),$$

where we used the boundedness of the loss.

Consequently, the sum of squared bounded differences appearing in McDiarmid's inequality satisfies

$$\sum_{i=1}^{cn} c_i^2 \le cn \cdot \left(\frac{2\lambda\Delta}{n}\left(\frac{1}{c} + \frac{1}{1-c}\right)\right)^2 = \frac{4c\lambda^2\Delta^2}{n}\left(\frac{1}{c} + \frac{1}{1-c}\right)^2 \tag{31}$$

$$= \frac{4\lambda^2\Delta^2}{nc(1-c)^2}.$$

Next, we evaluate the third and fourth terms in Eq. (28). Using an argument analogous to that of Eq. (15), we obtain

$$\mathbb{E}_{X^n,U} \sum_{k=1}^{K} \frac{1}{|U^c|} \sum_{m\notin U} \mathbb{E}_{q(J_{U_m^c}|\phi,X_{U_m^c})q(\phi,\theta|X_{U^c})} \left(\frac{1}{|U|}\sum_{m'\in U} \ell_C(X_{U_{m'}},\theta,k)\right) \mathbb{1}_{k=J_{U_m^c}}$$

$$- \mathbb{E}_{X^n,U}\sum_{k=1}^{K}\frac{1}{|U^c|}\sum_{m\notin U} \mathbb{E}_{q(J_{U_m^c}|\phi,X_{U_m^c})q(\phi,\theta|X_{U^c})}\ell_C(X_{U_m^c},\theta,k)\mathbb{1}_{k=J_{U_m^c}}$$

$$= \mathbb{E}_{X^n,U}\frac{2}{|U^c|}\sum_{m\notin U}\left(\frac{1}{|U|}\sum_{m'\in U}X_{U_{m'}} - X_{U_m^c}\right)\cdot \mathbb{E}_{q(J_{U_m^c}|\phi,X_{U_m^c})q(\phi,\theta|X_{U^c})}\sum_{k=1}^{K}\bar{g}_{\theta,k}\mathbb{1}_{k=J_{U_m^c}}$$

$$\le \mathbb{E}_X\frac{1}{\lambda}\mathrm{KL}(\mathbf{Q}\|\mathbf{P}) + \mathbb{E}_X\frac{1}{\lambda}\log\mathbb{E}_\mathbf{P}\exp\left(\frac{2\lambda}{|U^c|}\sum_{m\notin U}\left(\frac{1}{|U|}\sum_{m'\in U}X_{U_{m'}} - X_{U_m^c}\right)\cdot\sum_{k=1}^{K}\bar{g}_{\theta,k}\mathbb{1}_{k=J_{U_m^c}}\right)$$

$$\le \mathbb{E}_X\frac{1}{\lambda}\mathrm{KL}(\mathbf{Q}\|\mathbf{P}) + \mathbb{E}_X\frac{1}{\lambda}\log\mathbb{E}_{P(U)q(\phi,\theta|X_{U^c})\underset{P(U')}{\mathbb{E}}q(J^n|\phi,X_U,X_{U^c})}\mathbb{E}_{P(U'')}\exp\left(\frac{2\lambda}{|U''^c|}\sum_{m\notin U''}\right.$$

$$\left.\left(\frac{1}{|U''|}\sum_{m'\in U''}X_{U''_{m'}} - X_{U''_m{}^c}\right)\cdot\sum_{k=1}^{K}\bar{g}_{\theta,k}\mathbb{1}_{k=J_{U''_m{}^c}}\right), \tag{32}$$

We first evaluate the expectation of the exponential moment. Define

$$\Omega := \mathbb{E}_{P(X^n)P(U)q(\phi,\theta|X_{U^c})}\frac{2}{|U^c|}\sum_{m\notin U}\left(\frac{1}{|U|}\sum_{m'\in U}X_{U_{m'}} - X_{U_m^c}\right)\cdot\sum_{k=1}^{K}\bar{g}_{\theta,k}\underbrace{\underset{\underset{P(U')}{\mathbb{E}}}{\mathbb{E}}q(J^n|\phi,X_{U'},X_{U'^c})\mathbb{1}_{k=J_{U_m^c}}}_{=: P_k}.$$

By symmetry of sampling without replacement, the quantity $\mathbb{E}_{\underset{P(U')}{\mathbb{E}}}q(J^n|\phi,X_{U'},X_{U'^c})\mathbb{1}_{k=J_{U_m^c}}$ does not depend on the index $m$, and we denote it by $P_k$. Using this notation, we obtain

$$\mathbb{E}_{P(X^n)P(U)q(\phi,\theta|X_{U^c})}\frac{1}{|U^c|}\sum_{m\notin U}\left(\frac{1}{|U|}\sum_{m'\in U}X_{U_{m'}} - X_{U_m^c}\right)\cdot\sum_{k=1}^{K}\bar{g}_{\theta,k}P_k$$

$$= \mathbb{E}_{P(X^n)P(U)}\left(\frac{1}{|U|}\sum_{m'\in U}X_{U_{m'}} - \frac{1}{|U^c|}\sum_{m\notin U}X_{U_m^c}\right)\cdot\mathbb{E}_{q(\phi,\theta|X_{U^c})}\sum_{k=1}^{K}\bar{g}_{\theta,k}P_k$$

$$\le \sqrt{\mathbb{E}_{P(U)P(X)}\left\|\frac{1}{|U|}\sum_{m'\in U}X_{U_{m'}} - \mathbb{E}_X X\right\|^2}\sqrt{\Delta} + \sqrt{\mathbb{E}_{P(U)P(X)}\left\|\frac{1}{|U^c|}\sum_{m\notin U}X_{U_m^c} - \mathbb{E}_X X\right\|^2}\sqrt{\Delta}.$$

We bound the first factor exactly as in Eq. (26), i.e., by the variance of a bounded random variable, which yields

$$\sqrt{\mathbb{E}_{P(U)P(X)}\left\|\frac{1}{|U|}\sum_{m'\in U}X_{U_{m'}}-\mathbb{E}_X X\right\|^2}\sqrt{\Delta}\leq\sqrt{\frac{\Delta}{4cn}}.$$

Consequently,

$$\Omega\leq\frac{\Delta}{\sqrt{cn}}+\frac{\Delta}{\sqrt{(1-c)n}}.$$

Substituting this bound into the exponential moment in Eq. (32), we obtain

$$\mathbb{E}_X\frac{1}{\lambda}\mathrm{KL}(\mathbf{Q}\|\mathbf{P})+\mathbb{E}_X\frac{1}{\lambda}\log\mathbb{E}_{P(U)q(\phi,\theta|X_{U^c})}\mathop{\mathbb{E}}_{P(U')}q(J^n|\phi,X_U,X_{U^c})\mathbb{E}_{P(U'')}$$

$$\exp\left(\frac{2\lambda}{|U''^c|}\sum_{m\notin U''}\left(\frac{1}{|U''|}\sum_{m'\in U''}X_{U''_{m'}}-X_{U''_m^c}\right)\cdot\sum_{k=1}^K\bar{g}_{\theta,k}\mathbb{1}_{k=J_{U''_m^c}}-\lambda\Omega\right)+\Omega.$$

We then apply McDiarmid's inequality. The exponential moment is upper bounded by the bounded-difference estimate in Eq. (31), yielding

$$\frac{1}{8}\left(\frac{4\lambda\Delta}{n}\left(\frac{1}{c}+\frac{1}{1-c}\right)\right)^2=\frac{2\lambda^2\Delta^2}{nc(1-c)^2}.$$

Summarizing the above estimates, we conclude that

$$\mathrm{gen}(n,\mathcal{D})\leq\mathbb{E}_X\frac{2}{\lambda}\mathrm{KL}(\mathbf{Q}\|\mathbf{P})+\frac{5\lambda\Delta^2}{2nc(1-c)^2}+\frac{\Delta}{\sqrt{cn}}+\frac{\Delta}{\sqrt{(1-c)n}}.$$

We can further tighten this bound in the same manner as in Eq. (18). In the above derivation, the Donsker–Varadhan variational formula is applied twice. Instead, applying it only once as in Eq. (18), the bounded difference becomes

$$\frac{1}{8}\left(\frac{6\lambda\Delta}{n}\left(\frac{1}{c}+\frac{1}{1-c}\right)\right)^2=\frac{9\lambda^2\Delta^2}{2nc(1-c)^2}.$$

This leads to

$$\mathrm{gen}(n,\mathcal{D})\leq\mathbb{E}_X\frac{1}{\lambda}\mathrm{KL}(\mathbf{Q}\|\mathbf{P})+\frac{9\lambda\Delta^2}{2nc(1-c)^2}+\frac{\Delta}{\sqrt{cn}}+\frac{\Delta}{\sqrt{(1-c)n}}.$$

Optimizing over $\lambda$ yields

$$\mathrm{gen}(n,\mathcal{D})\leq 3\sqrt{2}\Delta\mathbb{E}_X\sqrt{\frac{\mathrm{KL}(\mathbf{Q}\|\mathbf{P})}{nc(1-c)^2}}+\frac{\Delta}{\sqrt{cn}}+\frac{\Delta}{\sqrt{(1-c)n}}.$$

Finally, by upper bounding the KL divergence in terms of the conditional mutual information, we obtain the desired bound. □

Next, we provide the proof of Remark 2

*Proof.* Consider the function

$$F(c):=\frac{ab}{\sqrt{c(1-c)^2}}+\frac{b}{\sqrt{c}}+\frac{b}{\sqrt{1-c}},\qquad c\in(0,1),$$

where $a, b > 0$ are fixed constants. Using $\sqrt{c(1-c)^2} = \sqrt{c}(1-c)$, we can rewrite $F(c)$ as

$$F(c) = b\left(\frac{a}{\sqrt{c}(1-c)} + \frac{1}{\sqrt{c}} + \frac{1}{\sqrt{1-c}}\right) =: b\,g(c).$$

Hence, the minimizer of $F(c)$ over $c \in (0,1)$ does not depend on $b$ and is characterized by the minimization of $g(c)$.

The derivative $g'(c)$ is given by

$$g'(c) = -\frac{a}{2c^{3/2}(1-c)} + \frac{a}{\sqrt{c}(1-c)^2} - \frac{1}{2c^{3/2}} + \frac{1}{2(1-c)^{3/2}}.$$

After multiplying by $2c^{3/2}(1-c)^2\sqrt{1-c}$, the stationary condition $g'(c) = 0$ is equivalent to

$$a(-1 + 3c) - (1-c)^2 + c^{3/2}\sqrt{1-c} = 0. \tag{33}$$

Since $g(c) \to \infty$ as $c \to 0$ or $c \to 1$, the solution $c^\star$ of (33) lies in the interior $(0,1)$ and gives the unique minimizer.

The behavior of the minimizer $c^\star$ can be characterized as follows.

- (*Large-$a$ regime*) When $a \to \infty$, the dominant term in $g(c)$ is $a/(\sqrt{c}(1-c))$, whose unique minimizer is $c = 1/3$. Consequently, $c^\star \longrightarrow \frac{1}{3}$ as $a \to \infty$.

- (*Vanishing-$a$ regime*) When $a = 0$, the objective reduces to $g(c) = 1/\sqrt{c} + 1/\sqrt{1-c}$, which is symmetric around $c = 1/2$ and minimized at $c^\star = \frac{1}{2}$.

- (*Small-$a$ expansion*) For sufficiently small $a$, a first-order expansion around $c = 1/2$ yields

$$c^\star = \frac{1}{2} - \frac{a}{3} + \mathcal{O}(a^2), \qquad a \to 0.$$

In summary, as $a$ increases, the minimizer $c^\star$ continuously shifts from $1/2$ toward $1/3$, and satisfies $\frac{1}{3} \le c^\star \le \frac{1}{2}$. $\qquad\square$

While our main focus is a computable LOO-based diagnostic, one can extend the framework to withholding a fraction $cn$ of samples as test points, which yields an explicit decay with $n$ via a modified bounded-difference argument. However, this extension leads to high-dimensional test-index variables and makes the corresponding information terms substantially harder to estimate in continuous latent spaces. We therefore focus on the LOO setting for numerical tractability, and discuss the $c$-fraction variant as a theoretical extension.

## I.2. Discussion about the sample complexity of VAE

In this section, we discuss the convergence rate of the generalization bound in the extended fractional setting, as stated in Theorem 6. To this end, we need to evaluate the two conditional mutual information (CMI) terms appearing in the bound.

### I.2.1. THE DEGREE OF OVERFITTING FOR THE ENCODER

We begin with the second term, $I(\phi; U \mid X^n)$. As discussed in prior work such as Haghifam et al. (2022), this term can be upper-bounded via the data processing inequality. Denoting the training dataset by $S$, we obtain

$$I(\phi; U \mid X^n) \le I(\phi; S).$$

This quantity corresponds to the mutual information between the learned parameters and the training data, which has been extensively studied in the information-theoretic analysis of learning algorithms.

For instance, Xu & Raginsky (2017) analyzed this term in settings where the parameter space is discrete or can be controlled via covering numbers. In typical finite-dimensional parametric models, they showed that this mutual information scales on the order of $\log n$. Moreover, when the parameters are sampled from a Gibbs posterior induced by a bounded loss function, the mutual information can be further reduced to the order of $1/n$.

Beyond such static analyses, information-theoretic bounds on $I(\phi; S)$ have also been widely investigated from an algorithmic perspective. Pensia et al. (2018) first established a connection between noisy iterative algorithms and mutual information, deriving bounds that depend explicitly on the loss function, the injected noise, and the number of algorithmic iterations. Building on this framework, Wang et al. (2021) and Wang et al. (2023) studied the parameter mutual information of stochastic gradient Langevin dynamics (SGLD), while Futami & Fujisawa (2023) analyzed the same quantity in the continuous-time limit. For standard stochastic gradient descent (SGD), Neu et al. (2021) were the first to investigate parameter mutual information, and Wang & Mao (2022) subsequently improved the dependence of these bounds on the step size. Furthermore, Haghifam et al. (2023) established formal limitations of such information-theoretic approaches in the context of stochastic convex optimization.

In a related but distinct Bayesian setting, where the training dataset is assumed to be conditionally i.i.d. (see Clarke & Barron (1994) for a formal formulation), Clarke & Barron (1994) (see also Rissanen (2006); Haussler & Opper (1997)) showed that the mutual information between the learned parameter and the training dataset admits an explicit asymptotic characterization.

In all of the above scenarios, by appropriately tuning the algorithmic hyperparameters, one can ensure that, in the fractional setting where $c$ does not depend on $n$,

$$\sqrt{\frac{I(\phi; U \mid X^n)}{n\,c(1-c)^2}} \leq \sqrt{\frac{I(\phi; S)}{n\,c(1-c)^2}} \tag{34}$$

and this can be small as we increase $n$.

### I.2.2. THE COMPLEXITY OF LV

We next bound the latent-variable conditional mutual information $I(Z^n; U \mid \phi, X^n)$ by an $n$-point $\sqrt{\mathrm{KL}}$-cover of the Gaussian encoder posterior kernels. In this subsection, all covering numbers are external covering numbers: the cover elements are auxiliary Gaussian kernels used only for the reference-measure argument, and they need not be outputs of the learning algorithm.

Let $x^n = (x_1, \ldots, x_n)$ be a fixed realization of $X^n$. For an encoder parameter value $\varphi$, write

$$q_\varphi(\cdot \mid x) = \mathcal{N}\big(\mu_\varphi(x), \mathrm{diag}(\sigma_\varphi^2(x))\big)$$

for the encoder posterior kernel. For two encoder kernels $q_\varphi$ and $q_\psi$, define the empirical $\sqrt{\mathrm{KL}}$-distance on $x^n$ by

$$d_{\sqrt{\mathrm{KL}},x^n}(\varphi, \psi) := \max_{i \in [n]} \sqrt{\mathrm{KL}(q_\varphi(\cdot \mid x_i) \,\|\, q_\psi(\cdot \mid x_i))}.$$

Let

$$\mathcal{Q} := \{q_\varphi(\cdot \mid \cdot) : \varphi \in \Phi\}$$

be the encoder posterior family. The worst-case $n$-point external $\sqrt{\mathrm{KL}}$-covering number is defined by

$$\mathcal{N}_{\sqrt{\mathrm{KL}},n}(\eta, \mathcal{Q}) := \sup_{x^n \in \mathcal{X}^n} \mathcal{N}\Big(\eta, \mathcal{Q}, d_{\sqrt{\mathrm{KL}},x^n}\Big),$$

where $\mathcal{N}(\eta, \mathcal{Q}, d_{\sqrt{\mathrm{KL}},x^n})$ denotes the smallest number of auxiliary Gaussian kernels needed to $\eta$-cover $\mathcal{Q}$ on the fixed points $x_1, \ldots, x_n$. Equivalently, for each fixed $x^n$, there exists a finite collection of auxiliary Gaussian kernels

$$\widetilde{\mathcal{Q}}_\eta(x^n) = \{\widetilde{q}_1, \ldots, \widetilde{q}_{N_\eta(x^n)}\}, \qquad N_\eta(x^n) = \mathcal{N}(\eta, \mathcal{Q}, d_{\sqrt{\mathrm{KL}},x^n}),$$

such that, for every $\varphi \in \Phi$, there is an index $a_\eta(\varphi; x^n) \in \{1, \ldots, N_\eta(x^n)\}$ satisfying

$$\max_{i \in [n]} \sqrt{\mathrm{KL}\big(q_\varphi(\cdot \mid x_i) \,\big\|\, \widetilde{q}_{a_\eta(\varphi; x^n)}(\cdot \mid x_i)\big)} \leq \eta. \tag{35}$$

For fixed $(\varphi, x^n)$, define the product encoder-posterior law of the continuous latent vector $Z^n = (Z_1, \ldots, Z_n)$ in the original sample order by

$$\mathsf{P}_{\varphi,x^n}^{Z^n}(dz_{1:n}) := \prod_{i=1}^{n} q_\varphi(dz_i \mid x_i).$$

Similarly, for each cover element $\widetilde{q}_a$, define the corresponding product law

$$\widetilde{\mathsf{P}}_{a,x^n}^{Z^n}(dz_{1:n}) := \prod_{i=1}^{n} \widetilde{q}_a(dz_i \mid x_i).$$

The mixture reference measure used in the variational upper bound is

$$\mathsf{R}_{\eta,x^n}^{Z^n} := \frac{1}{N_\eta(x^n)} \sum_{a=1}^{N_\eta(x^n)} \widetilde{\mathsf{P}}_{a,x^n}^{Z^n}.$$

This is a probability measure on the same latent space as $Z^n$. It depends on $x^n$ and on the chosen cover, but it does not depend on the realized value of $U$. It is only an auxiliary reference measure used to upper-bound the mutual information.

If the fraction-CMI proof denotes by $\mathsf{P}_{\varphi,u,x^n}^{Z^n}$ the conditional law of $Z^n$ given $(U = u, \phi = \varphi, X^n = x^n)$, then in the canonical sample-order representation used here

$$\mathsf{P}_{\varphi,u,x^n}^{Z^n} = \mathsf{P}_{\varphi,x^n}^{Z^n}, \qquad u \in \mathcal{U}_c, \tag{36}$$

where $\mathcal{U}_c := \{u \subset [n] : |u| = cn\}$. The split variable $U$ selects which coordinates are used in the empirical loss comparison, while the latent vector itself is represented in the original sample order.

By tensorization of KL divergence and the cover property (35),

$$\mathrm{KL}\Big(\mathsf{P}_{\varphi,x^n}^{Z^n} \,\Big\|\, \widetilde{\mathsf{P}}_{a_\eta(\varphi;x^n),x^n}^{Z^n}\Big)$$
$$= \sum_{i=1}^{n} \mathrm{KL}\big(q_\varphi(\cdot \mid x_i) \,\big\|\, \widetilde{q}_{a_\eta(\varphi;x^n)}(\cdot \mid x_i)\big) \leq n\eta^2.$$

Moreover, since $\mathsf{R}_{\eta,x^n}^{Z^n}$ contains $\widetilde{\mathsf{P}}_{a_\eta(\varphi;x^n),x^n}^{Z^n}$ with weight $1/N_\eta(x^n)$,

$$\mathrm{KL}\Big(\mathsf{P}_{\varphi,x^n}^{Z^n} \,\Big\|\, \mathsf{R}_{\eta,x^n}^{Z^n}\Big) \leq \mathrm{KL}\Big(\mathsf{P}_{\varphi,x^n}^{Z^n} \,\Big\|\, \widetilde{\mathsf{P}}_{a_\eta(\varphi;x^n),x^n}^{Z^n}\Big) + \log N_\eta(x^n)$$
$$\leq n\eta^2 + \log N_\eta(x^n). \tag{37}$$

The proof of the first inequality is provided in Appendix I.2.4.

The variational characterization of conditional mutual information gives, for any reference measure $R^{Z^n}$ that may depend on $(\varphi, x^n)$ but not on $U$,

$$I(Z^n; U \mid \phi = \varphi, X^n = x^n) \leq \mathbb{E}_{U|\varphi,x^n} \mathrm{KL}\Big(\mathsf{P}_{\varphi,U,x^n}^{Z^n} \,\Big\|\, R^{Z^n}\Big).$$

Taking $R^{Z^n} = \mathsf{R}_{\eta,x^n}^{Z^n}$ and using (36) and (37), we obtain

$$I(Z^n; U \mid \phi = \varphi, X^n = x^n) \leq \mathrm{KL}\Big(\mathsf{P}_{\varphi,x^n}^{Z^n} \,\Big\|\, \mathsf{R}_{\eta,x^n}^{Z^n}\Big)$$
$$\leq n\eta^2 + \log N_\eta(x^n).$$

Taking the worst case over $x^n$, define the $n$-point latent-variable covering complexity by

$$\mathfrak{C}_{\mathrm{LV}}^{(n)}(\eta) := n\eta^2 + \log \mathcal{N}_{\sqrt{\mathrm{KL}},n}(\eta, \mathcal{Q}). \tag{38}$$

Then

$$I(Z^n; U \mid \phi, X^n) \leq \mathfrak{C}_{\mathrm{LV}}^{(n)}(\eta).$$

Consequently, in the fraction-CMI bound,

$$\sqrt{\frac{I(Z^n; U \mid \phi, X^n)}{nc(1-c)^2}} \leq \frac{1}{\sqrt{c}(1-c)} \sqrt{\eta^2 + \frac{\log \mathcal{N}_{\sqrt{\mathrm{KL}},n}(\eta, \mathcal{Q})}{n}}.$$

We now upper-bound this $n$-point $\sqrt{\mathrm{KL}}$-covering number by $n$-point covers of the mean and standard-deviation classes. Define

$$\mathcal{M} := \{\mu_\varphi : \varphi \in \Phi\}, \qquad \mathcal{S} := \{\sigma_\varphi : \varphi \in \Phi\}.$$

Assume that the encoder standard deviations are uniformly bounded as

$$0 < \sigma_{\min} \le \sigma_{\varphi,j}(x) \le \sigma_{\max} < \infty, \qquad \forall x, \forall \varphi, \forall j \in [d_z].$$

For a vector-valued function class $\mathcal{F} \subseteq \{f : \mathcal{X} \to \mathbb{R}^{d_z}\}$, define the $n$-point empirical sup-covering number by

$$\mathcal{N}_{\infty,n}(\delta, \mathcal{F}) := \sup_{x_1,\ldots,x_n \in \mathcal{X}} \mathcal{N}(\delta, \mathcal{F}, d_{\infty,x_{1:n}}),$$

where

$$d_{\infty,x_{1:n}}(f,g) := \max_{i \in [n]} \|f(x_i) - g(x_i)\|_2.$$

Appendix I.2.3 proves that

$$\mathcal{N}_{\sqrt{\mathrm{KL}},n}(\eta, \mathcal{Q}) \le \mathcal{N}_{\infty,n}(\sigma_{\min}\eta, \mathcal{M})\, \mathcal{N}_{\infty,n}\left(\sqrt{\frac{\sigma_{\min}^8}{2\sigma_{\max}^6}}\eta, \mathcal{S}\right). \tag{39}$$

Combining (38) and (39), we obtain

$$\mathfrak{C}_{\mathrm{LV}}^{(n)}(\eta) \le n\eta^2 + \log \mathcal{N}_{\infty,n}(\sigma_{\min}\eta, \mathcal{M}) + \log \mathcal{N}_{\infty,n}\left(\sqrt{\frac{\sigma_{\min}^8}{2\sigma_{\max}^6}}\eta, \mathcal{S}\right). \tag{40}$$

Therefore the latent-variable contribution is bounded by

$$\sqrt{\frac{I(Z^n; U \mid \phi, X^n)}{nc(1-c)^2}} \le \frac{1}{\sqrt{c}(1-c)}\left[\eta^2 + \frac{\log \mathcal{N}_{\infty,n}(\sigma_{\min}\eta, \mathcal{M})}{n} + \frac{\log \mathcal{N}_{\infty,n}\left(\sqrt{\frac{\sigma_{\min}^8}{2\sigma_{\max}^6}}\eta, \mathcal{S}\right)}{n}\right]^{1/2}. \tag{41}$$

Suppose now that the $n$-point covering entropies of the mean and standard-deviation classes satisfy, for some constants $C > 0$ and $d_{\mathrm{eff}} > 0$,

$$\log \mathcal{N}_{\infty,n}(\delta, \mathcal{M}) + \log \mathcal{N}_{\infty,n}(\delta, \mathcal{S}) \le d_{\mathrm{eff}} \log\left(\frac{C}{\delta}\right)$$

for all sufficiently small $\delta > 0$. Absorbing constants depending only on $\sigma_{\min}$ and $\sigma_{\max}$ into $C$, and choosing $\eta_n = 1/n$, (41) gives

$$\sqrt{\frac{I(Z^n; U \mid \phi, X^n)}{nc(1-c)^2}} \le \sqrt{\frac{\mathfrak{C}_{\mathrm{LV}}^{(n)}(\eta_n)}{nc(1-c)^2}} \lesssim \frac{1}{\sqrt{c}(1-c)}\sqrt{\frac{1}{n^2} + \frac{d_{\mathrm{eff}}\log(Cn)}{n}} \lesssim \frac{1}{\sqrt{c}(1-c)}\sqrt{\frac{d_{\mathrm{eff}}\log n}{n}}. \tag{42}$$

Thus, for fixed $c \in (0,1)$, the $n$-point covering argument gives a vanishing latent-variable contribution of order

$$O\left(\sqrt{\frac{d_{\mathrm{eff}}\log n}{n}}\right).$$

Combining (34) with (42), the fraction-CMI bound has a vanishing complexity term whenever the parameter-CMI contribution in (34) is also sublinear in $n$. In particular, for fixed $c$, if $I(\phi; U \mid X^n) = o(n)$, then the parameter and latent contributions vanish as $n \to \infty$.

I.2.3. PROOF OF EQ. (40)

To prove Eq. (40), we use the following relation;

**Lemma 6** (Pointwise KL bound for diagonal Gaussian encoders). *Let*

$$P_x = \mathcal{N}\big(\mu(x), \mathrm{diag}(\sigma^2(x))\big), \qquad \widetilde{P}_x = \mathcal{N}\big(\widetilde{\mu}(x), \mathrm{diag}(\widetilde{\sigma}^2(x))\big),$$

*where*

$$\sigma_{\min} \le \sigma_j(x), \widetilde{\sigma}_j(x) \le \sigma_{\max}, \qquad j \in [d_z].$$

*Then*

$$\mathrm{KL}(P_x \| \widetilde{P}_x) \le \frac{1}{2\sigma_{\min}^2} \| \mu(x) - \widetilde{\mu}(x) \|_2^2 + \frac{\sigma_{\max}^6}{\sigma_{\min}^8} \| \sigma(x) - \widetilde{\sigma}(x) \|_2^2.$$

*Proof.* For each coordinate $j$, write $P_{x,j} = \mathcal{N}(\mu_j, \sigma_j^2)$ and $\widetilde{P}_{x,j} = \mathcal{N}(\widetilde{\mu}_j, \widetilde{\sigma}_j^2)$. The one-dimensional Gaussian KL divergence is

$$\mathrm{KL}(P_{x,j} \| \widetilde{P}_{x,j}) = \frac{(\mu_j - \widetilde{\mu}_j)^2}{2\widetilde{\sigma}_j^2} + \frac{1}{2}\left(r_j - 1 - \log r_j\right), \qquad r_j := \frac{\sigma_j^2}{\widetilde{\sigma}_j^2}.$$

The mean term satisfies

$$\frac{(\mu_j - \widetilde{\mu}_j)^2}{2\widetilde{\sigma}_j^2} \le \frac{(\mu_j - \widetilde{\mu}_j)^2}{2\sigma_{\min}^2}.$$

For the variance term, let $g(r) = r - 1 - \log r$. Then $g(1) = g'(1) = 0$ and $g''(r) = 1/r^2$. Since

$$r_j \in \left[ \left(\frac{\sigma_{\min}}{\sigma_{\max}}\right)^2, \left(\frac{\sigma_{\max}}{\sigma_{\min}}\right)^2 \right],$$

Taylor's theorem gives

$$g(r_j) \le \frac{1}{2}\left(\frac{\sigma_{\max}}{\sigma_{\min}}\right)^4 (r_j - 1)^2.$$

Moreover,

$$|r_j - 1| = \left| \frac{\sigma_j^2 - \widetilde{\sigma}_j^2}{\widetilde{\sigma}_j^2} \right| \le \frac{|\sigma_j - \widetilde{\sigma}_j|(\sigma_j + \widetilde{\sigma}_j)}{\sigma_{\min}^2} \le \frac{2\sigma_{\max}}{\sigma_{\min}^2}|\sigma_j - \widetilde{\sigma}_j|.$$

Therefore,

$$\frac{1}{2}\left(r_j - 1 - \log r_j\right) \le \frac{\sigma_{\max}^6}{\sigma_{\min}^8}(\sigma_j - \widetilde{\sigma}_j)^2.$$

Summing over $j = 1, \ldots, d_z$ proves the claim. $\qquad\square$

Fix arbitrary points $x_1, \ldots, x_n \in \mathcal{X}$. Let $\mathcal{C}_\mu$ be a $\sigma_{\min}\eta$-cover of $\mathcal{M}$ under $d_{\infty, x_{1:n}}$, and let $\mathcal{C}_\sigma$ be a $\sqrt{\sigma_{\min}^8/(2\sigma_{\max}^6)}\eta$-cover of $\mathcal{S}$ under the same empirical metric. For any $\mu \in \mathcal{M}$ and $\sigma \in \mathcal{S}$, choose $\widetilde{\mu} \in \mathcal{C}_\mu$ and $\widetilde{\sigma} \in \mathcal{C}_\sigma$ such that

$$\max_{i \in [n]} \| \mu(x_i) - \widetilde{\mu}(x_i) \|_2 \le \sigma_{\min}\eta,$$

$$\max_{i \in [n]} \| \sigma(x_i) - \widetilde{\sigma}(x_i) \|_2 \le \sqrt{\frac{\sigma_{\min}^8}{2\sigma_{\max}^6}}\eta.$$

For each pair $(\widetilde{\mu}, \widetilde{\sigma})$, define the auxiliary Gaussian kernel

$$\widetilde{q}_{\widetilde{\mu}, \widetilde{\sigma}}(\cdot \mid x) := \mathcal{N}\left(\widetilde{\mu}(x), \operatorname{diag}(\widetilde{\sigma}^2(x))\right).$$

By Lemma 6, for every $i \in [n]$,

$$\operatorname{KL}\left(\mathcal{N}(\mu(x_i), \operatorname{diag}(\sigma^2(x_i))) \,\middle\|\, \mathcal{N}(\widetilde{\mu}(x_i), \operatorname{diag}(\widetilde{\sigma}^2(x_i)))\right)$$

$$\leq \frac{1}{2\sigma_{\min}^2} \|\mu(x_i) - \widetilde{\mu}(x_i)\|_2^2 + \frac{\sigma_{\max}^6}{\sigma_{\min}^8} \|\sigma(x_i) - \widetilde{\sigma}(x_i)\|_2^2$$

$$\leq \frac{\eta^2}{2} + \frac{\eta^2}{2} = \eta^2.$$

Taking square roots and maximizing over $i \in [n]$, the auxiliary Gaussian kernel $\widetilde{q}_{\widetilde{\mu}, \widetilde{\sigma}}$ is an $\eta$-approximation of $q_\varphi$ under $d_{\sqrt{\mathrm{KL}}, x^n}$. Therefore the Cartesian product $\mathcal{C}_\mu \times \mathcal{C}_\sigma$ induces an external $\eta$-cover of the Gaussian encoder posterior family under $d_{\sqrt{\mathrm{KL}}, x^n}$. Hence, for these fixed points,

$$\mathcal{N}\left(\eta, \mathcal{Q}, d_{\sqrt{\mathrm{KL}}, x^n}\right) \leq \mathcal{N}(\sigma_{\min}\eta, \mathcal{M}, d_{\infty, x_{1:n}}) \mathcal{N}\left(\sqrt{\frac{\sigma_{\min}^8}{2\sigma_{\max}^6}}\eta, \mathcal{S}, d_{\infty, x_{1:n}}\right).$$

Taking the supremum over $x_1, \ldots, x_n \in \mathcal{X}$ gives

$$\mathcal{N}_{\sqrt{\mathrm{KL}}, n}(\eta, \mathcal{Q}) \leq \mathcal{N}_{\infty, n}(\sigma_{\min}\eta, \mathcal{M}) \mathcal{N}_{\infty, n}\left(\sqrt{\frac{\sigma_{\min}^8}{2\sigma_{\max}^6}}\eta, \mathcal{S}\right). \tag{43}$$

Combining (38) with (43) proves (40).

### I.2.4. UPPER BOUND OF THE MI BASED ON THE MIXTURE

We use the following elementary finite-mixture KL bound.

**Lemma 7** (Finite-mixture KL bound). *Let $P, Q_1, \ldots, Q_N$ be probability measures on a common measurable space, and let*

$$R := \frac{1}{N} \sum_{a=1}^{N} Q_a.$$

*Then*

$$\operatorname{KL}(P\|R) \leq \min_{a \in [N]} \left\{\operatorname{KL}(P\|Q_a) + \log N\right\}.$$

*More generally, if $R = \sum_{a=1}^{N} w_a Q_a$ with $w_a > 0$ and $\sum_{a=1}^{N} w_a = 1$, then*

$$\operatorname{KL}(P\|R) \leq \min_{a \in [N]} \left\{\operatorname{KL}(P\|Q_a) - \log w_a\right\}.$$

*Proof.* It suffices to prove the weighted version. Fix $a \in [N]$. Since

$$R = \sum_{b=1}^{N} w_b Q_b$$

contains $Q_a$ with weight $w_a$, we have the measure domination

$$R \geq w_a Q_a.$$

Equivalently, for any common dominating measure, if $p, r, q_a$ denote the densities of $P, R, Q_a$, respectively, then

$$r \geq w_a q_a.$$

Therefore,

$$\log \frac{p}{r} \le \log \frac{p}{w_a q_a} = \log \frac{p}{q_a} - \log w_a.$$

Integrating with respect to $P$ gives

$$
\begin{aligned}
\mathrm{KL}(P\|R) &= \int \log \frac{p}{r}\, dP \\
&\le \int \log \frac{p}{q_a}\, dP - \log w_a \\
&= \mathrm{KL}(P\|Q_a) - \log w_a.
\end{aligned}
$$

Since this holds for every $a \in [N]$, taking the minimum over $a$ gives the claim. The uniform-mixture case follows by setting $w_a = 1/N$. □

## J. Experimental Methodology

In this section, we provide a comprehensive description of our experimental methodology, including the datasets, cross-validation procedure, model training, objective functions, and the techniques used for mutual information estimation. The source code to reproduce all experiments is available at https://github.com/msfuji0211/vae_it_analysis.

### J.1. Datasets, Preprocessing, and Cross-Validation

We used the MNIST and Fashion-MNIST datasets. Both consist of $28 \times 28$ grayscale images. For all MLP-based models, images were flattened into 784-dimensional vectors. Input pixel values were normalized to the range $[0, 1]$.

**Training size ($n_{\text{train}}$) configuration:** To analyze the dependency of our results on the amount of training data, we systematically varied the training set size $n_{\text{train}}$. The values for $n_{\text{train}}$ were chosen on a logarithmic scale, ranging from approximately 1,000 to 30,000, with 7 distinct points within this range, as automated by our experimental scripts.

**Cross-validation scheme:** To ensure robust evaluation, we employed a 10-fold cross-validation scheme for each configured training size $n_{\text{train}}$. Our procedure is designed to create non-overlapping training and validation sets for each fold. For a given $n_{\text{train}}$, we first create a data pool large enough to accommodate both training and validation sets. For instance, to achieve a training set of $n_{\text{train}} = 9,000$ with a validation ratio $r = 0.1$, we first create a larger pool of $9,000/(1 - 0.1) = 10,000$ samples. This pool is then subjected to 10-fold cross-validation. In each of the 10 folds, one distinct block of 1,000 samples is designated as the validation set, while the remaining nine blocks (totaling 9,000 samples) are used for training. This ensures that for each of the 10 runs, the model is trained on exactly $n_{\text{train}}$ samples. A fresh model was trained from scratch for each fold, and the final reported metrics are the mean and standard deviation computed across these 10 independent runs.

### J.2. Generalization Gap and Split-Wise Correlation Computation

We clarify how the empirical generalization gap and the correlation coefficients reported in Table 2 were computed. The population loss $L_{\mathcal{D}}$ appearing in the definition of the generalization error is not directly observable in experiments. We therefore approximate it by a held-out loss using the cross-validation scheme described above. For each fold, after training a fresh model on the training block, we compute

$$\widehat{\text{gap}} = |L_{\text{test}} - L_{\text{train}}|,$$

where $L_{\text{train}}$ is the empirical reconstruction loss on the training block and $L_{\text{test}}$ is the corresponding loss on the held-out block. The plots report the mean and standard deviation of this held-out estimate across the 10 folds.

For the correlation analysis in Table 2, we fix $n_{\text{train}} = 30,000$. For each of the 10 cross-validation splits, we compute one scalar value for each of the following four quantities:

1. the empirical gap $|L_{\text{test}} - L_{\text{train}}|$;

2. the influence-function-based proxy for the encoder term $I(\phi; U \mid X^n)$;

3. the computable maximum-entropy upper bound for the latent term $I(Z^n; U \mid \phi, X^n)$, evaluated using the uniform prior on $U$ described in Appendix J.5;

4. the combined diagnostic quantity $\sqrt{I(Z^n; U \mid \phi, X^n) + I(\phi; U \mid X^n)}$, after applying the same normalization used in the plots.

The Pearson, Spearman, and Kendall coefficients are then computed across these 10 split-wise scalar values. Thus, Table 2 should be interpreted as a diagnostic correlation analysis based on computable proxies and upper bounds, rather than as an exact direct estimation of the mutual information terms in Theorem 2.

### J.3. Training Procedure and Objective Functions

All models were trained using the Adam optimizer (Kingma & Ba, 2015). The model state that achieved the lowest validation loss was saved for all subsequent analyses. The common training hyperparameters are listed in Table 4.

*Table 4.* Common training hyperparameters for MNIST and Fashion-MNIST experiments.

| Hyperparameter | Value |
|---|---|
| Optimizer | Adam |
| Learning Rate | 0.0005 |
| Weight Decay | 0.00001 |
| Batch Size | 128 |
| Maximum Epochs | 100 |
| Early Stopping Patience | 10 |
| Input Dimension | 784 |

The **early stopping patience** of 10 means that training is terminated if the validation loss fails to decrease for 10 consecutive epochs. An improvement is defined as any strict decrease in the validation loss compared to the best value observed in previous epochs; no minimum improvement threshold is applied. The input dimension of 784 corresponds to the dimensionality of the flattened $28 \times 28$ pixel images from both datasets.

**Objective functions:** Our models were trained using two distinct variational objectives. For the standard VAE, we maximized the conventional ELBO, with the KL divergence term weighted by $\beta = 0.1$:

$$\mathcal{L}_{\text{ELBO}}(x) = \mathbb{E}_{q_\phi(z|x)}[\log p_\theta(x|z)] - \beta \cdot \text{KL}(q_\phi(z|x)||p(z)).$$

For the hierarchical VAE (H-VAE), we maximized the importance weighted autoencoder (IWAE) bound (Burda et al., 2015). This choice is motivated by the fact that powerful generative models like H-VAEs can be susceptible to *posterior collapse*, where LVs are ignored by the decoder. The IWAE objective is known to mitigate this issue by providing a tighter lower bound on the log-likelihood and is a common choice for training deep hierarchical VAEs. The IWAE objective with $K$ samples is defined as:

$$\mathcal{L}_K(x) = \mathbb{E}_{z_1,\ldots,z_K \sim q_\phi(z|x)} \left[ \log \left( \frac{1}{K} \sum_{k=1}^{K} \frac{p_\theta(x, z_k)}{q_\phi(z_k \mid x)} \right) \right].$$

The number of samples $K$ used for each latent layer in our hierarchical models is specified in Table 5.

### J.4. Model Architectures

Our experiments utilize two main architectural backbones for the VAE models: a Multi-Layer Perceptron (MLP) and a Convolutional Neural Network (CNN). The MLP-based models operate on flattened 784-dimensional image vectors, while the CNN-based models take the original $28 \times 28$ images as input. For both backbones, we define architectures for a Standard VAE and a Hierarchical VAE. All hidden layers, whether fully-connected or convolutional, are followed by a ReLU activation and Batch Normalization. The detailed configurations for all four model variants are summarized in Table 5.

For the CNN-based models, the encoder utilizes a series of 2D convolutional layers. The first two layers have a stride of 2 to downsample the spatial resolution, while subsequent layers maintain it. The decoder mirrors this structure using transposed convolutions for upsampling.

*Table 5.* Architectural configurations and objective-specific parameters for all VAE models.

| Model Type | Backbone | Encoder(-like) layer | Decoder(-like) layer | Latent Specification | Objective-Specific Params |
|---|---|---|---|---|---|
| Standard VAE | MLP | Hidden Dims: `[512, 256, 128, 64]` | Hidden Dims: `[64, 128, 256, 512]` | Dim ($d_z$): 32 | IWAE ($K = 50$) |
| | CNN | Channels: `[32,64,128,128]` | Channels: `[32,64,128,128]` | Dim ($d_z$): 32 | IWAE ($K = 50$) |
| Hierarchical VAE | MLP ($L = 8$) | Stoc. layer Dims: `[512, 256, 128, 64]` | Stoc. layer Dims: `[64, 128, 256, 512]` | Dims ($d_{z_l}$): `[32,16,8,4]` | IWAE ($K = [50,5,1,1]$) |
| | CNN ($L = 8$) | Channels: `[32,64,64,64]` | Channels: `[32,64,64,64]` | Dims ($d_{z_l}$): See note below | IWAE ($K = [50,5,1,1]$) |

**Note:** For the Hierarchical CNN model, two latent configurations were used: `[64,32,16,8]` for IWAE and `[32,16,8,4]` for standard ELBO.

Note that for the Hierarchical CNN model, two different latent dimension configurations were used: `[64, 32, 16, 8]` for experiments with the IWAE objective, and `[32, 16, 8, 4]` for those with the ELBO objective.

## J.5. Computation of Information-Theoretic Upper Bounds

We computed computable upper bounds for two key MI quantities in our theoretical results (Theorems 2 and 3). These bounds serve as complexity indicators and were evaluated for each trained model across all splits and training sizes.

**Upper Bound on $I(\phi; U|X^n)$:** We compute a quadratic upper bound on the mutual information between model parameters $\phi$ and the data selection variable $U$, based on the influence function (Rammal et al., 2022). The Hessian term $H$ is approximated using a diagonal Empirical Fisher (EF) matrix. The $j$-th diagonal element of the EF matrix is defined as the expectation of the squared gradient with respect to the $j$-th parameter over the empirical data distribution $\hat{p}_n$:

$$\text{EF}_{jj} = \mathbb{E}_{x \sim \hat{p}_n}[(\nabla_{\phi_j} \ell(\phi, x))^2],$$

where $\ell(\phi, x)$ is the loss of VAE or IWAE for a single sample $x$. The bound is given by:

$$I(\phi; U|X^n) \lesssim \frac{1}{2n_{\text{train}}} \mathbb{E}_{i,j \sim U} \left[ (\text{grad}_i - \text{grad}_j)^T H^{-1} (\text{grad}_i - \text{grad}_j) \right],$$

where $\text{grad}_i = \nabla_\phi \ell(\phi, x_i)$ is the gradient for sample $i$, and the inverse Hessian is approximated as $H^{-1} \approx \text{diag}(1/(\text{EF}+\lambda))$, with $\lambda$ being a damping constant.

**Upper Bound on $I(Z^n; U|\phi, X^n)$:** We compute a maximum-entropy upper bound on the mutual information between the latent representation $Z^n$ and the index $U$. Strictly speaking, conditioned on the trained model $\phi$, the posterior distribution of the index $p(U \mid \phi, X^n)$ is not uniform. However, our LOO construction induces the following inequality. For the discretization $J^n$ of $Z^n$, which is introduced in the proof of Theorem 2 (see the corresponding Theorem 5), the following relationship holds.

$$I_{P(U|\phi,X^n)}(J^n; U \mid X^n, \phi) \leq I_{\text{Unif}}(J^n; U \mid X^n, \phi). \tag{44}$$

The proof of Eq. (44) is provided in Appendix J.5.1. Then by using the data-processing inequality, we have $I_{\text{Unif}}(J^n; U \mid X^n, \phi) \leq I_{\text{Unif}}(Z^n; U|\phi, x^n)$. This means that, in the upper bound of Theorem 2, the term $I(Z^n; U \mid \phi, X^n)$ is replaced by $I_{\text{Unif}}(Z^n; U \mid \phi, X^n)$. Therefore, throughout the following analysis, we employ the uniform distribution $U \sim \text{Unif}(1, \ldots, n)$ thereby obtaining a numerically evaluable upper bound.

Under this upper-bound, we estimate the conditional mutual information as

$$I(Z^n; U \mid \phi, X^n) = H(Z^n \mid \phi, X^n) - H(Z^n \mid \phi, X^n, U).$$

Both entropy terms can be evaluated under the Gaussian assumption. Specifically, the latent variables $Z_1, \ldots, Z_n$ are independent, with

$$Z_i \sim q(Z_i \mid \phi, X_i) = \mathcal{N}(\mu_i, \Sigma_i), \qquad \Sigma_i = \text{diag}(\sigma_\phi^2(X_i)).$$

Accordingly, the concatenated vector $Z^n = (Z_1, \ldots, Z_n)$ follows a Gaussian distribution whose parameters depend on the random index $U$.

The second term satisfies

$$H(Z^n \mid \phi, X^n, U) = \mathbb{E}_{i \sim U} \big[ H(Z^n \mid \phi, X^n, U = i) \big],$$

and since $q(Z_i \mid \phi, X_i)$ is Gaussian, each conditional entropy $H(Z^n \mid \phi, X^n, U = i)$ can be computed analytically from the variances $\{\sigma_{i,d_z}^2\}$ of $q(Z_i \mid \phi, X_i)$.

We now turn to $H(Z^n \mid \phi, X^n)$. Under the uniform choice $U \sim \text{Unif}([n])$, the covariance of $Z^n$ admits an explicit decomposition via the law of total variance:

$$\text{Cov}(Z^n \mid \phi, X^n) = \mathbb{E}_{U \sim \text{Unif}([n])}[\text{Cov}(Z^n \mid \phi, X^n, U)] + \text{Cov}_{U \sim \text{Unif}([n])}(\mathbb{E}[Z^n \mid \phi, X^n, U]). \qquad (45)$$

First, conditional on $(\phi, X^n, U = u)$, the latent variables remain independent Gaussians, and hence $\text{Cov}(Z^n \mid \phi, X^n, U = u)$ is a block-diagonal matrix whose blocks are given by the per-sample covariance matrices $\Sigma_i$. Averaging over $u$ under the uniform distribution yields

$$\mathbb{E}_{U \sim \text{Unif}([n])}[\text{Cov}(Z^n \mid \phi, X^n, U)] = \frac{1}{n} \sum_{u=1}^{n} \text{Cov}(Z^n \mid \phi, X^n, U = u).$$

The second term in (45), $\text{Cov}_U(\mathbb{E}[Z^n \mid \phi, X^n, U])$, captures the variability of the conditional mean vector induced by the random index $U$. In the LOO construction, changing the value of $U$ alters which sample is treated as held out, thereby modifying the corresponding components of the conditional expectation vector. As a result, this term is fully determined by the collection of encoder means $(\mu_1, \ldots, \mu_n)$ and reflects the information about $U$ retained in the latent representation.

Finally, we apply the maximum entropy principle: among all distributions with a fixed covariance matrix, the Gaussian distribution maximizes differential entropy. Consequently,

$$H(Z^n \mid \phi, X^n) \leq \frac{1}{2} \log \det \big( \text{Cov}(Z^n \mid \phi, X^n) \big) + \text{const},$$

where the additive constant depends only on the dimensionality.

In conclusion, we obtain

$$I(Z^n; U \mid \phi, X^n) \ \leq \ \frac{1}{2} \log \det \big( \text{Cov}(Z^n \mid \phi, X^n) \big) - \sum_{i=1}^{n} \sum_{d=1}^{d_z} \log \sigma_{i,d}^2,$$

and we treat this quantity as a realization-dependent proxy for the complexity of the latent representation.

### J.5.1. PROOF OF EQ. (44)

We fix the dataset $X^n$ and encoder parameters $\phi$ throughout this discussion. We work on the discretized latent variables $J$ obtained by assigning each latent variable $Z$ to the index of its nearest centroid in a fixed codebook. Let $J^n = (J_1, \ldots, J_n) \in \mathcal{J}^n$ denote the discretized version of $Z^n$.

Under the LOO construction, the index of the held-out sample $U \in [n] := \{1, \ldots, n\}$ serves as the channel input, and the discretized latent vector $J^n$ serves as the channel output. Conditioned on $(X^n, \phi)$, this induces a discrete channel

$$W_{X^n, \phi}(j^n \mid u) \ := \ q(J^n = j^n \mid X^n, \phi, U = u), \qquad u \in [n], \ j^n \in \mathcal{J}^n.$$

For any input distribution $P$ on $[n]$, we denote by $I_P(J^n; U \mid X^n, \phi)$ the mutual information induced by this channel.

A key property of the LOO construction is its invariance to relabeling of sample indices. Formally, for any permutation $\tau$ of $[n]$, there exists a corresponding permutation $\Pi_\tau$ of the coordinates of $J^n$ such that

$$W_{X^n, \phi}(\Pi_\tau(j^n) \mid \tau(u)) = W_{X^n, \phi}(j^n \mid u), \qquad \forall u \in [n], \ \forall j^n \in \mathcal{J}^n.$$

This follows because the learning algorithm and the discretization codebook do not depend on the specific label of the held-out index, but only on which sample is omitted. Given any input distribution $P$ on $[n]$ and permutation $\tau$, define the transformed distribution $P_\tau(u) := P(\tau^{-1}(u))$.

By the above permutation equivariance, the joint distributions induced by $(U, J^n) \sim P(u) W_{X^n, \phi}(j^n \mid u)$ and $(\tau(U), \Pi_\tau(J^n))$ are identical. Since mutual information is invariant under bijective transformations of the variables, this implies

$$I_P(J^n; U \mid X^n, \phi) = I_{P_\tau}(J^n; U \mid X^n, \phi), \qquad \forall \tau.$$

For a fixed channel $W_{X^n, \phi}$, the mutual information $P \mapsto I_P(J^n; U \mid X^n, \phi)$ is a concave function of the input distribution $P$ (Cover & Thomas, 2012). Define the symmetrized input distribution

$$\bar{P} := \frac{1}{|\mathfrak{S}_n|} \sum_{\tau \in \mathfrak{S}_n} P_\tau = \frac{1}{n!} \sum_{v=1}^n (n-1)! \, P(v) = \frac{(n-1)!}{n!} \sum_{v=1}^n P(v) = \frac{1}{n},$$

where $\mathfrak{S}_n$ denotes the symmetric permutation on $[n]$. The second equality is because each element of $[n]$ appears equally often as $\tau^{-1}(u)$ when averaging over all permutations. By concavity and the invariance established above, we obtain

$$I_{\text{Unif}}(J^n; U \mid X^n, \phi) = I_{\bar{P}}(J^n; U \mid X^n, \phi) \geq \frac{1}{|\mathfrak{S}_n|} \sum_{\tau \in \mathfrak{S}_n} I_{P_\tau}(J^n; U \mid X^n, \phi) = I_P(J^n; U \mid X^n, \phi).$$

Hence, the symmetrized distribution $\bar{P}$ coincides with the uniform distribution on $[n]$.

### J.6. $\beta$-VAE Sweep Experiments

We provide the experimental details for the $\beta$-VAE sweep summarized in Table 3. The purpose of this experiment is to vary the strength of latent regularization while keeping the rest of the experimental pipeline fixed. This isolates whether the computable quantities in Theorem 2 follow changes in the reconstruction generalization gap induced through the latent bottleneck.

All models in this sweep are standard single-latent-layer VAEs. For MNIST and Fashion-MNIST, the architecture is the same as the standard-VAE rows in Table 5 (architecture columns only): the MLP backbone for MNIST and the CNN backbone for Fashion-MNIST. The common optimizer and early-stopping settings are those in Table 4. CIFAR-10 is not covered by Table 5; for this dataset, we use a standard single-latent-layer CNN VAE with channels [64, 128, 128, 128] and latent dimension $d_z = 64$. The same optimizer settings are used for CIFAR-10. The only hyperparameter changed in the sweep is the KL weight $\beta$ in the ELBO objective.

We us $\beta \in \{0.01, 0.03, 0.3, 1.0, 3.0, 10.0\}$. For every value of $\beta$, the training size is fixed to 30,000, and the implementation uses a test size of 10,000. As in the cross-validation protocol described in Appendix J.2, each value of $\beta$ is evaluated using ten split-wise runs. The reported statistics therefore aggregate split-wise runs.

For each split-wise run, we compute the empirical reconstruction generalization gap $|\text{gap}_{\text{recon}}| = |L_{\text{test}}^{\text{recon}} - L_{\text{train}}^{\text{recon}}|$. This choice is consistent with our theory, which concerns reconstruction-based generalization. We then pair this gap with the same computable quantities used in the experiments above: the influence-function-based proxy for $I(\phi; U \mid X^n)$, the maximum-entropy upper bound for $I(Z^n; U \mid \phi, X^n)$ described in Appendix J.5, and the combined quantity $\sqrt{I(Z^n; U \mid \phi, X^n) + I(\phi; U \mid X^n)}$.

For each dataset, we pool the $6 \times 10 = 60$ split-wise pairs across the six values of $\beta$ and compute Pearson, Spearman, and Kendall correlation coefficients. This pooled evaluation is intended to test whether each computable quantity tracks the variation of the empirical gap over the entire $\beta$ sweep. Computing correlations separately for each fixed $\beta$ would instead mainly measure split-to-split variability at a fixed latent capacity, and would use only ten points per correlation.

Finally, as with the other numerical results, these quantities are computable proxies or upper bounds, not exact mutual information values. The $\beta$-sweep correlations should therefore be interpreted as a diagnostic analysis of whether the proposed encoder–latent-variable decomposition reflects empirical behavior under controlled changes of latent capacity, rather than as tight finite-sample certificates of the exact information-theoretic bound.

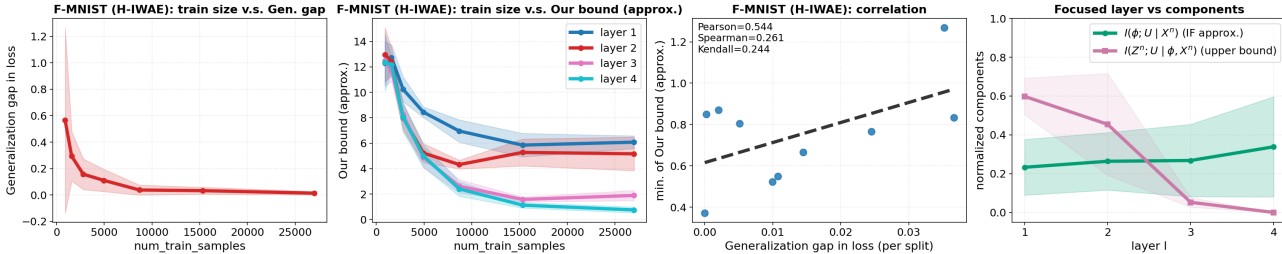

*Figure 3.* Visualization of our layer-wise analysis (Theorem 3) for a hierarchical IWAE trained on Fashion-MNIST. The model consists of a 4-layer encoder and a 4-layer decoder ($L = 8$); we analyze the properties of each encoder layer ($l = 1, 2, 3, 4$). The panels show: (1) Empirical generalization gap vs. training set size $n$. (2) Our **computable layer-wise upper bounds** vs. $n$. (3) Correlation at $n = 30000$ between the generalization gap and the minimum of the layer-wise bounds. (4) The two components of our bound (computable proxies) plotted against the encoder layer index $l$. Note that in Panel (4), the latent complexity term decreases for deeper layers, illustrating the progressive compression of information. All plots show the mean $\pm$ standard deviation. Values in panels (3) and (4) are normalized.

# K. Additional Experiments

In this section, we present a quantitative evaluation of the models used in our study. We analyze their performance across different datasets (MNIST and Fashion-MNIST), objective functions (standard ELBO and IWAE), and varying training set sizes ($n$). The primary metrics for this analysis are the test/train total loss and the reconstruction loss, with results aggregated as the mean and standard deviation over 10 cross-validation splits.

## K.1. Additional results and Limitation

Here, we detail the performance statistics of our models and discuss the observed trends and potential limitations.

### K.1.1. PERFORMANCE STATISTICS

The performance of the Standard VAE and the Hierarchical VAE are summarized in Tables 6 through 11.

**Standard VAE:** For the Standard VAE trained with the standard ELBO objective (Table 6 and 7), we observe a consistent trend across both MNIST and Fashion-MNIST datasets. As the training size $n$ increases, both test and train losses steadily decrease, indicating that the model effectively leverages larger datasets to improve its generative performance. The generalization gap, i.e., the difference between test and train loss, also tends to narrow with more data. Notably, the overall loss values for Fashion-MNIST are higher than for MNIST, which is expected as Fashion-MNIST is a more complex image reconstruction task. When trained with the IWAE objective (Table 8 and 9), the model exhibits a similar trend of performance improvement with increasing data.

**Hierarchical VAE:** The Hierarchical VAE, trained with the IWAE objective (Table 10 and 11), also shows a clear benefit from larger training sets. The reconstruction loss, in particular, shows a notable decrease as $n$ increases. For instance, on MNIST, the test reconstruction loss drops from approximately $5.89$ at $n = 900$ to $0.51$ at $n = 27000$. This indicates that with sufficient data, the hierarchical architecture can learn a latent structure that leads to more accurate reconstructions.

Beyond standard performance metrics, we further applied our layer-wise information-theoretic analysis to this model to examine how the generalization components evolve across the hierarchy. The model consists of $L = 8$ stochastic layers, organized into two symmetric blocks: the first four layers ($l = 1, \ldots, 4$) serve an encoder-like role by progressively compressing information into LVs of decreasing dimensionality, $(d_{z_1}, \ldots, d_{z_4}) = (32, 16, 8, 4)$, while the latter four layers ($l = 5, \ldots, 8$) perform a decoder-like role (see Table 5 for full details).

To evaluate the terms in our bound, we employed the same methodology as in the single-layer experiments. Specifically, we computed the influence-based proxy for the encoder term and the computable upper bound for the latent term (using a Gaussian approximation under a uniform prior on $U$, as justified in Appendix J.5).

Figure 3 visualizes the results on Fashion-MNIST. Panels (1) and (2) display the empirical generalization gap alongside our layer-wise computable bounds. While these proxies are not exact estimators, they exhibit qualitative behavior that aligns with the empirical generalization gap across different layers. A key observation regarding the hierarchical structure is visible in Panel (4), which plots the individual bound components against the layer depth. We observe that as the hierarchy

deepens ($l = 1 \rightarrow 4$), the latent complexity term $I(Z_l^n; U \mid X^n, \phi_{1:l})$ decreases, whereas the parameter complexity term $I(\phi_{1:l}; U \mid X^n)$ accumulates. This behavior is consistent with the theoretical trade-off discussed in Section 4.3 and provides empirical support for the principle of progressive abstraction, where higher layers form more compressed representations at the cost of increased parameter usage.

### K.1.2. POTENTIAL LIMITATIONS

While our theoretical results provide a framework for understanding generalization, their practical application relies on tractable estimators for the MI terms. A key limitation, observable in our experiments, is the coarseness of these estimations and the challenges that arise from using upper bounds.

For the standard VAE, the approximations appear particularly loose. As shown in the correlation plots for both MNIST and Fashion-MNIST (see Figure 4), we observe a weak negative or near-zero correlation between our total bound and the empirically measured generalization gap. This failure to track the true gap stems primarily from the behavior of the upper bound on the latent variable term, $I(Z^n; U|\phi, X^n)$. The component plots reveal that this term dominates the total bound and, counter-intuitively, tends to increase with the training set size, likely because the upper bound becomes looser.

For the hierarchical VAE, the situation is more nuanced but reveals the same core issue. While the overall bound shows a decreasing trend with sample size on MNIST (see Figure 6), the correlation with the generalization gap remains weakly negative. The layer-wise analysis highlights a complex interplay between the MI components: as we consider deeper layers, the parameter information term $I(\phi; U|X^n)$ increases while the latent information term $I(Z^n; U|\phi, X^n)$ decreases. However, the latent term still problematically increases with sample size on the more complex Fashion-MNIST dataset (see Figure 5).

These findings consistently point to the limitations of the tractable estimators used for our information-theoretic terms. The looseness of the maximum-entropy upper bound for $I(Z^n; U|\phi, X^n)$ in particular appears to be a significant bottleneck as in Appendix J.5. This highlights the development of tighter and more reliable estimators as a crucial direction for future work.

**Scope of losses and model classes.** Our proof is stated for a bounded squared reconstruction loss. This boundedness is used to control exponential moments in the Donsker–Varadhan and McDiarmid-type arguments, and the Gaussian posterior assumption is used to control the latent tail before taking the truncation-and-discretization limit. The argument is therefore not a direct proof for arbitrary VAE objectives, Wasserstein autoencoders, or deterministic autoencoders. Extensions to such models may be possible when one can impose an explicit bounded reconstruction loss, a bounded latent domain, or a comparable tail-control condition together with a continuity condition for the encoder. Without such assumptions, the truncation step and the data-processing step used in the present proof do not apply verbatim.

**Full ELBO and log-loss objectives.** Although VAEs are commonly trained by maximizing the ELBO, our theoretical generalization bound concerns the expected reconstruction loss, specifically the squared Euclidean reconstruction loss. A full generalization analysis of the entire ELBO would also require controlling the KL regularization term and, for likelihood-based reconstruction losses, the log-loss. If the log-loss is uniformly bounded, or if one works under a Bernoulli or Gaussian likelihood with additional boundedness or variance assumptions, variants of the present strategy may be applicable. In contrast, strictly unbounded log-losses would require a different proof framework, because the bounded-difference and exponential-moment arguments used here no longer hold directly.

**Finite-diameter assumption.** The finite-diameter assumption on the data space is mainly used to obtain a uniformly bounded reconstruction loss, which in turn enables the bounded-difference estimates and exponential-moment controls used in the LOO-CMI analysis. This assumption can make the numerical constants in the bound loose for complex or high-dimensional datasets, but it is not the conceptual source of the encoder–latent-variable decomposition. The core decomposition is driven by the LOO-CMI construction and the latent truncation-and-discretization argument. Relaxing this assumption to sub-Gaussian or moment-based conditions is an interesting direction for future work.

**Looseness of the computable bound.** The empirical quantities used in our experiments are computable proxies or upper bounds rather than exact mutual information values. In particular, the maximum-entropy upper bound for $I(Z^n; U \mid \phi, X^n)$ can be coarse and may dominate the combined bound in some regimes. We therefore interpret the numerical experiments as proof-of-concept diagnostics for the proposed decomposition, not as tight finite-sample certificates. Developing estimators that are both computationally scalable and statistically tighter remains an important direction for future work.

*Table 6.* Standard VAE (MNIST, MLP): mean $\pm$ standard deviation (aggregated over splits; $n$ after 10% leave-one-out)

| $n$ | Test loss | Train loss | Test reconstruction loss | Train reconstruction loss |
|---|---|---|---|---|
| 900 | $0.2389 \pm 0.0053$ | $0.2359 \pm 0.0056$ | $0.2221 \pm 0.0058$ | $0.2175 \pm 0.0063$ |
| 1587 | $0.1887 \pm 0.0011$ | $0.1769 \pm 0.0016$ | $0.1606 \pm 0.0011$ | $0.1487 \pm 0.0015$ |
| 2796 | $0.1760 \pm 0.0006$ | $0.1643 \pm 0.0010$ | $0.1448 \pm 0.0011$ | $0.1332 \pm 0.0014$ |
| 4929 | $0.1697 \pm 0.0008$ | $0.1598 \pm 0.0012$ | $0.1368 \pm 0.0016$ | $0.1268 \pm 0.0020$ |
| 8690 | $0.1652 \pm 0.0004$ | $0.1576 \pm 0.0007$ | $0.1308 \pm 0.0011$ | $0.1231 \pm 0.0013$ |
| 15317 | $0.1623 \pm 0.0007$ | $0.1575 \pm 0.0008$ | $0.1275 \pm 0.0015$ | $0.1226 \pm 0.0016$ |
| 27000 | $0.1600 \pm 0.0007$ | $0.1571 \pm 0.0007$ | $0.1249 \pm 0.0014$ | $0.1220 \pm 0.0013$ |

## K.2. Possible Way to Overcome the Limitation in MI Estimation

As established in the previous section, the looseness of the maximum-entropy upper bound for $I(Z^n; U | \phi, X^n)$ was a primary limitation. To address this, we consider a more direct, and potentially more accurate, method for estimating this mutual information term. Here, we drop the conditioning on $\phi$ and $X^n$ for notational simplicity.

The mutual information can be expressed as the expected KL divergence between each component distribution $q_i(z)$ and the full mixture distribution $q_{\mathrm{mix}}(z)$:

$$I(Z; U) = \mathbb{E}_{i \sim \mathrm{Unif}[n]} \left[ \mathrm{KL}\big(q_i(z) \| q_{\mathrm{mix}}(z)\big) \right] = \mathbb{E}_{i \sim \mathrm{Unif}[n], z \sim q_i} \left[ \log q_i(z) - \log q_{\mathrm{mix}}(z) \right],$$

where the mixture distribution is defined as $q_{\mathrm{mix}}(z) = \frac{1}{n} \sum_{j=1}^{n} q_j(z)$.

**Direct Monte Carlo estimation:** A straightforward approach to estimate this quantity is via Monte Carlo sampling. For each component $q_i$ in the expectation, we can estimate the KL divergence by drawing $m$ samples $z_{i,s}$ from $q_i$. This yields the following direct estimator:

$$\widehat{I}_{\mathrm{MC}} = \frac{1}{n} \sum_{i=1}^{n} \frac{1}{m} \sum_{s=1}^{m} \left[ \log q_i(z_{i,s}) - \log \left( \frac{1}{n} \sum_{j=1}^{n} q_j(z_{i,s}) \right) \right], \quad \text{where } z_{i,s} \sim q_i.$$

**Properties and practical challenges:** This estimator has several desirable theoretical properties. It provides an unbiased estimate of the KL divergence for each component and is consistent, converging to the true value of $I(Z; U)$ as the number of samples $m \to \infty$. By avoiding any bounding approximations, it has the potential to be significantly more accurate than the maximum-entropy bound.

However, this direct approach presents a substantial practical challenge: **computational cost**. The evaluation of the mixture density term, $\log q_{\mathrm{mix}}(z_{i,s})$, requires summing over all $n$ component densities for every single sample point $z_{i,s}$. This leads to a total computational complexity of approximately $O(n \cdot m \cdot n) = O(n^2 m)$, which is often intractable for large datasets. Furthermore, this estimator can suffer from high variance, a common issue with Monte Carlo methods, potentially requiring a large number of samples $m$ for a reliable estimate. Consequently, a choice must be made between two different sources of estimation error: the variance of this direct estimator, versus the bias of the computationally cheaper, approximation-based upper bound. Determining which source of error is more acceptable for a given application is a non-trivial issue.

For our specific case with Gaussian components, each $q_j(z)$ can be evaluated in closed form. The expensive summation term $\log\big(\frac{1}{n} \sum_j q_j(z)\big)$ should be computed using a numerically stable log-sum-exp implementation to mitigate numerical underflow or overflow issues.

This direct Monte Carlo estimator should therefore be viewed as a possible route toward tighter diagnostics, rather than as the estimator used in our main experiments. Although it can avoid the bias introduced by the maximum-entropy upper bound, its $O(n^2 m)$ cost and potential high variance make it impractical for the large-scale correlation analyses reported in this paper. In the current work, we use the computationally cheaper upper bound for reproducibility and scalability, and leave the development of scalable low-variance Monte Carlo or mixture-compression estimators as future work.

*Table 7.* Standard VAE (Fashion-MNIST, CNN): mean $\pm$ standard deviation (aggregated over splits; $n$ after 10% leave-one-out)

| $n$ | Test loss | Train loss | Test reconstruction loss | Train reconstruction loss |
|---|---|---|---|---|
| 900 | $0.3678 \pm 0.0055$ | $0.3497 \pm 0.0051$ | $0.3359 \pm 0.0048$ | $0.3178 \pm 0.0050$ |
| 1763 | $0.3431 \pm 0.0015$ | $0.3332 \pm 0.0023$ | $0.3151 \pm 0.0013$ | $0.3047 \pm 0.0020$ |
| 3107 | $0.3387 \pm 0.0027$ | $0.3315 \pm 0.0038$ | $0.3114 \pm 0.0031$ | $0.3039 \pm 0.0042$ |
| 5477 | $0.3368 \pm 0.0029$ | $0.3303 \pm 0.0035$ | $0.3098 \pm 0.0032$ | $0.3032 \pm 0.0037$ |
| 9655 | $0.3308 \pm 0.0011$ | $0.3258 \pm 0.0016$ | $0.3033 \pm 0.0015$ | $0.2981 \pm 0.0019$ |
| 15317 | $0.3278 \pm 0.0010$ | $0.3230 \pm 0.0014$ | $0.2999 \pm 0.0011$ | $0.2950 \pm 0.0015$ |
| 27000 | $0.3258 \pm 0.0007$ | $0.3226 \pm 0.0009$ | $0.2977 \pm 0.0009$ | $0.2944 \pm 0.0011$ |

*Table 8.* IWAE (MNIST, MLP): mean $\pm$ standard deviation (aggregated over splits; $n$ after 10% leave-one-out)

| $n$ | Test loss | Train loss | Test reconstruction loss | Train reconstruction loss |
|---|---|---|---|---|
| 900 | $5.4509 \pm 0.5314$ | $6.0559 \pm 1.4288$ | $5.4298 \pm 0.5312$ | $6.0335 \pm 1.4286$ |
| 1586 | $6.0934 \pm 0.2946$ | $6.6542 \pm 0.4344$ | $6.0604 \pm 0.2939$ | $6.6206 \pm 0.4339$ |
| 2796 | $5.6460 \pm 0.6224$ | $5.9619 \pm 0.7024$ | $5.6083 \pm 0.6222$ | $5.9240 \pm 0.7021$ |
| 4929 | $3.2625 \pm 0.5939$ | $3.3923 \pm 0.5324$ | $3.2230 \pm 0.5934$ | $3.3527 \pm 0.5318$ |
| 8689 | $1.6936 \pm 0.6031$ | $1.7934 \pm 0.7024$ | $1.6519 \pm 0.6030$ | $1.7518 \pm 0.7021$ |
| 15317 | $0.8025 \pm 0.3017$ | $0.7802 \pm 0.2549$ | $0.7584 \pm 0.3014$ | $0.7360 \pm 0.2545$ |
| 27000 | $0.3883 \pm 0.0920$ | $0.3868 \pm 0.0864$ | $0.3429 \pm 0.0918$ | $0.3414 \pm 0.0863$ |

### K.3. Hierarchical $\beta$-VAE Sweep Experiments

We additionally report a hierarchical counterpart of the $\beta$-sweep experiment. This experiment differs from the standard $\beta$-VAE sweep in Appendix J.6 and from the hierarchical IWAE experiment in Table 12: here, the model is hierarchical, the objective is the ELBO with a varying KL weight $\beta$, and the training size is fixed to $n_{\text{train}} = 30{,}000$. Thus, this experiment directly tests whether the layer-wise quantities in Theorem 3 track changes in the reconstruction generalization gap induced by changing the latent regularization strength.

The architectural settings are the same as the hierarchical ELBO configurations described in Table 5. Specifically, MNIST uses the hierarchical MLP backbone with encoder hidden dimensions [512, 256, 128, 64] and latent dimensions [32, 16, 8, 4], while Fashion-MNIST uses the hierarchical CNN backbone with encoder and decoder channels [32, 64, 64, 64] and latent dimensions [32, 16, 8, 4]. The common optimization settings are those in Table 4: Adam with learning rate $5 \times 10^{-4}$, weight decay $10^{-5}$, batch size 128, maximum 100 epochs, and early stopping patience 10. For each dataset, we evaluate

$$\beta \in \{0.01, 0.03, 0.3, 1.0, 3.0, 10.0\}.$$

Each value of $\beta$ is evaluated using the same 10-fold split-wise protocol with random seed 42, and the test size is 10,000.

For each dataset, $\beta$, split, and layer $\ell$, we compute the reconstruction generalization gap $|L_{\text{test}}^{\text{recon}} - L_{\text{train}}^{\text{recon}}|$, the influence-function-based computable proxy for $I(\phi_{1:\ell}; U \mid X^n)$, and the maximum-entropy upper bound for $I(Z_\ell^n; U \mid X^n, \phi_{1:\ell})$. We

*Table 9.* IWAE (Fashion-MNIST, CNN): mean $\pm$ standard deviation (aggregated over splits; $n$ after 10% leave-one-out)

| $n$ | Test loss | Train loss | Test reconstruction loss | Train reconstruction loss |
|---|---|---|---|---|
| 900 | $2.0802 \pm 0.8590$ | $1.9907 \pm 0.8069$ | $2.0025 \pm 0.8516$ | $1.9096 \pm 0.7943$ |
| 1586 | $2.8367 \pm 0.2620$ | $3.0171 \pm 0.4348$ | $2.7956 \pm 0.2625$ | $2.9749 \pm 0.4346$ |
| 2796 | $1.9683 \pm 0.2818$ | $2.0150 \pm 0.3944$ | $1.9305 \pm 0.2818$ | $1.9763 \pm 0.3943$ |
| 4929 | $1.0996 \pm 0.3362$ | $1.1292 \pm 0.3842$ | $1.0642 \pm 0.3362$ | $1.0935 \pm 0.3841$ |
| 8689 | $0.6213 \pm 0.1176$ | $0.6107 \pm 0.1049$ | $0.5856 \pm 0.1179$ | $0.5748 \pm 0.1052$ |
| 15317 | $0.4823 \pm 0.0399$ | $0.4769 \pm 0.0348$ | $0.4464 \pm 0.0401$ | $0.4408 \pm 0.0352$ |
| 27000 | $0.4278 \pm 0.0266$ | $0.4259 \pm 0.0318$ | $0.3912 \pm 0.0265$ | $0.3891 \pm 0.0318$ |

*Table 10.* IWAE (MNIST, Hierarchical, MLP): mean $\pm$ standard deviation (aggregated over splits; $n$ after 10% leave-one-out)

| $n$ | Test loss | Train loss | Test reconstruction loss | Train reconstruction loss |
|---|---|---|---|---|
| 900 | $6.4940 \pm 0.3735$ | $6.7420 \pm 1.2147$ | $5.8868 \pm 0.3701$ | $6.1170 \pm 1.2270$ |
| 1586 | $6.6945 \pm 0.5423$ | $6.8002 \pm 0.6169$ | $5.7409 \pm 0.5498$ | $5.8480 \pm 0.6226$ |
| 2796 | $7.5624 \pm 0.6805$ | $7.6568 \pm 0.6905$ | $6.5050 \pm 0.6714$ | $6.6014 \pm 0.6767$ |
| 4929 | $6.4859 \pm 0.6429$ | $6.4424 \pm 0.7168$ | $5.3827 \pm 0.6252$ | $5.3398 \pm 0.7033$ |
| 8689 | $3.8671 \pm 0.7693$ | $3.8652 \pm 0.7466$ | $2.6734 \pm 0.7843$ | $2.6738 \pm 0.7610$ |
| 15317 | $2.2513 \pm 0.4346$ | $2.2598 \pm 0.4195$ | $1.0334 \pm 0.4240$ | $1.0413 \pm 0.4107$ |
| 27000 | $1.7819 \pm 0.1375$ | $1.7867 \pm 0.1046$ | $0.5067 \pm 0.1453$ | $0.5092 \pm 0.1104$ |

*Table 11.* IWAE (Fashion-MNIST, Hierarchical, CNN): mean $\pm$ standard deviation (aggregated over splits; $n$ after 10% leave-one-out)

| $n$ | Test loss | Train loss | Test reconstruction loss | Train reconstruction loss |
|---|---|---|---|---|
| 900 | $2.4870 \pm 0.6161$ | $2.9538 \pm 1.1893$ | $1.7973 \pm 0.6195$ | $2.2549 \pm 1.1807$ |
| 1586 | $3.1713 \pm 0.4093$ | $2.9677 \pm 0.5010$ | $2.2877 \pm 0.4063$ | $2.0695 \pm 0.5019$ |
| 2796 | $2.6761 \pm 0.7173$ | $2.6209 \pm 0.5837$ | $1.7635 \pm 0.7170$ | $1.6983 \pm 0.5828$ |
| 4929 | $2.0733 \pm 0.3539$ | $2.0019 \pm 0.2606$ | $1.1514 \pm 0.3645$ | $1.0761 \pm 0.2744$ |
| 8689 | $1.5953 \pm 0.1467$ | $1.5654 \pm 0.1397$ | $0.6008 \pm 0.1587$ | $0.5660 \pm 0.1511$ |
| 15317 | $1.4293 \pm 0.0641$ | $1.4186 \pm 0.0657$ | $0.4149 \pm 0.0561$ | $0.3991 \pm 0.0489$ |
| 27000 | $1.4434 \pm 0.0719$ | $1.4524 \pm 0.0674$ | $0.3915 \pm 0.0592$ | $0.3957 \pm 0.0596$ |

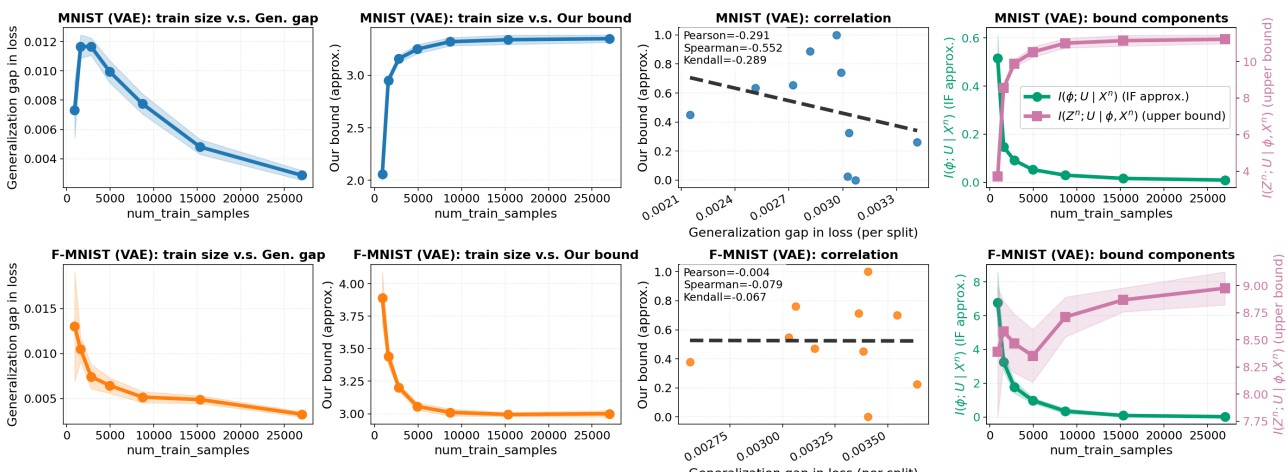

*Figure 4.* Visualization of generalization behavior and our bound for the standard VAE (trained with ELBO) on MNIST (top row) and Fashion-MNIST (bottom row). From left to right, each row shows: (1) the empirical generalization gap versus training set size $n$; (2) our information-theoretic bound on the generalization gap versus $n$; (3) the per-split correlation between the gap and our bound at the largest training size; and (4) the decomposition of our bound into its two components, $I(\phi; U | X^n)$ and $I(Z^n; U | \phi, X^n)$, as a function of $n$.

also report the layer-wise combined quantity $\sqrt{I(\phi_{1:\ell}; U \mid X^n) + I(Z^n_\ell; U \mid X^n, \phi_{1:\ell})}$ and the minimum of this combined quantity over layers. No min–max normalization is applied. The correlations in Table 13 are computed after pooling the $6 \times 10 = 60$ split-wise points for each dataset.

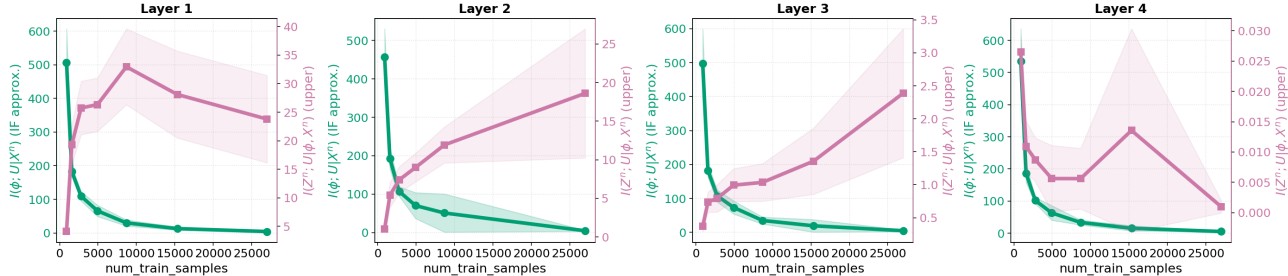

*Figure 5.* Layer-wise analysis of the MI bound components for the Hierarchical VAE (H-IWAE) on the Fashion-MNIST dataset. Each panel corresponds to a layer of the hierarchy, from $l = 1$ (left) to $l = 4$ (right). The plots illustrate the behavior of the parameter information term, $I(\phi; U | X^n)$ (green, left axis), and the latent variable information term, $I(Z^n; U | \phi, X^n)$ (pink, right axis), as a function of the training set size $n$. For each layer $l$, the parameter term is calculated with respect to the parameters influencing the hierarchy up to that specific layer.

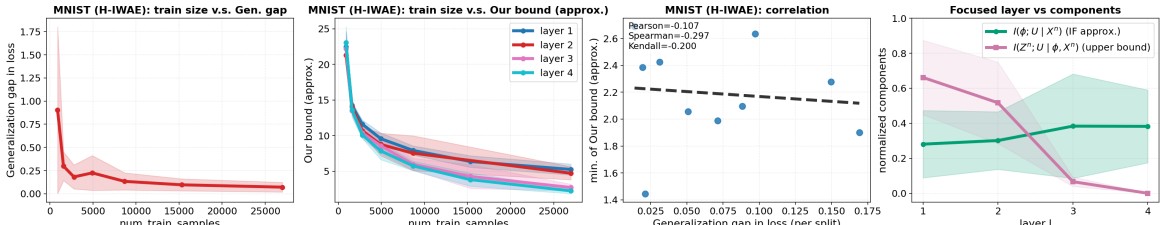

*Figure 6.* Visualization of generalization behavior and our bound for hierarchical IWAE on MNIST. From left to right: (1) generalization gap in loss vs. number of training samples (mean $\pm$ std.), (2) $\sqrt{I(Z^n; U | \phi, X^n) + I(\phi; U | X^n)}$ in Theorem 2 vs. training size (mean $\pm$ std.), (3) per-split correlation at $n = 30000$ between generalization gap and the normalized $\sqrt{I(Z^n; U | \phi, X^n) + I(\phi; U | X^n)}$ including Pearson/Spearman/Kendall coefficients, and (4) bound components: $I(Z^n; U | \phi, X^n)$ and $I(\phi; U | X^n)$ per selection of $l$. Note that the values of our bound in (3) and (4) are normalized.

*Table 12.* Correlation between generalization gap in loss and our bounded components at $n = 30000$ under the IWAE experiments (Hierarchical VAE).

| Dataset | Component | Pearson | Spearman | Kendall |
|---------|-----------|---------|----------|---------|
| MNIST | $I(\phi; U \mid X^n)$ | **0.221** | **0.200** | **0.156** |
| MNIST | $I(Z^n; U \mid \phi, X^n)$ | -0.487 | -0.648 | -0.511 |
| MNIST | $\sqrt{I(Z^n; U \mid \phi, X^n) + I(\phi; U \mid X^n)}$ | -0.107 | -0.297 | -0.200 |
| F-MNIST | $I(\phi; U \mid X^n)$ | 0.543 | **0.261** | **0.244** |
| F-MNIST | $I(Z^n; U \mid \phi, X^n)$ | -0.056 | -0.273 | -0.200 |
| F-MNIST | $\sqrt{I(Z^n; U \mid \phi, X^n) + I(\phi; U \mid X^n)}$ | **0.544** | **0.261** | **0.244** |

*Table 13.* Correlation between the reconstruction generalization gap and the layer-wise computable information-theoretic quantities under the hierarchical $\beta$-VAE sweep. For each dataset, correlations are computed by pooling $6 \times 10 = 60$ split-wise points over $\beta \in \{0.01, 0.03, 0.3, 1.0, 3.0, 10.0\}$.

| Dataset | Layer | Diagnostic quantity | Pearson | Spearman | Kendall |
|---|---|---|---|---|---|
| MNIST | 1 | $I(\phi_{1:1}; U \mid X^n)$ | 0.462 | 0.369 | 0.253 |
| MNIST | 1 | $I(Z_1^n; U \mid X^n, \phi_{1:1})$ | 0.415 | 0.366 | 0.256 |
| MNIST | 1 | $\sqrt{I(\phi_{1:1}; U \mid X^n) + I(Z_1^n; U \mid X^n, \phi_{1:1})}$ | 0.479 | 0.366 | 0.256 |
| MNIST | 2 | $I(\phi_{1:2}; U \mid X^n)$ | 0.466 | 0.359 | 0.244 |
| MNIST | 2 | $I(Z_2^n; U \mid X^n, \phi_{1:2})$ | 0.348 | 0.193 | 0.122 |
| MNIST | 2 | $\sqrt{I(\phi_{1:2}; U \mid X^n) + I(Z_2^n; U \mid X^n, \phi_{1:2})}$ | 0.435 | 0.228 | 0.150 |
| MNIST | 3 | $I(\phi_{1:3}; U \mid X^n)$ | 0.471 | 0.365 | 0.246 |
| MNIST | 3 | $I(Z_3^n; U \mid X^n, \phi_{1:3})$ | 0.353 | 0.535 | 0.369 |
| MNIST | 3 | $\sqrt{I(\phi_{1:3}; U \mid X^n) + I(Z_3^n; U \mid X^n, \phi_{1:3})}$ | 0.406 | 0.553 | 0.385 |
| MNIST | 4 | $I(\phi_{1:4}; U \mid X^n)$ | 0.471 | 0.365 | 0.246 |
| MNIST | 4 | $I(Z_4^n; U \mid X^n, \phi_{1:4})$ | 0.151 | 0.436 | 0.296 |
| MNIST | 4 | $\sqrt{I(\phi_{1:4}; U \mid X^n) + I(Z_4^n; U \mid X^n, \phi_{1:4})}$ | 0.529 | 0.365 | 0.242 |
| MNIST | $\min_\ell$ | $\min_\ell \sqrt{I(\phi_{1:\ell}; U \mid X^n) + I(Z_\ell^n; U \mid X^n, \phi_{1:\ell})}$ | 0.529 | 0.365 | 0.242 |
| Fashion-MNIST | 1 | $I(\phi_{1:1}; U \mid X^n)$ | 0.761 | 0.660 | 0.440 |
| Fashion-MNIST | 1 | $I(Z_1^n; U \mid X^n, \phi_{1:1})$ | 0.928 | 0.764 | 0.567 |
| Fashion-MNIST | 1 | $\sqrt{I(\phi_{1:1}; U \mid X^n) + I(Z_1^n; U \mid X^n, \phi_{1:1})}$ | 0.952 | 0.761 | 0.560 |
| Fashion-MNIST | 2 | $I(\phi_{1:2}; U \mid X^n)$ | 0.761 | 0.659 | 0.436 |
| Fashion-MNIST | 2 | $I(Z_2^n; U \mid X^n, \phi_{1:2})$ | 0.382 | 0.741 | 0.521 |
| Fashion-MNIST | 2 | $\sqrt{I(\phi_{1:2}; U \mid X^n) + I(Z_2^n; U \mid X^n, \phi_{1:2})}$ | 0.837 | 0.738 | 0.513 |
| Fashion-MNIST | 3 | $I(\phi_{1:3}; U \mid X^n)$ | 0.761 | 0.657 | 0.433 |
| Fashion-MNIST | 3 | $I(Z_3^n; U \mid X^n, \phi_{1:3})$ | 0.371 | 0.710 | 0.488 |
| Fashion-MNIST | 3 | $\sqrt{I(\phi_{1:3}; U \mid X^n) + I(Z_3^n; U \mid X^n, \phi_{1:3})}$ | 0.873 | 0.701 | 0.476 |
| Fashion-MNIST | 4 | $I(\phi_{1:4}; U \mid X^n)$ | 0.761 | 0.656 | 0.432 |
| Fashion-MNIST | 4 | $I(Z_4^n; U \mid X^n, \phi_{1:4})$ | 0.896 | 0.750 | 0.558 |
| Fashion-MNIST | 4 | $\sqrt{I(\phi_{1:4}; U \mid X^n) + I(Z_4^n; U \mid X^n, \phi_{1:4})}$ | 0.876 | 0.708 | 0.486 |
| Fashion-MNIST | $\min_\ell$ | $\min_\ell \sqrt{I(\phi_{1:\ell}; U \mid X^n) + I(Z_\ell^n; U \mid X^n, \phi_{1:\ell})}$ | 0.876 | 0.708 | 0.486 |

