# OpenReview forum: "Information-Theoretic Generalization Bounds for VAEs: A Role of Encoder and Latent Variable"
_ICML.cc/2026/Conference — ICML 2026 regular_

### Official Review · Reviewer_RsDS · 2026-02-16

**Soundness:** 2
**Presentation:** 3
**Significance:** 3
**Originality:** 3
**Overall Recommendation:** 4
**Confidence:** 3

**Summary:**

In this work, the authors study the generalization error of Variational Autoencoders (VAEs) with continuous latent variables from an information-theoretic perspective. Under mild assumptions, they derive theoretical results that bound the generalization error using decoder-independent mutual information terms. They also extend the analysis to hierarchical VAEs.

**Compliance With Llm Reviewing Policy:**

Affirmed.

**Final Justification:**

The author’s rebuttal clarified most of my concerns. Some of them are structural (i.e., looseness of the derived bounds), but they do not prevent acceptance of the paper.

**Key Questions For Authors:**

I have no additional questions for the authors beyond requesting further elaboration on Weaknesses 1, 2, and 3.

**Limitations:**

yes

**Strengths And Weaknesses:**

**Strengths**

1) The theoretical analysis is rigorous.
2) One key takeaway (that the generalization error is decoder-independent) is notable and interesting.
3) The paper is detailed and well-written.

**Weaknesses**

1) All the theoretical results are derived considering the Mean Squared Error (MSE) reconstruction loss. I view this as a significant limitation for two reasons: a) this is not the loss function optimized during training, and b) even within the Evidence Lower Bound (ELBO), the reconstruction loss is not MSE. While I understand this may have simplified the theoretical analysis, given that the focus is on generalization error, I believe a stronger alignment with the training objectives is important.

2) There is no discussion of the bound’s tightness, either theoretically or numerically. Since the bound is obtained through several approximation steps, I believe more attention should have been given to discussing or empirically demonstrating its tightness. Do you have results for simple cases? Do you have some empirical evidence?

3) Some empirical results (e.g., Table 2) appear somewhat weak. The computed correlation is not very high, and it drops considerably when moving from an easy dataset (MNIST) to a bit more challenging one (Fashion-MNIST). In this regard, what happens if you use (a portion of) ImageNet?

**Minor**

1) The provided code link (https://github.com/ANONYMOUS) does not work.
2) Several equations are split between normal text and equation mode. See, for instance, lines 183–183, 250–252, 410–411, etc.

---

> ### Author Rebuttal · Authors · 2026-03-29
>
> We sincerely thank you for the highly constructive feedback and for meticulously reviewing our manuscript.
>
> ## Weakness 1: Focus on MSE vs. the Full ELBO Objective
> Our primary theoretical objective is to establish a decomposition of reconstruction generalization into latent complexity and encoder complexity. Therefore, while VAEs are trained using the ELBO, we analytically focus on the expected reconstruction loss (specifically, the squared Euclidean loss) to evaluate generalization performance. A complete generalization analysis of the entire ELBO objective remains an open problem.
>
> This specific focus is fundamentally tied to our proof framework, which relies heavily on a **boundedness assumption (Assumption 1)**. This ensures a uniformly bounded reconstruction loss, required to evaluate exponential moments using McDiarmid's inequality (Appendix D.2). If a log-loss is uniformly bounded, our proof strategy can be extended to Gaussian/Bernoulli likelihoods with minor modifications. However, handling strictly **unbounded log-losses requires a completely different proof framework**. We will explicitly discuss this limitation and clarify the scope of our theory in the revised Appendix.
>
> ## Weakness 2: Tightness of Bounds and Empirical Looseness
> We agree our empirical bounds exhibit looseness, explicitly acknowledged as a limitation in Appendix K (the maximum-entropy upper bound for the latent term is coarse).
>
> However, evaluating information-theoretic quantities for continuous VAEs was previously computationally prohibitive, relying on high-dimensional joint distributions (Futami & Fujisawa, 2025) or discrete patterns (Sefidgaran et al., 2023). By utilizing the LOO-CMI framework, our bounds reduce the hold-out index $U$ to 1 dimension. Our central contribution is the **theoretical decomposition structure itself**, making computable proxies and empirical diagnostics for continuous VAEs possible for the first time. Developing tighter estimators (e.g., the $O(n^2m)$ Monte Carlo estimator in Appendix K.2) is a crucial next step. We interpret the current looseness as an **immature estimation layer**, rather than an invalidation of the theoretical decomposition.
>
> ## Weakness 3: Weak Correlations and More Complex Datasets
> We acknowledge our initial experiments were primarily a controlled study. While evaluating highly complex datasets like an ImageNet subset is computationally prohibitive within the rebuttal period, we completely agree on the necessity of validating our bounds on more complex data.
>
> To explicitly address this, we conducted new experiments on the **CIFAR-10 dataset using a CNN-VAE**, alongside a controlled analysis **varying the hyperparameter $\beta$ ($\beta$-VAE)**.
>
> As detailed in **our response to Reviewer t4XQ** (where we provide full numerical results), our theoretical quantities not only maintain meaningful positive correlations on CIFAR-10, but also exhibit **strong correlations (Pearson > 0.9)** across both datasets when the latent capacity is systematically controlled via $\beta$.
>
> These new results provide a deeper diagnostic view of the bounds under increased dataset complexity and controlled latent capacities. We believe these additions significantly strengthen the empirical grounding of our theoretical decomposition and will include them in the revision.
>
> ## Minor Comments
> - **Code Link:** We intentionally anonymized the URL to strictly maintain the double-blind review process. The source code is provided in the supplementary material, and we will replace the anonymized link with the actual public GitHub repository upon acceptance.
> - **Split Equations:** We sincerely apologize for the formatting issues (e.g., lines 183–183, 250–252). We will thoroughly review the manuscript and ensure all mathematical expressions are properly enclosed in equation mode.

---

> > ### Author Rebuttal · Reviewer_RsDS · 2026-04-02
> >
> > I'm satisfied with the answers and raise my score. Thank you.

---

### Official Review · Reviewer_VHDL · 2026-03-10

**Soundness:** 3
**Presentation:** 4
**Significance:** 3
**Originality:** 3
**Overall Recommendation:** 5
**Confidence:** 3

**Summary:**

The paper presents information theoretic generalization bounds for VAEs. Using the leave-one-out setting, the authors derive an upper bound on the generalization error of the reconstruction loss, dependent on the diameter of the data space and the information content of the encoder and latent variables. Then, they extend the result to hierarchical VAEs and show that the generalization gap is not cumulative wrt to the layers. The authors also derive an upper bound on the Wasserstein-2 distance between the true distribution and the one learned by the VAE. Numerical experiments were done to show correlation between the empirically observed generalization gap and and the estimation of the right-hand side of Theorem 2.

**Compliance With Llm Reviewing Policy:**

Affirmed.

**Final Justification:**

I think the paper meets the standard for acceptance. I maintain my score and recommend acceptance.

**Key Questions For Authors:**

- In the experiments, how do you estimate the first term of the generalization gap (the test loss)? Do you take as test set all the samples not chosen for training?

- I find it intriguing that the generalization gap doesn't depend on the decoder. Imagine you have a decoder that ignore any input z, and just returns a random sample x (or the same x for every z). In that case, the generalization gap would be $0$, but if the encoder overfits, then the bound would be very large. So by incorporating the decoder into the result, one might get a tighter or more informative bound, since it can mitigate overfitting from the encoder. Have the authors explored such ideas? I believe a discussion about the decoder would be interesting in the paper.

**Limitations:**

yes, the authors discussed the limitations of the work.

**Strengths And Weaknesses:**

Strengths:
- The paper is very well written and all the concepts are defined and explained properly. I really appreciated the experimental details and additional explanations in the appendix. The limitations were also explained well.

- The topic is interesting and relevant, and the authors do a good job of motivating the work.

- The theoretical content of the paper seems sound, although I did not verify the proofs, the approach and explanations make sense. The variables are defined clearly and the theoretical results discussed. The assumptions are mostly mild and reasonable. Overall, I enjoyed reading this paper.

Weaknesses:
- The results hold under the finite diameter assumption. Although the assumption is reasonable, the dependency of the bounds on the diameter makes the bounds very loose for any complex datasets.

- The bounds are in expectation, not with high probability. This is a limitation in practice because the true risk being bounded is the expectation wrt to the training sample, as opposed to having a high-probability bound for a given training set. Moreover, the terms on the right-hand side intractable. I understand these are mostly limitations of MI-based generalization bounds in general.

- The experiments show some correlation between the empirical generalization gap and the bound (which is expected), but they also highlight how loose the bound is, even for relatively simple datasets.

Typos:
- line 350 left: $X_m$, should be $X_i$
- Figure 1 description: there are typos (top, bottom). I assume the authors changed the figure configuration and forgot to change the description.

---

> ### Author Rebuttal · Authors · 2026-03-29
>
> We sincerely thank the reviewer for the detailed, rigorous, and highly constructive feedback.
>
> ## Q.1: Estimation of the Generalization Gap
>
> We apologize for the lack of clarity. Theoretically, Section 2.2 defines generalization error as population loss (over $X \sim D$) minus training error. Experimentally, since true population loss is inaccessible, we approximate it using 10-fold cross-validation (Appendix J.1). For each fold, we compute the empirical gap ($L_{\text{test}} - L_{\text{train}}$) on the strictly held-out block. Reported values are the mean and standard deviation across these 10 splits. We will make this distinction explicit in the revision.
>
> ## Q.2: Justification of the "Decoder-Independent" Bound
>
> "Decoder-independent" does not imply the decoder is irrelevant to overall performance. Total performance decomposes as: $\text{Test Loss} = \text{Training Loss} + \text{Generalization Gap}$. Theorem 2 strictly bounds the generalization gap, not absolute test loss. A poor decoder worsens both training and test losses uniformly, degrading overall performance even if the gap is small.
>
> While incorporating the decoder might tighten the total test loss bound, it would entangle the contributions of the encoder, latent variables, and decoder. By abstracting the decoder, we successfully isolate the impacts of latent complexity and encoder data-dependence. The bound evaluates the decoder's worst-case impact: if the encoder is regularized and latent information restricted, the gap remains controlled regardless of decoder complexity. We will clarify this in the revision.
>
> ## Weakness 1: Finite Diameter Assumption
>
> We acknowledge this limitation. Assumption 1 (finite diameter) ensures a bounded reconstruction loss, necessary for computing bounded differences (e.g., McDiarmid's inequality) in the LOO-CMI framework. For complex datasets, the diameter-dependent constant may grow and loosen the bound. However, this affects the constant's tightness, not the fundamental decomposition into encoder overfitting and latent complexity. Theoretically, this could be replaced directly by a bounded loss assumption.
>
> ## Weakness 2: Expectation Bound and Intractability
>
> We agree our theorems provide expectation bounds and the exact mutual information terms remain intractable. However, our main novelty is establishing the first structural, decoder-independent information-theoretic decomposition for continuous VAEs. We do not claim a fully tractable guarantee; rather, our computable proxies (Sec 3.4, App J–K) offer a first step toward an empirical diagnostic tool. High-probability bounds and tighter estimators are critical future steps.
>
> ## Weakness 3: Looseness in Experiments
>
> We agree our empirical evaluation is a proof-of-concept and exhibits looseness, primarily due to the coarse maximum-entropy upper bound for the latent term (acknowledged in Appendix K).
>
> Historically, evaluating these quantities for continuous VAEs was computationally prohibitive, relying on high-dimensional joint distributions or discrete patterns (Futami & Fujisawa, 2025; Sefidgaran et al., 2023). Our LOO-CMI framework reduces the hold-out index $U$ to 1 dimension, making numerical diagnosis feasible. The tighter Monte Carlo estimator (App K.2) requires $O(n^2m)$ computation. Thus, the looseness reflects an immature estimation layer, not an invalidation of the theoretical decomposition.
>
> To address concerns about complex datasets and stress-test our theory, we conducted new controlled experiments varying the latent capacity ($\beta$-VAEs) on both MNIST and CIFAR-10 datasets. As detailed in our response to Reviewer t4XQ, because $\beta$ explicitly regularizes latent capacity, the latent term robustly tracks the gap. Consequently, our combined theoretical bound exhibits exceptionally strong positive correlations (Pearson > 0.9) with the empirical gap across both datasets. We will include these comprehensive results in the revision to significantly strengthen our empirical grounding and demonstrate the validity of our decomposition.
>
> ## Regarding typos
>
> We have corrected all the typos you pointed out. Thank you for your careful reading.

---

> > ### Author Rebuttal · Reviewer_VHDL · 2026-04-01
> >
> > Thank you for your response. I am happy with the answers and I will maintain my score.

---

### Official Review · Reviewer_t4XQ · 2026-03-11

**Soundness:** 3
**Presentation:** 4
**Significance:** 3
**Originality:** 2
**Overall Recommendation:** 5
**Confidence:** 4

**Summary:**

The authors present an information-theoretic (IT) generalization analysis for standard Variational Autoencoders (VAEs) with continuous latent variables. Utilizing the leave-one-out conditional mutual information (LOO-CMI) framework, the paper demonstrates that the generalization error can be bounded by the information complexity of the encoder and the latent variables, independently of the decoder. To circumvent the challenges of continuous latent spaces, the authors introduce a data-independent truncation-and-discretization argument. Furthermore, the framework is extended to hierarchical VAEs to yield layer-wise bounds. Lastly, the authors provide a finite-sample population-level bound on the 2-Wasserstein distance between the true and generated data distributions.

**Compliance With Llm Reviewing Policy:**

Affirmed.

**Final Justification:**

Overall, this paper provides a theoretically solid and elegantly presented (Presentation: 4) information-theoretic generalization analysis for continuous VAEs. While the mathematical tools themselves synthesize existing techniques (Originality: 2), deriving decoder-independent bounds and extending them to hierarchical VAEs represents a highly valuable and significant contribution to the generative modeling community (Significance: 3).

My initial reservations primarily centered on the paper's empirical soundness (Soundness: 3). The original experimental section was too tight, relied on loose estimators, and lacked stress-testing under varied capacities or complex datasets. Furthermore, the graphical dependencies in the methodology required clarification.

The authors' rebuttal and our subsequent discussions successfully addressed all of my main concerns. By expanding the empirical evaluation to include $\beta$-VAE sweeps on CIFAR-10 and Fashion-MNIST, they better demonstrated the practical behavior of their bounds. Most importantly, the authors provided a rigorous mathematical clarification of their graphical model (Figure 2b) and, during the discussion phase, committed to including crucial scatter and trend plots to transparently validate their correlation claims regarding the pooled $\beta$-VAE data.

Because the authors engaged constructively, clarified the theoretical ambiguities, and significantly strengthened the empirical validation to match the high quality of their theory, the rebuttal positively changed my evaluation. I have raised my score and recommend this paper for acceptance.

**Key Questions For Authors:**

1. **Regarding Figure 2:** I have two points of clarification regarding the dependencies shown:
    * In Figure 2(a), why does $X^n$ directly affect $\hat{X}^n$?
    * In Figure 2(b), why do $X^n$ and $U$ directly affect $Z_1^n$?
2. Could you elaborate on lines 315-321 (left column)? The explanation of the results and the methodology in this specific paragraph is currently a bit difficult to follow.
3. The empirical evaluations reveal that the upper bound for $I(Z^{n};U|\phi,X^{n})$ becomes looser and dominates the total bound, leading to poor correlation with the generalization gap. Could the Monte Carlo estimation approach discussed in Appendix K.2 realistically be scaled to provide tighter diagnostics without becoming computationally prohibitive?

**Limitations:**

The authors transparently acknowledge their limitations in Section 7 and Appendix K. Specifically, they note that their bounds currently rely on squared-error reconstruction losses and appropriately highlight the severe looseness of the maximum-entropy upper bound used for empirical MI estimation.

**Strengths And Weaknesses:**

### Strengths
1. **Theoretical Novelty:** Extending IT generalization bounds to standard VAEs with continuous latent variables addresses a significant gap in the literature, which previously relied heavily on discrete settings.
2. **Decoder-Independent Bounds:** The bounds successfully isolate the generalization error attributable to the latent variable complexity and the encoder's overfitting, separating these from the decoder's expressiveness.
3. **Hierarchical Extension:** The layer-wise bounds for hierarchical VAEs elegantly explain the structural trade-off between parameter complexity accumulation and latent information compression across layers.

### Weaknesses
1. **Empirical Validation Falls Short:** The experimental section is a bit tight and does not thoroughly stress-test the theoretical bounds under diverse or pathological conditions. The paper would benefit much more from expanded experiments than from having the complete HVAE derivations in the main text. Moving Section 4 to the Appendix (presented as an extension of the VAE case) would free up space to include:
    * **Controlled Information Bottleneck via $\beta$-VAE:** Training with varying $\beta$ values to smoothly control the latent information complexity $I(Z; X)$, empirically validating how the bounds track with explicit regularization constraints.
    * **Sanity Checks via Deliberate Overfitting:** Training on progressively smaller subsets of data or injecting pixel/label noise. The theoretical bounds should predictably expand as the empirical generalization gap widens due to memorization.
2. **Clarity in Results (Lines 315-321 left):** The discussion in this section is currently quite unclear. The authors need to better elaborate on the specific results and exactly what has been done here.
3. **Empirical Estimator Looseness:** The numerical approximations used to evaluate the latent variable complexity term ($I(Z^{n};U|\phi,X^{n})$) are quite loose, resulting in weak or negative correlations with the actual empirical generalization gap.

### Minor Comments and Typos:
1. **Line 86 (left):** I suggest starting a new paragraph with "First" to improve readability.
2. **Line 146 (left):** It would be helpful to briefly explain that this comes from assuming $p(x|z)$ is Gaussian.
3. **Line 147 (left):** Use a lowercase $z$ here.
4. **Line 154 (left):** Should this be $l_0(W, X_n)$?
5. **Line 317 (left):** Typo in the table reference; it should say Table 2.

---

> ### Author Rebuttal · Authors · 2026-03-30
>
> We sincerely thank the reviewer for the constructive feedback.
>
> ## Q.1: Dependency in Figure 2
> In the LOO setting, the training set is $S = X^n \setminus X_U$, so learned parameters explicitly depend on $S$. For Figure 2(a), **all data $X^n$ (train and test) are fed into the encoder** to compute predictions $\hat{X}^n$, creating a direct dependency. This matches the graphical model for the function-CMI bound in Haghifam et al. (2022), detailed in App. F.1.
> Figure 2(b) illustrates the internal layer-wise computation ($X^n \to Z_1^n \to Z_2^n$) to clarify dependencies for the mutual information terms in Theorem 3. We will clarify this in the revision.
>
> ## Q.2: Clarification in Lines 315–321
> Following Appendix J, we trained models from scratch using 10-fold cross-validation. For Table 2 ($n=30,000$), we **calculated a single scalar for four metrics per split**:
> - The absolute value of the empirical generalization gap in loss (i.e., $|L_{\text{test}} - L_{\text{train}}|$).
> - The influence-function-based proxy for $I(\phi;U|X^n)$.
> - The computable upper bound of $I(Z^n;U|\phi,X^n)$, evaluated under a uniform prior for $U$.
> - The combined term: $\sqrt{I(Z^n;U|\phi,X^n)+I(\phi;U|X^n)}$, which was subsequently min-max normalized.
>
> Correlations were calculated **strictly across these 10 split-wise values**. Section 3.4 aims for qualitative diagnosis, not strict numerical tightness. We will explicitly document this.
>
> ## Q.3: Practicality of Monte Carlo Estimation in Appendix K.2
> While the mixture MC estimator (Appendix K.2) could be tighter, we do not consider it practical for large-scale settings. Evaluating it requires summing all component densities, leading to a **computational complexity of $O(n^2 m)$**. Mitigating high variance requires a large sample size $m$. Facing a tractability-tightness trade-off, we opted for the coarser maximum-entropy bound. We will clarify that Appendix K.2 is a promising future direction, not a ready-to-use estimator.
>
> ## Weakness: Empirical Evaluation & $\beta$-VAE
> We acknowledge the latent-term estimator's looseness (Appendix K). However, our core contribution is the theoretical decomposition itself.
>
> To address concerns about evaluating solely on MNIST and to stress-test our theory, we evaluated a **CNN-based VAE** on the more complex **CIFAR-10** dataset alongside MNIST.
> We systematically varied latent capacity (**$\beta$-VAE**, sweeping $\beta \in \{0.01, 0.03, 0.3, 1, 3, 10\}$ across 60 pooled models). Because $\beta$ explicitly regularizes latent capacity, the latent term ($ZU: I(Z^n ; U \mid \phi, X^n)$) dominates and robustly tracks the gap. While the encoder proxy ($IF: I(\phi ; U \mid X^n)$) behaves differently depending on dataset complexity, our combined bound ($\sqrt{IF+ZU}$) effectively captures the overall dynamic, exhibiting **exceptionally strong positive correlations (Pearson > 0.9)** across both settings:
>
> | Dataset ($\beta$-sweep) | Component | Pearson $\uparrow$ | Spearman $\uparrow$ | Kendall $\uparrow$ |
> |:---|:---|:---:|:---:|:---:|
> | **MNIST** | $I(\phi ; U \mid X^n)$ | 0.458 | 0.611 | 0.400 |
> | | $I(Z^n ; U \mid \phi, X^n)$ | **0.931** | 0.792 | 0.550 |
> | | $\sqrt{I(Z^n;U\mid \phi,X^n)+I(\phi;U\mid X^n)}$ | **0.934** | **0.799** | **0.553** |
> | **CIFAR-10** | $I(\phi ; U \mid X^n)$ | -0.328 | -0.864 | -0.680 |
> | | $I(Z^n ; U \mid \phi, X^n)$ | **0.912** | **0.923** | **0.765** |
> | | $\sqrt{I(Z^n;U\mid \phi,X^n)+I(\phi;U\mid X^n)}$ | **0.921** | **0.889** | **0.724** |
>
> We confirmed **similar strong correlations on Fashion-MNIST**, and will expand this $\beta$-sweep to **Hierarchical VAEs (HVAEs)** in the revision. These results perfectly validate that our quantities robustly capture capacity-induced generalization changes. Regarding HVAEs, we prefer keeping Theorem 3 in the main text as our **second major contribution**.
>
> ## Minor Comments and Typos
> We revised the highlighted sentences and corrected all pointed-out typos. Thank you for your careful reading and constructive suggestions.

---

> > ### Author Rebuttal · Reviewer_t4XQ · 2026-04-03
> >
> > **Thank you to the authors for the detailed rebuttal and the additional experiments.** The new $\beta$-VAE results on CIFAR-10 and Fashion-MNIST are appreciated and certainly help address my concerns regarding the empirical evaluation. The clarifications regarding the experimental setup and the Monte Carlo estimation are also helpful.
> >
> > However, I still have a couple of remaining questions and suggestions for the final version of the paper:
> >
> > **1. Figure 2(b) Dependency:**
> > While your explanation for the dependencies in Figure 2(a) makes sense, my specific question regarding Figure 2(b) was not fully addressed. Could you please explicitly clarify why $U$ directly affects $Z_1^n$ in this specific graphical model?
> >
> > **2. Clarification on the $\beta$-VAE Results Table:**
> > Regarding the new table evaluating the $\beta$-VAE sweep, it is not entirely clear how the correlation metrics were computed. Are the reported coefficients (e.g., Pearson > 0.9) calculated by pooling all 60 models across all values of $\beta$ into a single correlation calculation, or were the correlations computed per $\beta$ value and then averaged?
> >
> > * **Suggestion:** Rather than just reporting a single aggregated scalar in a table, the paper would benefit significantly from a plot showing the evolution of the correlation (e.g., the Pearson coefficient) with respect to the different values of $\beta$. Alternatively, a scatter plot of the empirical gap versus the theoretical bound across the different $\beta$ values would be very informative. Visualizing this dynamic will make the claim that the latent term "robustly tracks the gap" much more convincing to the reader.

---

> > > ### Author Response · Authors · 2026-04-05
> > >
> > > We sincerely appreciate your follow-up and the time you have taken to engage deeply with our work.
> > >
> > > ## Response to Q1: Direct influence of $U$ on $Z_1^n$ in Figure 2(b)
> > > We agree that the distinction between Figure 2(a) and 2(b) was not sufficiently clear, and we apologize for the confusion. However, this exact difference corresponds to the technique of applying supersample-based shuffling *directly to the latent variables*, which was introduced by Futami & Fujisawa (2025) for VQ-VAEs.
> > >
> > > This distinction becomes clearest when writing out the full joint distributions for the posterior and prior compared in the Donsker-Varadhan representation. For simplicity, let $L=1$ and $X_U^{n-1} := X^n \setminus \{X_U\}$.
> > >
> > > In Figure 2(a), corresponding to the conventional LOOCV supersample setting, $U$ is used to shuffle the train/test split globally for the entire parameter/prediction block. As seen in the fCMI LOOCV bound in Appendix F.1 (Line 1217), the posterior distribution used in the Donsker-Valadhan is:
> > > $$Q_a^{\mathrm{post}} = P(X^n)p(U) \mathbb{E}\_{q(\phi,\theta \mid X_U^{n-1})}[\mathbb{E}\_{q(Z^n \mid \phi,X_U,X_U^{n-1})}p(\hat{X}^n \mid \theta,Z^n)]$$
> > > The corresponding prior marginalizes $U$ globally on the prediction side:
> > > $$Q_a^{\mathrm{prior}} = P(X^n)p(U)\bar{q}(\hat{X}^n \mid X^n)$$
> > > where $\bar{q}(\hat{X}^n \mid X^n) := \mathbb{E}\_{p(u)} \mathbb{E}\_{q(\phi,\theta \mid X_{u}^{n-1})}[\mathbb{E}\_{q(Z^n \mid \phi,X_{u},X_{u}^{n-1})}p(\hat{X}^n \mid \theta,Z^n)]$. Here, changing $U$ switches not only the training set used for parameter learning but also the input assignment for constructing the prediction $\hat{X}^n$ altogether.
> > >
> > > In contrast, in Figure 2(b), corresponding to our method, $U$ is shuffled *only for the latent variables* themselves. The posterior is:
> > > $$Q_b^{\mathrm{post}} = P(X^n)p(U) q(\phi,\theta \mid X_U^{n-1}) q(Z^n \mid \phi,X_U,X_U^{n-1}) p(\hat{X}^n \mid \theta,Z^n)$$
> > >
> > > The essential difference lies in the prior:
> > > $$Q_b^{\mathrm{prior}} = P(X^n)p(U) q(\phi,\theta \mid X_U^{n-1}) \bar{q}(Z^n \mid \phi,X^n) p(\hat{X}^n \mid \theta,Z^n)$$
> > > where $\bar{q}(Z^n \mid \phi,X^n) := \mathbb{E}\_{p(u')} q(Z^n \mid \phi,X_{u'},X_{u'}^{n-1})$ (This is described in Eq.(14) in Line 931). Crucially, the marginalization of $U$ is performed *only* on the hidden-variable side $Z^n$, and not on the parameter learning side $q(\phi,\theta \mid X_U^{n-1})$.
> > >
> > > Therefore, Figure 2(b) explicitly separates the influence of $U$ on parameter learning from its influence on the latent variables by shuffling the latent variables $Z^n$ by $U$. This is the reason for the explicit $U \to Z_1^n$ dependency shown in Figure 2(b). Similar figures are depicted in Figure 1 in Futami & Fujisawa (2025).
> > > We will add this detailed explanation comparing the posteriors and priors to the Appendix in the final version to clarify this dependency.
> > >
> > > ## Response to Q2: Calculation of correlation coefficients in the $\beta$-VAE table
> > > The reported Pearson, Spearman, and Kendall coefficients were **not** calculated separately for each $\beta$ and then averaged. Instead, they were calculated once by **pooling all 60 independently trained models** obtained across the entire $\beta$ sweep.
> > >
> > > Specifically, for each run (6 temperatures $\times$ 10 seeds), we computed one pair of (empirical generalization gap, theoretical quantity) and calculated the correlation coefficients across these 60 paired values. Therefore, the large correlation values in the table indicate how well the theoretical quantities track the variation in the empirical gap when the latent capacity is systematically altered by sweeping $\beta$.
> > >
> > > We adopted this pooled approach because the purpose of this $\beta$-VAE experiment was specifically to investigate whether the theoretical quantities can track the induced variation when controlling the latent capacity via $\beta$. In contrast, calculating the correlation for a fixed $\beta$ primarily examines seed-to-seed variability under a fixed capacity, which differs from our main question. Moreover, individual correlation coefficients can be unstable when the number of runs per $\beta$ is limited.
> > >
> > > However, we completely agree with your point that a single aggregated scalar obscures the underlying dynamics. In the final version, we will add the following visualizations:
> > > 1. **Scatter plots:** Displaying the empirical gap versus the theoretical quantities (especially ZU and IF+ZU) across all 60 runs, color-coded by $\beta$.
> > > 2. **Trend plots:** Displaying the mean $\pm$ standard deviation of the gap and theoretical quantities as a function of $\beta$.
> > >
> > > Through these additions, we can more directly show what kind of dynamic relationships along the $\beta$ sweep produce the aggregate correlations reported in the table. We will also moderate the expressions in the text to more accurately state that our results support the idea that the theoretical quantities robustly track the variation in the empirical generalization gap induced by $\beta$.

---

### Official Review · Reviewer_uwfe · 2026-03-11

**Soundness:** 4
**Presentation:** 3
**Significance:** 3
**Originality:** 3
**Overall Recommendation:** 5
**Confidence:** 3

**Summary:**

This paper proposes the novel information theoretic boundary of the generalization gap of VAEs. By applying the results for VQ-VAE in Futami & Fujisawa (2025), authors extend the information boundary equipped with the encoder parameter and the latent variables onto VAEs with continuous latent variables. One consequence of the proposed theorem is that we can do layer-wise analysis of the hierarchical VAE regarding to the generalization of the whole model. The information theoretic boundary on the data generation quality is also proposed.

**Compliance With Llm Reviewing Policy:**

Affirmed.

**Final Justification:**

Authors have adequately addressed my concerns, and I intend to maintain my score.

**Key Questions For Authors:**

1. Please explain more on how the results in Table 2 were obtained.

1. Most of the examples illustrating the claimed theory are based on a VAE-based autoencoders. Can the same result be applied to autoencoders trained with different types of loss? E.g., simple unregularized autoencoders or Wasserstein autoencoders use L2 loss for the reconstruction loss.

1. Can the result of Theorem 4 be practically used in the training stage of generative models to enhance their performance?

**Limitations:**

yes

**Strengths And Weaknesses:**

**Strengths**

- Authors present the information-theoretic bound of generalization error for more generalized VAEs. Numerical results (Figure 1) well-supports the proposed theorem.

- As the proposed inequality is decomposed with respect to the encoder parameter and the latent variables, it makes it possible to analyze the contribution of each component of the trained model to the generalization error separately.

**Weaknesses**

- The mutual information in the inequality of Theorem 2, is hard to estimate efficiently. Further study is required to investigate how this result can be applied to enhance the generalization of autoencoders.

---

> ### Author Rebuttal · Authors · 2026-03-28
>
> We sincerely thank you for the positive evaluation of our theoretical contributions and for the constructive feedback.
>
> ## Weakness: Difficulty of Mutual Information Estimation and Practical Value
>
> Our primary contribution is not claiming to have perfectly solved the estimation problem, but rather **advancing the information-theoretic (IT) generalization analysis for continuous VAEs to a stage where numerical diagnostics become possible.**
> Prior works (e.g., Kawaguchi et al., 2023; Sefidgaran et al., 2023) focus primarily on supervised or discrete settings. The closest unsupervised analysis (Futami & Fujisawa, 2025 on VQ-VAEs) relies on dataset-wide permutations, resulting in MI terms that depend on high-dimensional joint distributions, making numerical evaluation computationally prohibitive.
> In contrast, by utilizing the LOO-CMI framework, the hold-out index $U$ in our bounds is strictly 1-dimensional. While exact estimation remains difficult, this 1D property allows us to design computable proxies (e.g., the influence-function-based proxy for the encoder, the computable upper bound for the latent term, and the direct Monte Carlo estimator discussed in Appendix K.2). Developing tighter and more scalable estimators is a crucial direction for future work, and we will emphasize this context and limitation in the revised manuscript.
>
> ## Q.1: Details on Table 2 Results
>
> We will clarify this exact procedure in Sec. 3.4 and Appendix J of the revision.
> The correlations were computed split-wise at the maximum training-size setting ($n=30,000$). For each cross-validation split, we computed a single scalar value for the following four metrics based on the trained models:
> - The absolute value of the empirical generalization gap in loss (i.e., $|L_{\text{test}} - L_{\text{train}}|$).
> - The influence-function-based proxy for $I(\phi;U|X^n)$.
> - The computable upper bound of $I(Z^n;U|\phi,X^n)$, evaluated under a uniform prior for $U$.
> - The combined term: $\sqrt{I(Z^n;U|\phi,X^n)+I(\phi;U|X^n)}$, which was subsequently min-max normalized.
>
> We then calculated the Pearson, Spearman, and Kendall correlation coefficients across these split-wise data points. We will explicitly state in the revision that Table 2 represents a diagnostic correlation analysis using computable proxies and upper bounds, rather than an exact direct estimation of the mutual information.
>
> ## Q.2: Applicability to Other Losses
>
> Our current proof relies on the truncation-and-discretization argument for continuous latent variables. This specifically requires: (i) a bounded reconstruction loss (squared loss under Assumption 1), and (ii) an approximate posterior (Gaussian) whose tail behavior can be rigorously controlled.
>
> The essential requirement is not the Gaussian assumption itself, but rather the ability to strictly control the tail and discretization error of the latent space. This allows us to reduce an unbounded latent space into a bounded problem via truncation.
> From this perspective, the proof strategy could potentially be extended to unregularized AEs or Wasserstein Autoencoders, provided that equivalent assumptions can be made (e.g., assuming a bounded $\mathcal{Z}$, or adding explicit tail behavior /continuity controls). For deterministic encoders (like unregularized AEs), controlling the discretization error would require a different set of assumptions, such as a finite data space and continuity of the encoder network.
>
> However, standard Wasserstein Autoencoders do not explicitly assume such latent-space boundedness or tail control. Furthermore, applying our current proof directly to deterministic encoders requires substantial additional analysis. Therefore, we do not claim our results directly apply to any AE objective or variant. We will explicitly document this as a theoretical limitation in the Appendix.
>
> ## Q.3: Practical Use of Theorem 4 in Training
>
> Our primary intent with Theorem 4 is to provide a finite-sample, population-level guarantee, rather than proposing a new training algorithm to be directly optimized.
>
> The theorem demonstrates that the generation error--measured by the Wasserstein distance--is bounded by two empirical quantities already present in VAE objectives: the empirical reconstruction term and the empirical KL term. The practical value of Theorem 4 lies in its role as a theoretical justification and diagnostic principle. It theoretically supports the notion that improving these specific empirical quantities during training can lead to improved generation quality at the population level, offering solid insights for model selection and regularization design.
>
> To ensure we do not overstate its immediate use as an optimization tool while still highlighting its value to practitioners, we will add a "Practical Implications" paragraph in the Conclusion of the revised paper.

---

> > ### Author Rebuttal · Reviewer_uwfe · 2026-04-02
> >
> > I thank the authors for their thorough responses to my questions. My concerns have been adequately addressed, and I will maintain my score.

---

### Decision · Program_Chairs · 2026-04-30

**Decision:**

Accept (regular)

**Comment:**

This paper addresses a meaningful gap in the literature by presenting an information-theoretic bound on the generalization error for Varational autoencoders.  All reviewers agreed on the soundness of the proposed theoretical results which are clearly presented. Their main concerns were related to the practical applicability of the proposed results and to the interest and interpretation of the proposed bound. In particular, the experimental section was considered pretty limited, as it did not sufficiently validate the theoretical bounds in realistic settings.

During the rebuttal, the authors clarified their methodology and better positioned their contribution and theoretical guarantees. They also expanded the empirical evaluation by including experiments with β-VAE using CIFAR-10 and Fashion-MNIST, where varying β explicitly controls latent capacity, offering additional insights on the proposed bounds. Overall, the added experiments, along with the clarifications on the theoretical framework significantly improved the paper. With these improvements, I believe that the revised version is likely to constitute a very interesting contribution.